# Drainage area characterization for evaluating green infrastructure using the Storm Water Management Model

Joong Gwang Lee[1], Christopher T. Nietch[2], Srinivas Panguluri[3]

[1]Center for Urban Green Infrastructure Engineering (CUGIE Inc). Cincinnati, OH 45255, USA.
[2]Office of Research and Development, U.S. Environmental Protection Agency. Cincinnati, OH 45268, USA.
[3]Independent Consultant. Olney, MD 20832, USA.

*Correspondence to*: Joong Gwang Lee (jglee@ugiengineering.com)

**Abstract.** Urban stormwater runoff quantity and quality are strongly dependent upon catchment properties. Models are used to simulate the runoff characteristics, but the output from a stormwater management model is dependent on how the catchment area is subdivided and represented as spatial elements. For green infrastructure modeling, we suggest a discretization method that distinguishes directly connected impervious area from the total impervious area. Pervious buffers, which receive runoff from upgradient impervious areas should also be identified as a separate subset of the entire pervious area. This separation provides an improved model representation of the runoff process. With these criteria in mind, an approach to spatial discretization for projects using the U.S. Environmental Protection Agency's Storm Water Management Model (SWMM) is demonstrated for the Shayler Crossing watershed, a well–monitored, residential suburban area occupying 100 ha, east of Cincinnati, Ohio. The model relies on a highly resolved spatial database of urban land cover, stormwater drainage features, and topography. To verify the spatial discretization approach, a hypothetical analysis was conducted. Six different representations of a common urbanscape that discharges runoff to a single storm inlet were evaluated with eight 24 h synthetic storms. This analysis allowed us to select a discretization scheme that balances complexity in model set-up with presumed accuracy of the output with respect to the most complex discretization option considered. The balanced approach delineates directly and indirectly connected impervious areas, buffering pervious area receiving impervious runoff, and the other pervious area within a SWMM subcatchment. It performed well at the watershed scale with minimal calibration effort (Nash–Sutcliffe coefficient = 0.852; $R^2$ = 0.871). The approach accommodates the distribution of runoff contributions from different spatial components and flow pathways that would impact green infrastructure performance. A developed SWMM model using the discretization approach is calibrated by adjusting parameters per land cover component, instead of per subcatchment, and, therefore, can be applied to relatively large watersheds if the land cover components are relatively homogeneous and/or categorized appropriately in the GIS that supports the model parameterization. Finally, with a few model adjustments, we show how the simulated stream hydrograph can be separated into the relative contributions from different land cover types and subsurface sources, adding insight to the potential effectiveness of planned green infrastructure scenarios at the watershed scale.

# 1 Introduction

Conventional stormwater modeling has focused on the design of urban drainage systems and flood control practices that achieve fast drainage and reduce risk of flooding (NRC, 2009; WEF–ASCE, 2012). These objectives focus attention on larger storms, such as 2 to 10 yr return period storms for designing drainage systems and 25 to 100 yr storms for designing flood control practices (WEF–ASCE, 2012). Conversely, nearly 95 % of pollutant runoff from urban areas is produced from events smaller than a 2 yr storm (Guo and Urbonas, 1996; Pitt, 1999; NRC, 2009). It is well recognized that the best way to resolve this pollution problem is to implement controls as close to the source of runoff generation as possible (Debo and Reese, 2003; WEF–ASCE, 2012).

Green infrastructure (GI) practices were developed to correct this water pollution problem and restore the natural hydrologic cycle (WEF–ASCE, 2012; USEPA, 2014). GI includes structures like green roofs, rain barrels, bioretention areas, buffer strips, vegetated swales, permeable pavements, and infiltration trenches, or practices, such as, disconnecting downspouts. The specific design objectives for GI include minimizing the impervious areas directly connected to the storm sewer, increasing surface flow path lengths or time of concentration, and maximizing onsite depression storage and infiltration at the lot–level (WEF–ASCE, 2012). This translates operationally to individual stormwater management practices that are relatively small but densely distributed in space (USEPA, 2009). Although GI is distributed at higher spatial densities, each unit is relatively inexpensive if the unit can be considered as part of landscaping, and in total, may provide a cost–effective alternative to more traditional larger centralized practices, like detention ponds especially in cases where land is not available or very expensive.

There is a great deal of interest in modeling GI effects at watershed scales to help inform regional stormwater management planning and design decisions. However, from a stormwater modeling perspective, the approach taken for model representations of GI requires different methodological considerations compared to the traditional large–size, low spatial density of the more centralized and regional control features (Fletcher et al., 2013; USEPA, 2012; Guo, 2008). While conventional stormwater management practices have focused on end–of–pipe controls at the downstream end of the drainage area (i.e., centralized systems), GI practices focus on on–site controls at the upstream side (i.e., distributed systems). The surface hydrologic properties (e.g., land cover, slope, overland flow path, etc.) remain the same before or after applying centralized systems, but they are altered with on–site GI systems. GI practices aim to amend the landscape hydrologic properties to reduce the negative impacts of stormwater (USEPA, 2007). Hence, modeling approaches for evaluating GI should be able to account for the changing surface hydrologic properties that come with GI implementation.

The Storm Water Management Model (SWMM) of the United States Environmental Protection Agency (USEPA) is one tool that has a large user–base and a broad application history for informing stormwater management projects around the world (Niazi et al., 2017). In the current version of the model, GI effects are simulated using low impact development (LID) algorithms. LID is largely synonymous with GI in SWMM vernacular. The LID modeling options were added in 2010 (Rossman 2015; Rossman and Huber, 2016). Since then, while numerous LID/GI modeling studies have been introduced, best

modeling practices for simulating GI in SWMM have received comparatively little attention in the literature (Niazi et al., 2017).

This study was intent on evaluating approaches to modeling GI effects at a watershed scale using SWMM. In the set–up of a SWMM model, the urban area of interest is divided into smaller spatial units, referred to as subcatchments. To implement a traditional stormwater control feature like a retention pond, it is usually acceptable to provide minimal detail of the drainage area (Rossman and Huber, 2016). This leads to spatial aggregation which tends to produce larger subcatchment areas that aggregate land cover types and simplify the existing storm sewer system to realize a more cost–effective model set–up and output data management. If the simulated hydrographs are matched with the observed data during model calibration, the model is considered a sound representation of the drainage processes important for designing the retention pond. The trade–off is that coarser schematization requires more decisions on how to aggregate catchment properties (Rossman and Huber, 2016). In contrast, for simulation of GI, the construction reality is that GI is built as part of building and landscape arrangements all upgradient of the drainage network (Dietz, 2007; Montalto et al., 2007; USEPA, 2009; Zhou, 2014). Therefore, to accurately examine GI alternatives, a drainage area for modeling should be defined as an area that drains runoff to a storm sewer inlet with no or minimal spatial aggregation of landscape features affecting hydrologic properties. In SWMM, this drainage area is modeled as single or multiple subcatchments. Since SWMM version 4.4H, overland flow routing is allowed from one subcatchment to another (Huber, 2001; Huber and Cannon, 2002). To minimize confusion between real and modeled drainage areas, we coin the term hydrologic response element (HRE) in this study. An HRE is the drainage area (i.e., a real spatial element of the landscape being modeled) where GI practices may be implemented to control the element's surface runoff prior to discharge to the stormwater collection system. There are many alternatives for configuring the HRE for GI modeling in SWMM. We evaluate six different options in this study.

In SWMM, the subcatchment representation of the HRE is comprised of one or more homogeneous subareas, such as impervious or pervious area, impervious area with or without depression storage, directly connected impervious area (DCIA) or indirectly connected impervious area (ICIA), or LID area (Rossman, 2015; Rossman and Huber, 2016). In urban landscapes, DCIA discharges runoff to the existing storm sewer system without any control, while ICIA discharges to adjacent pervious area (PA). The PA that receives runoff from ICIA works like a buffer strip or swale, therefore acting like an existing GI practice albeit not intentionally designed as such. This is a real characteristic of urban areas that is termed buffering pervious area (BPA) in this study. The other pervious area is called standalone pervious area (SPA) that does not receive or control any runoff from impervious area. An HRE may consist of all or part of these subareas – DCIA, ICIA, BPA and SPA – and implementing GI practices can change subarea ratios and properties, surface runoff processes, and flow pathways within the HRE. Each subarea may also consist of different land cover components. For example, DCIA may include paved streets, building rooftops, driveways, or sidewalks, . In this study, we questioned how these spatial and hydrologic realities should be modeled using SWMM.

After the HRE delineation is performed in SWMM, each HRE undergoes a model parametrization procedure that defines the relative proportions of impervious and pervious subareas, how they interact in terms of surface flow pathways, and their

hydrologic properties (Rossman, 2015). The subcatchment/subarea configuration of each HRE ultimately specifies the physical conditions used by the model's mathematical algorithms to simulate the dynamics of hydrologic loading to the drainage network. The more subcatchments there are, the more input and output values there are to be managed by the modeler. When setting up a SWMM model using the conventional objectives, such as deriving hydrographs for designing storm collection

systems and/or detention/retention systems, the subcatchment parameterization remains the same before and after simulation of the management practice; however, for GI simulation, the internal properties of a subcatchment change, as mentioned earlier. Pending the type of GI, changes may need to be made to the hydrologic properties of subareas or individual land cover components, the proportions of impervious and pervious area, the specification for the routing of runoff between them, the flow path length, and infiltration or the depression storage properties. Adequately rationalizing and tracking these changes can

become a problem for the modeler when the total area being modeled is relatively large and aggregated, the GI scenarios are not the same among HREs, or the internal properties among HREs are heterogeneous. A systematic approach to characterizing HREs would help make SWMM GI simulation projects more efficient.

The question of how to best parameterize SWMM is not new especially when it comes to spatial resolution and scaling, but as mentioned, GI modeling, in particular, requires special considerations. This paper describes a suggested method for modelling

GI in SWMM and provides critical evaluation. Primary objectives of this study are to (1) examine how to configure HREs for GI modeling and (2) develop a methodology for parameterizing a SWMM model that reflects this configuration with the goal of demonstrating an urban watershed spatial discretization approach that optimizes model performance in terms of tracking model input values and presumed accuracy of the results. We hypothesized that conventional modeling approaches to subcatchment delineation are likely aggregating at too coarse resolution in space and hydrologic response to be appropriate

for highly spatially distributed modern GI. We also questioned how the SWMM setup could not only allow for modeling the effects of various GI scenarios, but also facilitate the scaling of GI scenarios from a small HRE, representing the parcel or lot–level, to a watershed level. To answer these questions, we examine several acceptable approaches to representing spatial reality in SWMM when the modeling objective is to inform decisions about GI implementation. We use a hypothetical HRE based analysis of spatial discretization alternatives to test our hypothesis related to the appropriateness of spatial and hydrologic

response resolution. The hypothetical HRE represents a typical residential area that drains to a storm sewer inlet. The hypothetical HRE is modeled in SWMM by combining six conceivable options in spatial representation and eight design storms that represent the full spectrum of runoff events. From this analysis, an appropriate option is selected for GI modeling in SWMM, and a baseline SWMM model is developed using it for representing the existing condition of a well–characterized 100 ha urban watershed in a headwater area east of Cincinnati, Ohio. To examine the proposed approach, another SWMM

model is developed for simulating a GI implementation scenario at the study watershed. Also, an approach for hydrograph separation is presented using the developed SWMM models, which can provide insight for arranging GI implementation scenarios. Currently, there is no provision for accomplishing this in SWMM.

## 2 Materials and methods

### 2.1 Study area

An experimental urban watershed drained by a natural headwater stream that does not have any surface stormwater inflows from outside its topographic boundaries was used for this study (Fig. 1). The Shayler Crossing watershed (SHC) is located east

of Cincinnati, Ohio and occupies approximately 100 ha that is characterized as 62.6 % urban or developed, 25.6 % agriculture, and 11.8 % forested based on the 2011 National Land Cover Database (Homer et al., 2015). The native soils of the watershed are characterized with high silty clay loam content and therefore are naturally poorly infiltrating.

**Figure 1. Location of the Shayler Crossing watershed. I-275 is an interstate highway around the Cincinnati metropolitan area.**

### 2.2 The baseline spatial database

### 2.2.1 Data from the County GIS

Spatial data for the study area was provided by the Clermont County Office of Environmental Quality, which included a detailed GIS of the existing stormwater drainage system and surface topography. The drainage system consists of storm sewer inlets (or catch basins), manholes, pipes, wet/dry detention ponds, and channel network. The County GIS contains the location

of the drainage system, invert elevations for inlets and manholes, and pipe sizes. Two types of surface topography data were also available; 0.76 m (2.5 feet) LiDAR (Light Detection and Ranging) data and 0.3 m contours. High–resolution aerial orthophotographs were also provided by the County. Existing databases that include the details for the stormwater infrastructure like in this watershed are not always available to the modeler. In these cases, to adopt the subsequently described approach to GI scenario modeling in SWMM could require considerable ground–truthing and site surveying. In lieu of onsite

visits, and as will become apparent from the descriptions below, what would be most important is determining the spatial location of storm sewer inlets. These are often visible from readily available aerial photographs, note that the visibility depends on the underlying image quality and the presence of obstacles such as trees or cars. When elevation data for the storm sewer network is unavailable, much can be inferred using surface elevation data and assuming local construction codes for stormwater infrastructures were applied, such as catch basin depths and conveyance pipe diameters and slopes. Such approximations would

suffice for GI scenario analysis considerations and where storm sewer design is not the primary focus.

### 2.2.2 Detailed land cover and subarea categorization

To obtain a high resolution digital characterization of spatial reality in the study watershed, 16 unique land cover types were identified and digitized using ArcGIS 10.2 (ESRI, 2013) spatial analysis tools on the aerial orthophotographs of the study area. These 16 types are later aggregated to 10 for setting-up the watershed SWMM model (See Appendix). The resulting baseline

spatial database included individual records of the watershed surface that could be used to access the location, pattern, and extent of the following sixteen land cover types: streets, parking areas, sidewalks, driveways, main buildings, miscellaneous

buildings, paved walking paths, patios, other miscellaneous impervious areas, landscaped or lawn areas, agriculture, forest, dry ponds, stormwater detention area (in SHC this is created by the addition of a control structure to the stream channel itself), swimming pools, and wet ponds. Each spatial record has its own attributes (i.e., fields in the database) representing the current conditions (e.g., area, land cover) and was characterized based on its future potential for GI implementation (e.g., to evaluate

the potential of downspout disconnection for a main building). The initial parameterization and GI modeling approaches described below for the SWMM model are based on content extracted from this land cover database created using ArcGIS tools. This database is often reused to perform model adjustments during calibration and GI scenario analysis. The developed land cover database for SHC contains a total of 3682 records and the median area of each record is 23.5 m$^2$.

Each surface record in the database is further classified into four types based on its hydrologic characteristics including 1)

DCIA, 2) ICIA, 3) Pervious area (PA), or 4) Water. The PA is subsequently split into two subcategories called BPA and SPA after the HRE delineation procedure for SWMM modeling is completed (see below). All main buildings are DCIA because the rooftop downspouts in the existing condition are plumbed to directly discharge to the storm water collection system through buried pipes or street gutters. All the miscellaneous buildings (e.g. storage sheds) are considered ICIA. Streets with curb–and– gutter drainage systems are identified as DCIA. Any directly connected upgradient impervious areas to these streets are initially

considered as DCIA. These areas include directly connected driveways, parking areas, and sidewalks. However, if both sides of a sidewalk are surrounded by pervious area, the sidewalk is categorized as ICIA. Streets without curb–and–gutter drainage are ICIA. The remaining miscellaneous impervious areas are ICIA.

Figure 2 contains a sample GIS representation of the 16 land cover types along with a corresponding attribute table, which indicates hydrologic characteristics representing the baseline classification and a GI scenario–related classification. In the

attribute table shown in Fig. 2, the first column contains the record identifier, the second column defines the land cover type, the third column defines how it was classified for modeling the baseline condition, the fourth column defines how it was classified or re–classified for modeling a specific GI scenario, and the fifth column specifies the contributing area. For example, the record ID 36 contained in the table is initially classified as DCIA, but after the rooftop drains were disconnected in the modeled GI scenario, the unit was reclassified as ICIA (in the fourth column). This methodology allows for GI–related

hydrology evaluation to be performed without impacting the overall SWMM model structure and setup. A companion USEPA report (Lee et al., 2017) has been prepared to provide the relevant details on the applied spatial analysis techniques such as clip, intersect, union, buffer, and manipulating attribute data.

**Figure 2. Sample GIS classified representation of the land cover and hydrologic characteristics.**

**2.2.3 Configuring the BPA and SPA**

BPA is not considered explicitly in a traditional urban stormwater modeling analysis using SWMM. Instead the modeler usually sets up PA within a subcatchment to receive a certain percentage of runoff from impervious areas; this is how ICIA is distinguished from DCIA. However, in reality, not all of the PA receives runoff from ICIA, rather just the part of the PA that

is immediately adjacent to the ICIA. When evaluating GI scenarios, one strategy might be to enlarge the size of the buffering area adjacent to ICIA, or engineer GI structures (e.g., cascading filtering or bioretention systems) around this buffering area (a.k.a. BPA) to reduce the direct runoff from impervious surfaces by routing them over grassy areas to slow down runoff and promote soil infiltration. Draining paved areas onto porous areas can reduce runoff volumes, rates, pollutants, and cost for

drainage infrastructure (NRC, 2009; WEF–ASCE, 2012). Therefore, because of the nuanced, yet important differences in the geospatial relationship of PA in different GI scenarios, we rationalized the need for retaining the ability to model this aspect while evaluating GI scenarios by splitting the PA into BPA and SPA for GI modeling in SWMM.

Characterizing the precise "physical" extent of BPA is a complicated process that would have to be defined from highly resolved surface topography around ICIA and an understanding of the unsaturated zone processes such as how infiltration and

depression storage interact across the pervious surface types to influence flow path length. The physical extent of BPA is also affected by storm intensity, with higher intensity storms creating a larger spread of water across the surface and thereby increasing the extent of available adjacent buffering areas. Lacking the ability to infer flow path length without extensive physical measurements, we instead treat the width of the BPA from ICIA as a calibration parameter. In preparation for this, BPA based on different buffer widths was established during the development of the spatial database. This was done in ArcGIS

using the geoprocessing tools "Buffer" and "Intersect". The "Buffer" tool established separate BPA area around all existing ICIA based on arbitrarily chosen distances that serve as equivalent "buffer widths" of 0.30, 0.61, and 1.52 m (Fig. 3). The "Intersect" tool establishes the area for the BPA and adjusts the area of the original pervious area from which it was subtracted, which is now SPA (Lee et al., 2017). Using this spatial information, we arranged three SWMM models that represent three different sizes of BPA. We determined which one among the three cases of BPA sizes provided the more accurate simulation

compared to the observed flow data, and as part of model calibration. In this way the BPA width was treated as a calibration parameter in this study.

**Figure 3. Depiction of the different distances applied for the estimation of BPA in the baseline condition using ArcGIS.**

**2.3 HRE delineation**

Urban HREs were delineated manually within the GIS using the surface topography (0.76 m LiDAR) and the layout of the storm sewer system (Rossman and Huber, 2016). Because GI is designed to capture and control stormwater runoff before it discharges to the storm sewer system, the HRE for GI analysis should be delineated as the area that drains runoff to an actual storm sewer inlet. With GI implementation, some inlets can be removed or combined for economic benefits because the peak and volume of stormwater discharge will be decreased after implementing GI practices (Sample et al., 2003; Braden and

Johnston, 2004; USEPA, 2012). Based on this, two HREs are combined into one HRE if the two HREs were located side–by– side at one street location, and one of the two HREs was smaller than 2023.4 m² (0.5 acre). For undeveloped or agricultural areas in the study watershed, the HRE boundaries were generally selected with an intent to keep all HREs a similar size to help

maintain hydrologic continuity among them. The result of the HRE delineation for the entire SHC watershed is shown in Fig. 4.

**Figure 4. Detailed spatial representation of the Shayler Crossing watershed.**

**2.4 SWMM parameterization**

SWMM developed by the USEPA, is a comprehensive mathematical model for analyzing hydraulics, hydrology, and water quality process dynamics in the urban environment (Huber and Dickinson, 1988; Gironás et al., 2009; Rossman, 2015; Rossman and Huber, 2016, Niazi et al., 2017). Here version 5.1.007 of SWMM was used. SWMM generates runoff when rainfall depth exceeds surface depression storage and infiltration capacity at the subcatchment scale. SWMM has extensive

routing capability that can simulate the runoff through a conveyance system of pipes, channels, storage/treatment devices, pumps, and regulators. SWMM can also estimate the quality of runoff discharging from subcatchments and route it through the conveyance system. The model can be used within a continuous or event–based framework.

Unique to our application of the SWMM model is the set-up of the BPA. This process is described in detail in Appendix. Because the natural stream draining the study area receives lateral inflow through subsurface soil media (a.k.a., subsurface

flow), SWMM's groundwater modeling options were implemented. The groundwater component of the SHC SWMM set-up is also described in Appendix.

A subcatchment is a fundamental hydrologic component of a SWMM application, and can be defined as an area that drains runoff to a storm sewer inlet, open channel, or another subcatchment. The SWMM subcatchments in this study will represent the HREs that were delineated during the development of the spatial database described above. Each SWMM subcatchment is

configured with a specific drainage area, % imperviousness, width, and slope. Subareas divide each subcatchment into impervious, pervious, and/or LID areas that are used to account for internal heterogeneity. These areas are modeled in the abstract based on the relative percentage of the subcatchment each occupies, i.e., subareas have no real spatial reference. Therefore, all pervious areas within one subcatchment, for example, are lumped and modeled as one contributing hydrologic entity no matter how disconnected or patchy the actual physical reality may be. This establishes a relationship between the

subcatchment size and the spatial resolution of the model. The larger the subcatchment area, especially in the urban environment, the more spatial lumping that results, and the more abstracted from reality the model becomes. The size of the subcatchment and the heterogeneity among land covers and their organization within each subcatchment or subareas interact to effect model complexity as well as accuracy. In most cases, modelers try to strike a balance between these when configuring a SWMM project. Subareas are parameterized by setting values characteristic of each, such as $n$ and $DS$ for both IA and PA.

The Green–Ampt option for infiltration modeling was used in this study, and this requires three parameters per subcatchment's PA including, the saturated hydraulic conductivity ($K_{sat}$), capillary suction head ($Suct$), and initial soil moisture deficit ($IMD$). Internal flow between the subareas can be routed from pervious to impervious, impervious to pervious, or directly to the outlet. LID areas have their own set of parameters.

The spatial database that included land cover digitization and HRE delineations was used to parameterize the SWMM model. With the land cover data spatially overlaid with the HRE delineation in ArcGIS (Fig. 4), the characteristics of each SWMM subcatchment could be defined using the detailed land cover status per subcatchment and unique hydrologic parameters per land cover component presented in Table 1. Each land cover type is either all impervious or all pervious. "Length" represents

a typical distance for overland flow before it turns into a concentrated flow path, which is controlled by the hydrologic design features of the land cover type. For example, overland flow at a rooftop is maintained only from the roof crest to the gutter because flow through a gutter is considered concentrated. The same regime change in flow (i.e., from overland flow to concentrated flow) may happen at any place where more than one instance of impervious land cover converge hydrologically, e.g., at a street gutter where overland flows from streets and driveways intersect. Using ArcGIS, the initial values for "Length"

were determined by averaging multiple field measurements of perceived overland flow lengths for each land cover type. More detailed procedures for the SWMM modeling methods used in this study are presented in Appendix.

**Table 1. Initial and calibrated modeling parameters for the Shayler Crossing watershed.**

## 2.5 Model set–up options for a hypothetical HRE in SWMM

As mentioned earlier, an HRE can be modeled as a single subcatchment or multiple subcatchments in SWMM. In the SWMM model set-up just described we used a single subcatchment set-up that was based on the results of an analysis done with the goal of determining which among a series of plausible HRE configuration options strikes a balance among the degree of spatial and hydrologic aggregation, output uncertainty, and computational effort. The most spatially refined approach to a SWMM set–up (Option 1 in this study as presented below) would be to discretize every piece of impervious and pervious surface as an

independent subcatchment. This promises a decrease in model output uncertainty (Krebs et al., 2014; Sun et al., 2014), but requires specifying all the modeling parameters and unique flow directions among all subcatchments, results in longer computational times, and produces data management burdens that are typically not practical. The opposite extreme would be a highly generalized subcatchment characterization where the entire area is modeled as one subcatchment with just two subareas, lumping all the spatial heterogeneity into a fictional space that has no basis in physical reality. Within this continuum,

we chose to consider six plausible options for representing urban spatial constructs that are constrained by the SWMM subcatchment/subarea paradigm were examined (Fig. 5). As shown in the legend of Fig. 5, each rectangle represents a subcatchment in SWMM, and the dotted line divides subareas within the subcatchment. A rectangle without a dotted line means the subcatchment consists of a single (homogeneous) subarea, either 100% impervious or pervious. The arrows represent flow routing directions. Conducting this assessment at the watershed scale would not only be tedious and time

consuming to configure, but could be inappropriate because of potentially confounding effects introduced when the drainage network and groundwater algorithm are included in the simulation. We felt it more rational to base our assessment of HRE set-up options at the scale of an HRE, i.e., the area that drains to a storm sewer inlet, and judge the results in comparison to

the most spatially explicit option. Note, we do not have supporting observational data at this scale to prove this assumption. This would require flow data at the point of entry to a storm sewer inlet, which is very difficult to obtain in practice.

Instead, a hypothetical representation of a typical urbanscape was defined as the HRE and used to model eight synthetic single storm events for each of the six set–up options (Fig. 5). The hypothetical HRE is meant to represent a typical 4041 $m^2$ (1 acre)

residential area consisting of 809.4 $m^2$ (0.2 acre) DCIA, 1214.1 $m^2$ (0.3 acre) ICIA, and 2023.4 $m^2$ (0.5 acre) PA. The DCIA consists of 607.0 $m^2$ (0.15 acre) transportation–related surfaces (e.g., streets, driveways) and 202.3 $m^2$ (0.05 acre) building rooftops. The runoff from ICIA discharges through 404.7 $m^2$ (0.1 acre) BPA, thus the SPA of the area is 1618.7 $m^2$ (0.4 acre).

**Figure 5. A conceptual representation of the hypothetical HRE (20% DCIA, 30% ICIA, 10% BPA and 40% SPA) and the 6 options**
**considered for representing this area in the set–up of a SWMM model.**

Referring to Fig. 5, in Option 1, five subcatchments are arranged for modeling the hypothetical HRE, separately modeling transportation DCIA (Trpt) and building DCIA (Bldg) along with ICIA, BPA, and SPA. DCIA is modeled with two sub–groups because buildings have slanted rooftops while paved areas for transportation are basically flat in a typical residential area. This is the lowest level of spatial aggregation among the six options. This option would result in the highest number of

subcatchments, and, therefore, number of data requirements. Option 2 combines the two DCIA subcatchments in Option 1, resulting in four subcatchments set–up for one HRE. In Option 3, the four subcachments in Option 2 are aggregated into two subcatchments, and each subcatchment is configured with two subareas. The imperviousness and the flow direction between subareas per subcatchment need to be specified in SWMM with "% Imperv", "Subarea Routing", and "Percent Routed". "Impervious" option for "Subarea Routing" means runoff from pervious area flows to impervious area whereas "Pervious"

does the opposite (Rossman, 2015). "Percent Routed" should be specified as 100 for both subcatchments. In Options 4 through 6, the areas are further aggregated to a single subcatchment representation for an HRE in SWMM. Option 4 configures the single subcatchment with only two subareas, impervious and pervious areas. The runoff from pervious area discharges through impervious area (i.e., TIA = DCIA). In Option 5, DCIA and ICIA are independently modeled by specifying the "Subarea Routing" option as "Pervious" and the "Percent Routed" as the ratio of ICIA/TIA. This option may be considered an unrealistic

'green' development condition where runoff from ICIA is evenly distributed throughout the entire pervious area, which means the entire pervious area works like a buffer (i.e., TPA = BPA). Finally, in Option 6, LID controls in SWMM are used for modeling BPA and ICIA. BPA is modeled as a vegetated swale with a very small berm height, 2.54–mm (0.1–inch). In the "LID Usage Editor", the "Area of Each Unit" specifies the size of BPA and the "% of Impervious Area Treated" is the fraction ICIA/TIA. With this configuration, the four hydrologically homogenous subareas – DCIA, ICIA, BPA, and SPA – are

accounted for.

Lengths for overland flow (or sheet flow) were assumed to be 4.57 m (15 feet), 9.14 m (30 feet), 12.19 m (40 feet), and 15.24 m (50 feet) for transportation related DCIA, building rooftops as DCIA, ICIA, and pervious area, respectively. The surface slopes of these were assumed to be 3 %, 11 %, 5 %, and 2 %, respectively. Surface dimensions and slopes of typical urban land cover components are based on construction codes or were inferred based on the GIS. The values selected are meant to

represent typical residential areas in the United States. For example, the assumed values were derived using overland flow from the center of the street to the curb in a crowned 9.14 m (30 feet) wide neighbourhood street with 3 % cross–sectional slope for the crown, 18.29 m (60 feet) wide gable houses with 11 % cross–sectional slope for the rooftops, and pervious surfaces with 2 % slope on average. Every IA is modeled with 0.01 for Manning's roughness coefficient (*n*) and 2.54 mm (0.1 inch) for depression storage (*DS*). Pervious area is modeled with 0.1 for *n* and 5.08 mm (0.2 inch) for *DS*. Identical infiltration parameters were applied to all the options. In this hypothetical HRE model set–up analysis, all of the six options were arranged using the same spatial and hydrologic characteristics. However, the ways DCIA, ICIA, BPA, and SPA were parametrized in SWMM were different among the options (Fig. 5).

**Table 2. Profile of the selected eight 24 h single storm statistics.**

Rainfall–runoff response is also affected by storm size, so we applied eight different 24 h single storms (Table 2) selected from a regional rainfall frequency report produced by the National Oceanic and Atmospheric Administration (NOAA) and the Illinois State Water Survey (Huff and Angel, 1992). Another data set was used to estimate the "Percentile" and "Cumulative" rainfall depths per year, i.e., annual statistics per 24 h storm. This data set covered about 35 years of hourly precipitation records from a local weather station in Milford, Ohio. A certain percentile rainfall event represents a precipitation amount that the same percent of all rainfall events for the period of record do not exceed (USEPA, 2009). The percentile values in Table 2 were estimated using the method presented in the same report (USEPA, 2009). For example, the 90th percentile rainfall event is defined as the measured precipitation depth accumulated over a 24 h period for the period of record that ranks as the 90th percentile rainfall depth based on the range of all daily event occurrences during this period. Values in the "Cumulative" column of Table 2 represent the percentage of annual cumulative precipitation depth, which are less than or equal to the specific rainfall depth during a 24 h period. In SWMM, the selected storms were distributed with 5 min intervals by applying the Natural Resources Conservation Service (NRCS) Type–II distribution (USDA, 1986).

## 2.6 Calibration of the SHC watershed SWMM model

Stream flows were measured at the outlet from a rating curve using water depth recorded at 10 min intervals. A tipping bucket rain gauge measured rainfall depths at 10 min intervals, with a minimum detectable rainfall depth of 0.254 mm (0.01 inch). The SWMM model for SHC (Fig. 6) was run for a six–month period (01 April 2009 to 31 August 2009) where the first four months of this period were used to stabilize the continuous simulation, in particular for the groundwater simulation. This is defined as the model 'warm–up' period, which is the time period required to achieve a stable condition wherein the groundwater level ceases to increase or decrease by a specified initial parameter threshold value. After the warm–up period, the last two months, from July to August 2009, were used for model calibration. Model calibration was done manually by adjusting the initial values for the 10 land cover types, and using the different sets of BPA (see Fig. 3.). Changes were integrated one at a time into every subcatchment using the area–weighting approach in an Excel spreadsheet. The calibrated modeling parameters for individual land cover types are given in Table 1 alongside their initial values. An Excel worksheet was created

with embedded look–up and averaging functions so that changes made to the original values in Table 1 or switches between BPA sets configured using the different buffer distances could be easily propagated to changes in the related parameter values used in the SWMM model using the SWMM Excel Editor function. With this approach, the calibration effort is evenly applied to the urban land cover types, which in turn are propagated to the parameterization of all subcatchments, instead of calibrating parameters individually for each subcatchment. This methodology assumes that urban land cover components are generalizable, and independent from scale even though the subcatchments themselves are not generalizable or easily scalable. Also, notable about this approach, the parameter calibration domain remains the same even if the total number of subcatchments is increased and/or the size of watershed area is increased. If a land cover type does not maintain a sufficient level of homogeneity across the watershed under study, we would need to divide the land cover into sub–categories and use more than one set of parameters for the land cover type in each category. For example, flat rooftops in commercial areas may need to be differentiated from slanted rooftops in residential areas, high and low sloped hillslope categories may need to be characterized in watersheds with a large range in topographic relief, or categories to account for different hydrologic soil types with different infiltration properties. This can be handled by spatial analysis in GIS by overlaying land use, topography, or soil property data with the land cover layer.

**Figure 6. Diagram of the developed SWMM model for the Shayler Crossing watershed.**

Sensitivity analysis was conducted for the modeling parameters width, slope, $n$ and $DS$ for IA and PA respectively, $K_{sat}$, and the size of BPA. Each parameter was decreased and increased 5, 10, and 20 %, respectively, one at a time, and in separate model runs. The sensitivity of each parameter was estimated as:

$$Sensitivity = (\Delta MR/MR)/(\Delta p/p) \tag{1}$$

Where, $MR$ = modeling result in units of flow volume from the SWMM run; $\Delta MR$ = change in SWMM modeling result based on change in parameter value; $p$ = parameter value; and $\Delta p$ = change in parameter value.

**2.7 Modeling GI scenarios**

GI scenarios are added to the model using the land cover database, soils, storm sewer systems, GIS techniques to derive relevant BPA, and may require some field investigation to ground truth the options. The general workflow for GI modeling are presented in bottom half of Fig. A-1. Implementing GI can be achieved by adjusting the hydrologic properties of individual land cover components, such as converting lawn area to shrub or forest (Lee et al., 2005). This sort of GI implementation can reduce the volume, peak, and speed of surface runoff and be modeled by adjusting $DS$, slope, $n$, or overland flow length for the converted land cover component. The one scenario we examined was decreasing DCIA by disconnecting the directly connected rooftop downspouts that directly route flow from the main buildings to the sewer system. This effectively reclassifies main buildings as ICIA. After the downspouts are disconnected, the PA that receives stormwater runoff from the disconnected rooftop now works as additional BPA. To model this additional buffering capacity, the size of BPA is re–estimated and the

percent of IA routed to BPA is changed in SWMM. The increase in size of the BPA under this GI scenario was estimated again using the spatial analysis tools in ArcGIS by changing the buffering distance value from the calibrated baseline value of 0.61 m (2 feet) to 3.1 m (10 feet) around ICIA, including the disconnected main buildings. As a result, the modeled GI scenario includes two types model changes: One that reflects the downspout disconnection and another the buffering area extension.

The characteristic width per subcatchment is a computed value that is usually treated as a calibration parameter in SWMM (see Appendix). Under conventional stormwater management modeling approaches, once the width value is set, it is not adjusted during management scenario analysis. However, GI, by design, changes the flow paths lengths and therefore the computed value of the "width" parameter as represented in SWMM should also change. The methodology we present here provides a systematic way of changing the width parameter in a rational and objective manner to account for the modeled GI

scenario. Unfortunately, the suitability of this modeling approach cannot be determined until a high density of GI has been implemented at a watershed scale with before and after field observations.

## 2.8 Hydrograph separation

With the approach taken for the SWMM set–up for both the baseline and GI scenario analysis adjustments can be made to apportion the simulated storm hydrologic loading from the watershed among the dominant sources: DCIA, ICIA+BPA, SPA,

and Subsurface flow. This can provide further insight into the effects of GI on watershed hydrology. For this purpose, the output from four SHC–SWMM runs were generated:

Run 1) Every subcatchment is specified as described under Option 6 (as conceptually represented in Fig. 7a) with groundwater options parameterized to represent the base SWMM model.

Run 2) Groundwater options were excluded from the base model set–up to remove any subsurface flow contributions to the

stream flow hydrographs. The difference between 1) and 2) represents the stormwater contributions to the stream as subsurface flow from the watershed.

Run 3) To estimate surface runoff from all impervious areas (i.e., runoff from DCIA, plus ICIA through BPA) in the models without the groundwater, the SPA was also omitted from every subcatchment (Fig. 7b).

Run 4) To estimate surface runoff from DCIA, only DCIA was modeled in this run (Fig. 7c).

An example result of the hydrograph flow pathway separation is presented in Fig. 7d, and the process is summarized mathematically as follows:

$$Q_{total} = Q_{DCIA} + Q_{ICIA+BPA} + Q_{SPA} + Q_{interflow} \tag{2}$$

$$Q_{surface} = Q_{DCIA} + Q_{ICIA+BPA} + Q_{SPA} \tag{3}$$

$$Q_{interflow} = Q_{total} - Q_{surface} \tag{4}$$

$$Q_{SPA} = Q_{surface} - Q_{imperv} \tag{5}$$

$$Q_{ICIA+BPA} = Q_{imperv} - Q_{DCIA} \tag{6}$$

Where, $Q_{total}$ = total runoff with groundwater flow in SWMM; $Q_{surface}$ = surface runoff without groundwater flow; $Q_{imperv}$ = runoff from impervious area (DCIA and ICIA through BPA); $Q_{DCIA}$ = runoff from DCIA only; $Q_{ICIA+BPA}$ = runoff from ICIA and BPA; $Q_{SPA}$ = runoff from SPA only; and $Q_{subsurface}$ = runoff through groundwater flow (i.e., subsurface or lateral flow).

**Figure 7. Conceptual representations of discrete SWMM models for hydrograph separation.**

## 3. Results and discussion

The SHC watershed model parameter values pre– and post–calibration are presented in Table 1.

### 3.1 Spatial analysis

Table 3 reveals the results of the detailed spatial analysis conducted using the described GIS techniques. The fractional DCIA for buildings, streets, driveways, parking areas, and sidewalks are 96.1 %, 79.5 %, 94.2 %, 42.8 %, and 14.2 %, respectively. Overall, the study watershed is covered by 18.8 % DCIA, and three sets of BPA were derived for 0.30, 0.61, and 1.52 m buffer lengths. After calibration, the 0.61 m buffer around ICIA was selected for SHC. This means that the runoff from ICIA is discharged to the adjacent pervious area with 0.61 m buffer width best mimicked hydrologic behaviour, based on runoff volume and timing in hydrographs (see Fig. 10 and Fig. 11). This existing buffer covers 22683.5 m$^2$ of pervious area, which is 2.3 % of the entire watershed and 3.0 % of the pervious area. As the baseline, the SHC watershed consists of 18.8 % DCIA, 5.2 % ICIA, 2.3 % BPA, 73.1 % SPA, and 0.6 % Water. Under the modeled GI scenario of disconnecting rooftop drains and extended BPA, the DCIA is reduced to 9.6 %, the ICIA increases to 14.4 %, the BPA increases to 17.2 %, and the SPA is reduced to 58.2 % of the total area, respectively.

**Table 3. Land cover status of Shayler Crossing watershed.**

### 3.2 The hypothetical HRE modeling analysis

The eight single storm–hypothetical HRE modeling analysis with the six discretization options resulted in 48 SWMM runs. As explained earlier, this was done to determine which HRE configuration option best balances model complexity and presumed accuracy. Each simulated storm was assumed to last from midnight to midnight. Results are presented as hydrographs between 11:00 and 13:00 hours where most concentrated rainfall occurs in the NRCS–Type II distribution (USDA, 1986) (Fig. 8). In large storms, larger than a 5 yr storm in particular, all six types of spatial discretization produce very similar hydrographs as shown in (g) and (h) of Fig. 8. The modeled flow rates and total runoff volumes are almost identical.

**Figure 8. Hypothetical HRE SWMM modeling results**

In the large storm situation, all of the PAs are saturated in the early stage of the storm. Once saturated, the PAs are not able to provide any additional onsite hydrologic control, and behave as IA. In view of this, any of the spatial discretization options would be suitable for analyzing flood controls and in designing a drainage system based on a 10 yr storm. However, this is not the relevant case for evaluating GI implementation, which focuses on controlling smaller storms. For storms smaller than a 2
yr event, considerable differences were found among the simulated hydrographs (Fig. 8a through e).

In the smallest storm situation (Fig. 8a) the options for spatial discretization result in almost identical hydrographs except Option 4, where only DCIA discharges runoff, as TIA is modeled as DCIA. Rainfall onto PA is completely captured by *DS* and/or infiltrated to the soils. Because Option 4 ignores the difference between DCIA and ICIA, the entire impervious area (subarea IA) is modeled the same as DCIA, which means all of the runoff is discharged to the storm drainage system directly
with no abatement. Under a small storm (like < 1 month storm), runoff occurs only from IA, more specifically, only from DCIA. For small storms, runoff from ICIA is completely controlled by BPA (if ICIA exists), but no ICIA is modeled under Option 4. Because of this, modeled runoff from this option is higher than any of the other options under the small storms. DCIA is modeled explicitly in the other five options. The relative difference in runoff estimates caused by modeling TIA as DCIA contribution diminished as larger storms are modeled, Fig. 8a to c. Option 4 is not suitable to modeling GI alternatives
because it ignores the significance of characterizing DCIA and ICIA within an HRE. Option 5 shows the most significant variation among the simulated hydrographs. This option estimates lower flow rates than the others for smaller storms, but higher peaks in medium–size storms (as 6 month to 2 yr return period storms; Fig. 8d through f). Option 5 is configured to simulate the "ideal" green implementation scenario of surface grading for stormwater discharge, in which the entire pervious area works like BPA. The expanded onsite pervious buffer can thoroughly control runoff from ICIA until the *DS* and infiltration
capacity of BPA are fully saturated. Once the hydrologic capacities for onsite controls are fully saturated, the entire PA hydrologically responds more or less like IA. Once a subcatchment *DS* fills and exceeds infiltration capacity, this unrealistic 'green' development condition may result in higher peak discharges than the other options.

From the hypothetical modeling analysis, it can be surmised that an extensive onsite green infrastructure implementation could result in more frequent local flooding, e.g., water intrusion into basements. This may be especially the case when evaluating
scenarios for locations where medium–size storms have a long duration, like during the wet season of the Pacific Northwest of the United States. The comparatively high runoff estimated for Option 5 (Fig. 8d through f) would be maintained until all PA is saturated by increased rainfall intensity. If a smaller portion of PA is modeled as BPA, while all the other conditions are kept the same, the BPA reaches the saturated condition under a smaller storm. Once the BPA is saturated the area hydrologically responds like IA. However, SPA (i.e., non–buffering pervious area) can still control rainfall within the area.
This analysis suggests that it is important to properly define the area of BPA especially when analyzing GI alternatives for onsite stormwater controls, as we surmised originally. Therefore, Option 5 is not suitable for modeling a GI scenario because it ignores the actual significance of variance in BPA. It is a common modeling practice in SWMM to treat all pervious area the same (as in Options 4 or 5 in Fig. 5), even though only the BPA can receive water from ICIA. As shown in Fig. 8, simulated runoff by Options 4 or 5 would presumably be inaccurate, especially for the <1 year small storms.

Figure 9 contains graphs comparing the results among the six options (Fig. 5), showing the relative difference for peak flow, average flow, and total runoff volume for each of the five other Options compared to Option 1, presumably the most accurate one, in terms of output uncertainty because the level of spatial lumping is the lowest.

**Figure 9. Comparison of the hypothetical HRE modeling results.**

The relative differences reported in Fig. 9 ('Variation from the result of 5-subs' in Y-axis) were estimated as $(MR_j^k - MR_{5subs}^k)/MR_{5subs}^k$, where $MR_j^k$ represents a modeling result from the $k^{th}$ synthetic storm with the $j^{th}$ discretization option and $5subs$ means the discretization with five subcatments (i.e., Option 6). Options 1, 2, and 3 types of multi–subcatchment discretization present similar hydrologic responses for all storm sizes. In comparison, both Options 4 and 5 result in significantly different hydrologic outcomes, particularly for smaller storms. Again, this is due to the unresolved spatial delineation of DCIA from TIA, and BPA from TPA, respectively. Whereas Option 6 is based on a single–subcatchment approach, but produces similar results to the multi–subcatchment discretization approaches under Options 1, 2 and 3, for all storm classes tested. The difference between Option 6 and Option 1, though worth noting, are marginal for the three important hydrologic characteristics (Fig. 9). This outcome of the hypothetical analysis supports our original rationale for the relevance of characterizing the BPA. Under Option 6, the four critical hydrologic components (i.e., DCIA, ICIA, BPA, and SPA) are distinctly modeled in SWMM within a single–subcatchment that is delineated based on the actual drainage area to a storm sewer inlet (termed an HRE in this study). Based on the results, Option 6 balances the combination of discretization criteria, especially in terms of the level of effort required in model set–up, configuring parameter values and output uncertainty.

## 3.3 SHC watershed-scale modeling results

Option 6 (Fig. 5) was used to set up the SWMM model for the SHC watershed as described above. The SHC model consists of 191 subcatchments and 269 junctions and conduits (Fig. 6). The model also includes two wet ponds, two dry ponds, and a 10 yr detention area modeled as storage structures with orifice–type hydrologic controls. The results of the model sensitivity analysis were summarized for the period 22 to 24 July 2009, using the total runoff volume as the endpoint being assessed with Eq. (2) (Fig. 10). There was a total of 164.6 mm rainfall during the three days of this period; this storm is smaller than the 1 yr return period design storm (61.0 mm/d) but larger than the 6 month storm (48.3 mm/d) based on the storm statistics for the study area (see Table 2). While 3% change in total runoff is not significant in sensitivity, the most sensitive parameter was $K_{sat}$, followed by BPA and $DS$. Whereas the changes in $K_{sat}$ affect the entire PA (75.4 % of SHC), the changes in BPA affect a much smaller area (2.3 % of SHC for the baseline condition) than PA. The other parameters (i.e., width, slope, and n) were found not to be as sensitive, with negligible changes in results $\leq \pm 0.15$ % even for $\pm 20$ % change in the individual parameter value. When land cover status is represented accurately in a SWMM model, certain parameters will be less sensitive because of the underlying hydraulic and spatial realities are well represented. For example, the parameters representing the impervious land cover types in this modeling analysis were found to be less sensitive than pervious area parameters.

**Figure 10. Sensitivity analysis of the SWMM parameters at SHC.**

Model calibration was conducted by adjusting the land cover–based modeling parameters and BPA to the entire study watershed. As shown in Table 1, parameters for the impervious land cover types changed little and were made equivalent for $n$ and $DS$. As expected, parameters for the pervious land cover types needed more adjustment than those for the impervious. The initial value of $K_{sat}$ was defined using the site–specific soil types (mainly silty loam clay), but the values for the individual pervious land cover types were varied by the model calibration effort. Whereas $K_{sat}$ for forest area was adjusted only slightly (i.e., 1.6 initially to 1.52 for the final calibration), the values for lawn (or landscaped area) and agriculture required more adjustment (from 1.6 initial to 1.02 for agriculture, and from 1.6 initial to 0.89 for lawns). The relatively large changes for $K_{sat}$ are indicative of more soil compaction for urban and agricultural soils compared to the expected native soil condition.

The measured rainfall intensities and stream flow rates, along with the calibrated model results are presented in Fig. 11. The modeled hydrographs are well matched with the measured data at the watershed scale with a Nash–Sutcliffe coefficient = 0.852 and $R^2 = 0.871$.

**Figure 11. Watershed-scale SWMM modeling results from 1 July 2009 to 31 August 2009.**

After making the model adjustments for the GI scenario, the relative percentages of the four classified subareas changed (Fig. 12a). Using the hydrograph separation approach, the relative contribution of the primary hydrologic components with and without GI implementation were estimated for the period 1 July 2009 to 31 August 2009 (Fig. 12b). A more detailed representation for the hydrograph separation is presented in Fig. 13, which covers 72 hours from 22 to 24 July 2009. It is interesting to note from Fig. 13 that the peak flow for the event depicted in the figure is slightly higher in the GI scenario, but that the duration of flows slightly smaller than this peak is longer in the baseline scenario.

**Figure 12. Relative percentages of land cover and hydrologic components computed for the period 1 July 2009 to 31August 2009. In (b) "Others" represents surface runoff from areas other than DCIA, "Subsurface flow" is the subsurface contribution, and "Loss" is rainfall loss by evaporation or deep percolation.**

**Figure 13. Hydrograph separation and volumetric percentages contributing to stream flow for the period 22 to 24 July 2009.**

While the results from applying the hydrograph separation cannot be validated without extensive field measurements, the exercise provides insight to the potential effectiveness and rationale for developing strategies for GI in the watershed. For instance, about 48 % of the volumetric stream flow was contributed through subsurface flow over the simulation period, even though the study watershed is characterized with poorly infiltrating soils. After applying the GI scenario, although the subsurface flow contributed a similar fraction to the stream flow, the fractional contributions of surface runoff from DCIA and the other areas are significantly changed (Fig. 12b and 13). This situation arises not from a change in land cover but the internal

flow paths taken by the runoff. The result is reduced runoff from DCIA but increased runoff from the other areas (i.e., ICIA, BPA, and SPA).

From a water quality management perspective, it is necessary to consider hydrologic and contaminant discharge processes with respect to their sources and transport pathways. For example, if the watershed has water quality issues related to nutrients, the management effort might pay more attention to the stormwater discharge from pervious areas that include fertilizer applications. If GI were designed to intercept runoff from DCIA in the watershed, an unintended consequence could result from increased runoff volume traveling through a pervious area with elevated standing stocks of soluble or erodible nutrients. In this case, it would be important to consider turf management practices.

Another example of how the hydrograph separation approach (Fig. 13) provides additional opportunities for interpreting hydrodynamics before and after applying the GI scenario is revealed by considering that disconnecting downspouts reduced the total runoff volume, but also resulted in a higher peak flow (note the 16:00 time point on 22 July 2009 in Fig. 13). This result is like the single storm analysis using Option 5 (Fig. 5). Overall the flow volume is reduced from the GI scenario. However, when the peak occurred around 15:30 (shown in Fig. 13) the capacity of the GI for controlling stormwater was already exceeded because of controlling runoff during the previous rainfall that occurred between 7:00 and 14:00. Under this saturated condition, even the direct rainfall to the GI area will be discharged with minimum abatement. If there is no GI (as in the baseline condition), the same area receives only direct rainfall, there is no additional run on from impervious area, and that rainfall is controlled by still available surface depression storage and not–saturated infiltration capacity. In the 22 July 2009 situation, the stormwater control capacity (mainly *DS* and infiltration) of the extended BPA is saturated by earlier rainfall. Once saturated the BPA discharges higher runoff. The modeled GI contributes much higher runoff volume from PA, which might be nutrient enriched. With the hydrograph separation analysis, we gain insight to the consideration of stormwater management objectives and extend the utility of SWMM.

## 4. Summary and conclusions

We demonstrate how high resolution spatial data can be applied to spatially discretize a watershed and develop a methodology that should decrease model output uncertainty with reduced calibration effort. The suitability of the spatial discretization approach for GI modelling was initially verified with a hypothetical urbanscape analysis using eight synthetic storms of various sizes. We evaluated our approach to SWMM subcatchment parameterization using the hypothetical analysis that allowed for the qualification of five different options relative to one that would be considered the most spatially explicit, and, therefore, result in the least amount of output uncertainty (see Fig 9). From the hypothetical analysis, the best option was selected to develop a watershed-scale SWMM model at the study area. The simulated hydrographs by the developed watershed-scale SWMM model were well matched with observed data over a two month continuous simulation (Nash–Sutcliffe coefficient = 0.852; $R^2$ = 0.871) after minimal calibration effort. A GI scenario that modeled downspout disconnection from all the main buildings that are DCIA was described. We demonstrate how simple model adjustments can be made to separate the total and

surface runoff among primary pathways that runoff takes before discharging to the natural stream network. This hydrograph separation procedure can shed light on GI design requirements and water quality management.

The optimal spatial discretization scheme distinguishes DCIA from ICIA, and BPA from SPA, and explicitly models these as subareas within each subcatchment parameterization in SWMM. This approach is particularly useful when modeling the

5     impact of small storms, i.e., when BPA can control all or most of ICIA runoff. The land cover based spatial discretization approach is scale–independent, can be applied directly to a larger watershed as long as any heterogeneity in landscape properties is accounted for in the GIS set–up (e.g., by dividing land cover components into multiple sub–groups such as flat and slanted rooftops, high and low sloped urban hillslopes, or B and C type hydrologic soil types), and affords the opportunity to evaluate urban stormwater management strategies with presumably decreased output uncertainty for small storms and

10    expanded applicability to GI planning, design, and implementation. Parameters are adjusted per SWMM subcatchment in a typical calibration approach, which is scale–dependent and requires more effort in larger watersheds. In our approach, a SWMM model is calibrated by adjusting parameters per land cover component, which are categorized by urban development codes or general construction specifications for land uses. Overall this study demonstrates the relative effectiveness of different approaches in drainage area characterization using highly resolved spatial data to the set–up and analysis of a SWMM model

15    that should improve its utility for simulation of GI.

**List of Abbreviations**

| | |
|---|---|
| $A_1$, $A_2$, and $A_3$ | Empirically derived coefficients |
| ASCE | American Society of Civil Engineers |
| 20    $B_1$ and $B_2$ | Empirically derived coefficients |
| Bldg | Building |
| BPA | Buffering pervious area that receives and controls runoff from impervious area |
| CHI | Computational Hydraulics Int. |
| DCIA | Directly connected impervious area |
| 25    $DS$ | Depression storage |
| $DS\_imp$ | Depression storage for impervious area |
| EFW | East Fork (of the Little Miami River) watershed |
| ESRI | Environmental Systems Research Institute |
| GI | Green infrastructure |
| 30    GIS | Geographic Information System |
| $H^*$ | Threshold groundwater height |
| $H_{gw}$ | Height of saturated zone above the bottom of aquifer |

| | | |
|---|---|---|
| | $H_{sw}$ | Height of surface water above the bottom of the aquifer |
| | IA | Impervious area |
| | ICIA | Indirectly connected impervious area |
| | $IMD$ | Initial (soil) moisture deficit |
| 5 | $K_{sat}$ | Saturated hydraulic conductivity |
| | LID | Low impact development |
| | LiDAR | Light Detection and Ranging |
| | $MR$ | Modeling result |
| | $\Delta MR$ | Change in modeling result based on change in parameter value |
| 10 | $n$ | Manning's roughness coefficient |
| | NOAA | National Oceanic and Atmospheric Administration |
| | NRC | National Research Council |
| | NRCS | Natural Resources Conservation Service |
| | NSE | Nash–Sutcliffe coefficient |
| 15 | $p$ | Parameter value |
| | $\Delta p$ | Change in parameter value |
| | PA | Pervious area |
| | $Q_{DCIA}$ | Runoff from DCIA only |
| | $Q_{gw}$ | Groundwater Flow Rate |
| 20 | $Q_{ICIA+BPA}$ | Runoff from ICIA and BPA |
| | $Q_{imperv}$ | Runoff from impervious area (DCIA and ICIA through BPA) |
| | $Q_{subsurface}$ | Runoff through groundwater flow (i.e., subsurface or lateral flow) |
| | $Q_{SPA}$ | Runoff from SPA only |
| | $Q_{surface}$ | Surface runoff without groundwater flow |
| 25 | $Q_{total}$ | Total runoff with groundwater flow in SWMM |
| | $R^2$ | Coefficient of determination |
| | SHC | Shayler Crossing Watershed |
| | $Suct$ | Capillary Suction Head |
| | SPA | Standalone pervious area that does not receive or control any impervious area runoff |
| 30 | SWMM | Storm Water Management Model |
| | TPA | Total pervious area |
| | TIA | Total impervious area |
| | Trpt | Transport |
| | USDA | United States Department of Agriculture |

| USEPA | United States Environmental Protection Agency |
| WEF | Water Environment Federation |

**Disclaimer**

The U.S. Environmental Protection Agency, through its Office of Research and Development, funded, managed, and collaborated in the research described herein. It has been subjected to the Agency's administrative review and has been approved for external publication. Any opinions expressed in this paper are those of the author(s) and do not necessarily reflect the views of the Agency, therefore, no official endorsement should be inferred. Any mention of trade names or commercial products does not constitute endorsement or recommendation for use.

**Acknowledgments**

The authors would like to thank Mr. Bill Mellman of Clermont County; Mr. Paul Weaver of APTIM; and Dr. Michael Tryby, Dr. Michael Elovitz, and Dr. William Shuster of USEPA. They provided critical data, suggestions, or critical reviews.

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

**Appendix. Miscellaneous procedures for SWMM modeling for GI analysis**

For a full description of the steps used to set-up a SWMM model for GI analysis using the methods outlined here see Lee et al 2017 that is available for free download from the USEPA at: https://nepis.epa.gov/Exe/ZyPDF.cgi/P100TJ39.PDF?Dockey=P100TJ39.PDF. The procedure for developing the baseline model is schematically diagrammed in Fig. A-1. To reduce model complexity, the original 16 land cover types (mentioned in Sect. 2.2.2) were reduced to 10 by merging the paved walking paths, patios and miscellaneous impervious areas with other impervious areas, the dry ponds were merged with lawn areas, the detention area was merged with forest, and the surface areas for wet ponds and pools were modeled as IA without *DS*. The structures of the dry ponds, detention areas, and wet ponds were modeled as SWMM storage units. The final 10 land cover classifications include: main buildings, miscellaneous buildings, streets, driveways, parking, sidewalks, other impervious areas, lawn, forest, and agriculture.

**A.1 Determining values for SWMM subcatchment parameters: Area-weighting approach**

Unique values for representing the corresponding area, width, slope, imperviousness, *n*, *DS*, and infiltration parameters ($K_{sat}$, *Suct*, and *IMD* for the Green–Ampt) were defined per SWMM subcatchment using GIS and a spreadsheet.

**Figure A-1. Procedures for SWMM modeling for GI analysis. LC stands for land cover. Conceptual workflow for each of the colored boxes is given in the middle of the diagram. Symbols ( ◐ ◉ ◆ □ ◇ ) are used to label workflow connections within the colored boxes. Top panel depicts the general steps used for baseline model set–up, while the bottom panel adds GI considerations. "Acceptable?" means whether the statistical significance between the modeled and the observed hydrographs (e.g., Nash–Sutcliffe coefficient or $R^2$) is 'acceptable'. "Satisfied?" means whether the modeling results are 'satisfied' with the GI requirements.**

The area of each land cover type within a subcatchment was estimated using ArcGIS. The SWMM parameter 'characteristic width' was estimated using an area–weighted flow length, as recommended in the SWMM Applications Manual (Gironás et al., 2009). This manual suggests "If the overland flow length varies greatly within the subcatchment, then an area–weighted average should be used." Comparatively, in conventional SWMM modeling, the 'characteristic width' is computed by dividing the subcatchment area by the average maximum overland flow length. Then adjustments are made to the characteristic width during model calibration to produce the best fit to the measured runoff hydrographs. The following area–weighting calculation describes how the characteristic width for a SWMM subcatchment was estimated in this study, where *i* represents all the individual land cover types within the subcatchment:

$$Length = \sum(Length_i \, Area_i) \, / \sum(Area_i) \tag{A-1}$$

$$Width = \sum(Area_i) \, / \, Length \tag{A-2}$$

Other parameters were also defined as area–weighted averages per subcatchment (e.g., slope and infiltration parameters) or subarea (e.g., *n* and *DS*). This area–weighting step is used following the typical approach recommended to account for the spatial lumping that effectively averages patchy land cover types within SWMM subcatchments (Gironás et al., 2009; Rossman and Huber, 2016). Hence, the extent of each individual type of land cover in a subcatchment is used to area–weight the assigned

parameter values. For example, if IA within a subcatchment consists of two building rooftops, two driveways, and one section of street, the associated IA in SWMM is assigned values for *DS* based on an area–weighted average using the corresponding nominal values presented in Table 1, $[DS\_imp = \sum(DS_i A_i) / \sum(A_i)]$, where *DS_imp* is the assigned *DS* for the impervious subarea within the subcatchment, *A* is the size of an individual impervious land cover type within the subcatchment, and *i* is an individual land cover type. Calculations for area–weighting were all done in a Microsoft Excel spreadsheet that was configured for direct copy and pasting as the SWMM input file using the EPA–SWMM Excel Editor function (See Lee et al., 2017 for specifics and Fig. A-1). Two sets of *n* and *DS* were defined per SWMM subcatchment – one set for the impervious subarea and the other for the pervious subarea. Where available, relevant values were obtained from experience in the watershed or using GIS (e.g., overland flow length), or as suggested by the SWMM manual (Huber and Dickinson, 1988; Rossman, 2015). The SHC watershed has silty loamy clay soils throughout. Based on this soil type, *Suct* was set to be 165 mm using the SWMM User's Manual (Rossman, 2015). *IMD* was modeled using the default value (0.22) in the EPA–SWMM. The actual *IMD* is dynamically updated at every modeling time step. The developed SWMM model runs for a six–month period where the first four months of this period are used to stabilize the continuous simulation (i.e., the warm–up period). Hence, the *IMD* value when modeling results are first reported may not reflect the initial value assigned during model setup. The values for $K_{sat}$ were amended during the model calibration to comprise surface compaction in lawn, agricultural, and forest areas (Horton et al., 1994; Gregory et al., 2006). However, we did not try to change the values of *Suct* and *IMD* during the calibration. Although the status and spatial extent of each land cover type within each HRE are different, the parameter values were assigned independent of the HRE in which it resided (Table 1). This parameter assignment methodology at the level of land cover components reduces model complexity by minimizing the amount of subcatchment specific parameterizations that may need to be considered during calibration.

**A.2 Setting up the BPA**

The baseline BPA (that controls runoff from ICIA) was modeled by parameterizing the subcatchment LID controls of SWMM. The LID process 'vegetated swale' was selected among the LID control options in SWMM as the most appropriate option to represent the actual BPA. The BPA area estimated from the geoprocessing steps described above was added as well as values for the width, initial saturation, and % of subcatchment imperviousness draining to the BPA. The width was set to 18.3 m (60 feet), was equal across all subcatchments, and was based on the average linear footage of BPA around the existing ICIA from distance measurements made using the GIS on a number of common ICIA features in the watershed, e.g., driveways, sidewalks, and miscellaneous outbuildings. Individual BPAs within a subcatchment are assumed to be parallelly aggregated in setting-up the vegetated swale. The initial saturation was also equal across all subcatchments; set at 25 % (this value self–equilibrates after the model warm–up period, see Sect. 2.6). The berm height of the vegetated swale was set at 2.54 mm (0.1 inch) to minimize any storage effect within the berm, which is the case for real BPA, and vegetation volume fraction was set to be 0. The percentage of subcatchment imperviousness contributing to the BPA (i.e., the ICIA) is obtained by dividing the ICIA by the total IA. Since the total pervious area (TPA) remains identical for each HRE, the sizes of SPA for individual HREs can be

determined as SPA = TPA – BPA for the three different sizes of BPA, which were derived by applying three different distances for the proximity analysis in GIS. When we calibrated the model, we checked which one, among the three cases of BPA sizes established above would calibrate the best for various storm sizes.

**A.3 The groundwater component**

In SWMM groundwater flow is estimated by the following equation (Rossman, 2015):

$$Q_{gw} = A_1(H_{gw} - H^*)^{B_1} - A_2(H_{sw} - H^*)^{B_2} + A_3 H_{gw} H_{sw} \tag{A-3}$$

Where, $Q_{gw}$ = groundwater flow rate [L$^3$T$^{-1}$]; $H_{gw}$ = height of saturated zone above the bottom of aquifer [L]; $H_{sw}$ = height of surface water above the bottom of the aquifer [L]; $H^*$ = threshold groundwater height [L]; and $A_1$, $A_2$, $A_3$, $B_1$, and $B_2$ = empirically derived coefficients.

The top of the saturated zone is placed somewhere between the soil surface and the bottom of the aquifer. The $H^*$ is identical to the height of the streambed above the bottom of the aquifer (Rossman, 2015). No measurement data were available for relative elevations of the saturated zone or the bottom of the aquifer for the study area, but even with these values the groundwater parameterization in SWMM cannot be explicitly configured given the five coefficients that need specification (Eq. 1). Therefore, as is typical, we based the groundwater simulation on the elevation difference between individual

subcatchment surface and its nearest stream bottom, which affects $H_{gw}$. Groundwater modeling parameters were defined using the SWMM Reference Manual (Rossman and Huber, 2016) and SWMM users' group knowledge base (e.g., https://www.openswmm.org/Topic/1465/groundwater-parameters; https://www.openswmm.org/Topic/4840/groundwater-values). As part of simulating soil moisture content, evaporation is modeled by localized average daily rates for individual months obtained from an existing report (NOAA, 1982). The rates were taken directly based on the location of the study site

without adjustment.

**Table 1. Initial and calibrated modeling parameters for the Shayler Crossing watershed.**

| Land Cover | Length (m) | | Slope (%) | | $n$ | | DS (mm) | | $K_{sat}$ (mm/hr) | |
|---|---|---|---|---|---|---|---|---|---|---|
| | Initial | Calibrated | Initial | Calibrated | Initial | Calibrated | Initial | Calibrated | Initial | Calibrated |
| Main Building | 9.1 | 7.6 | 10 | 15 | 0.014 | 0.01 | 2.0 | 1.3 | n/a | n/a |
| Misc. Building | 4.6 | 4.6 | 10 | 15 | 0.014 | 0.01 | 2.0 | 1.3 | n/a | n/a |
| Street | 3.0 | 3.0 | 2 | 2.5 | 0.011 | 0.01 | 2.5 | 1.3 | n/a | n/a |
| Driveway | 4.6 | 3.7 | 2 | 1.5 | 0.012 | 0.01 | 2.5 | 1.3 | n/a | n/a |
| Parking | 3.0 | 3.0 | 1 | 1.5 | 0.012 | 0.01 | 3.0 | 1.3 | n/a | n/a |
| Sidewalk | 0.9 | 0.9 | 1 | 1.5 | 0.012 | 0.01 | 3.0 | 1.3 | n/a | n/a |
| Other Impervious | 3.0 | 2.4 | 1 | 1.5 | 0.012 | 0.01 | 3.0 | 1.3 | n/a | n/a |
| Lawn | 24.4 | 24.4 | 2 | 2 | 0.2 | 0.3 | 5.1 | 5.1 | 1.6 | 0.89 |
| Forest | 24.4 | 24.4 | 3 | 2 | 0.6 | 0.6 | 10.2 | 7.6 | 1.6 | 1.52 |
| Agriculture | 30.5 | 30.5 | 2 | 2 | 0.3 | 0.3 | 7.6 | 5.1 | 1.6 | 1.02 |

**Table 2. Profile of the selected eight 24 h single storm statistics.**

| Rain (mm) | Frequency | Percentile | Cumulative |
|:---:|:---:|:---:|:---:|
| 12.7 | < 1 month | 64.8 % | 32.7 % |
| 25.4 | 1–2 months | 87.4 % | 63.1 % |
| 36.8 | 3 months | 95.0 % | 80.7 % |
| 48.3 | 6 months | 97.7 % | 89.2 % |
| 61 | 1 year | 99.2 % | 95.3 % |
| 73.7 | 2 years | 99.6 % | 97.3 % |
| 108 | 10 years | 100 % | 99.8 % |
| 149.8 | 50 years | 100 % | 100 % |

Note: The Percentile and Cumulative, percentage-based statistics qualify the exceedance probability of each event and the relative contribution of events of similar size or lower to the annual rainfall, respectively.

**Table 3. Land cover status of Shayler Crossing watershed.**

| Surface Components | | DCIA (m$^2$) | ICIA (m$^2$) | Sum (m$^2$) | Fraction |
|---|---|---|---|---|---|
| Impervious areas | Building | 91770.0 | 3756.2 | 95526.2 | 9.6 % |
| | Street | 57610.5 | 14897.2 | 72507.7 | 7.3 % |
| | Driveway | 33554.7 | 2083.7 | 35638.4 | 3.6 % |
| | Parking | 2362.7 | 3154.1 | 5516.8 | 0.6 % |
| | Sidewalk | 1646.9 | 9990.3 | 11637.2 | 1.2 % |
| | Miscellaneous | – | 17766.8 | 17766.8 | 1.8 % |
| | Sum of IA | 186944.7 | 51648.4 | 238593.1 | 24.0 % |
| Pervious areas | Lawn | | | 400667.4 | 40.3 % |
| | Agriculture | | | 219430.4 | 22.1 % |
| | Forest | | | 128558.1 | 12.9 % |
| | Sum of PA | | | 748655.9 | 75.4 % |
| Water | Wet pond | | | 5014.2 | 0.5 % |
| | Swimming pool | | | 998.9 | 0.1 % |
| | Sum of Water | | | 6013.0 | 0.6 % |
| Sum | | | | 993262.0 | 100 % |

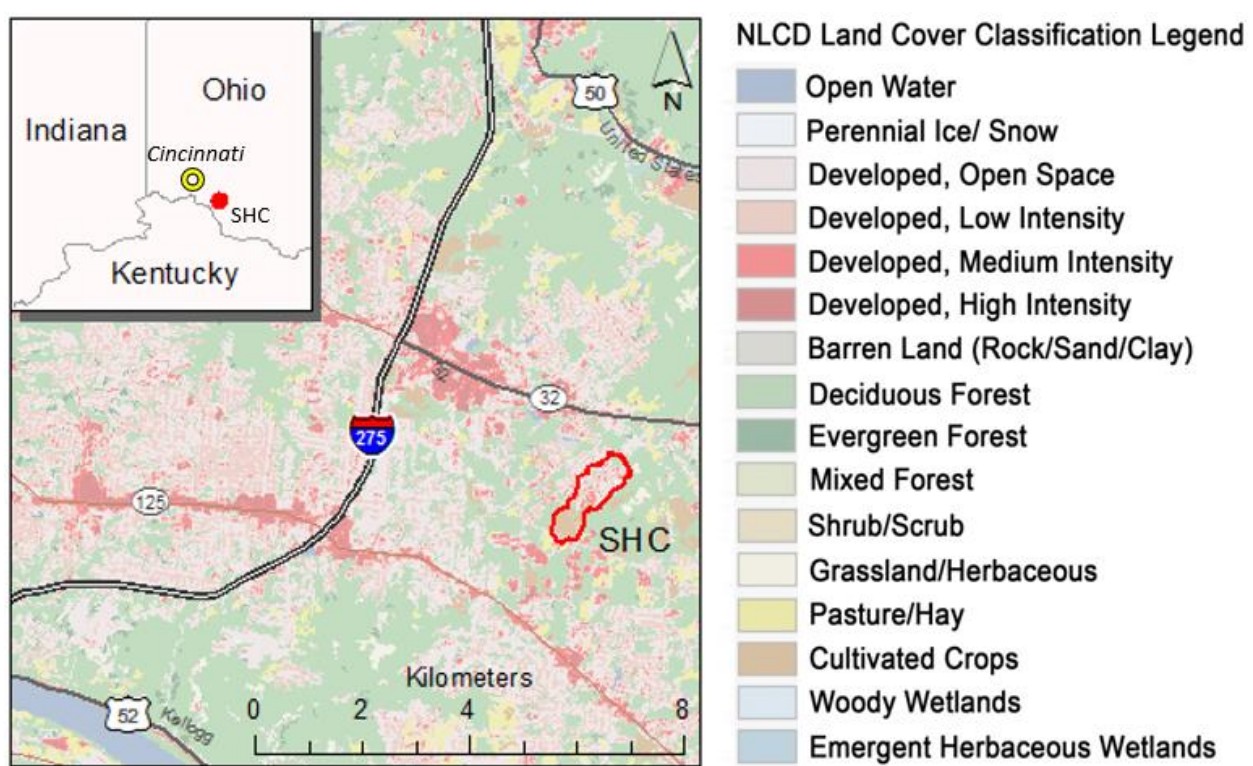

**Figure 1. Location of the Shayler Crossing watershed. I-275 is an interstate highway around the Cincinnati metropolitan area.**

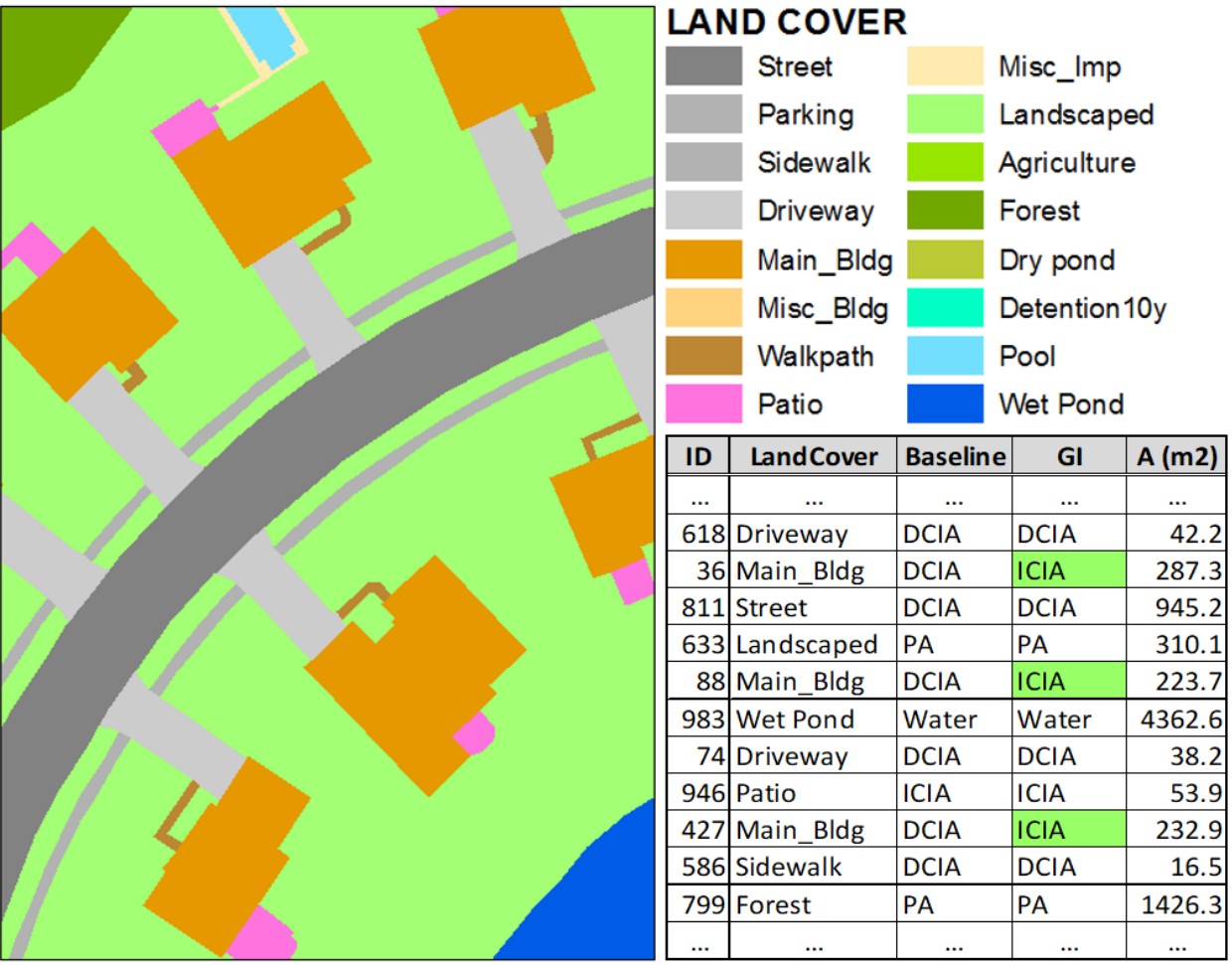

| ID | LandCover | Baseline | GI | A (m2) |
|---|---|---|---|---|
| ... | ... | ... | ... | ... |
| 618 | Driveway | DCIA | DCIA | 42.2 |
| 36 | Main_Bldg | DCIA | ICIA | 287.3 |
| 811 | Street | DCIA | DCIA | 945.2 |
| 633 | Landscaped | PA | PA | 310.1 |
| 88 | Main_Bldg | DCIA | ICIA | 223.7 |
| 983 | Wet Pond | Water | Water | 4362.6 |
| 74 | Driveway | DCIA | DCIA | 38.2 |
| 946 | Patio | ICIA | ICIA | 53.9 |
| 427 | Main_Bldg | DCIA | ICIA | 232.9 |
| 586 | Sidewalk | DCIA | DCIA | 16.5 |
| 799 | Forest | PA | PA | 1426.3 |
| ... | ... | ... | ... | ... |

(Note: The data table in this figure does not show the entire records of the database.)

**Figure 2. Sample GIS classified representation of the land cover and hydrologic characteristics.**

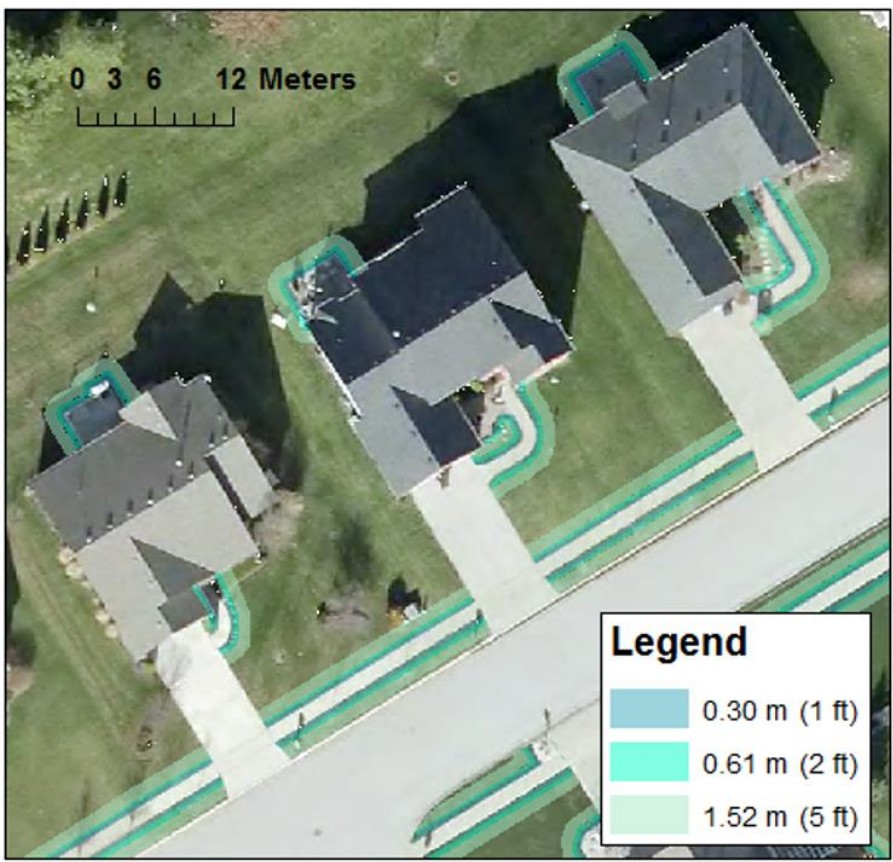

**Figure 3. Depiction of the different distances applied for the estimation of BPA in the baseline condition using ArcGIS.**

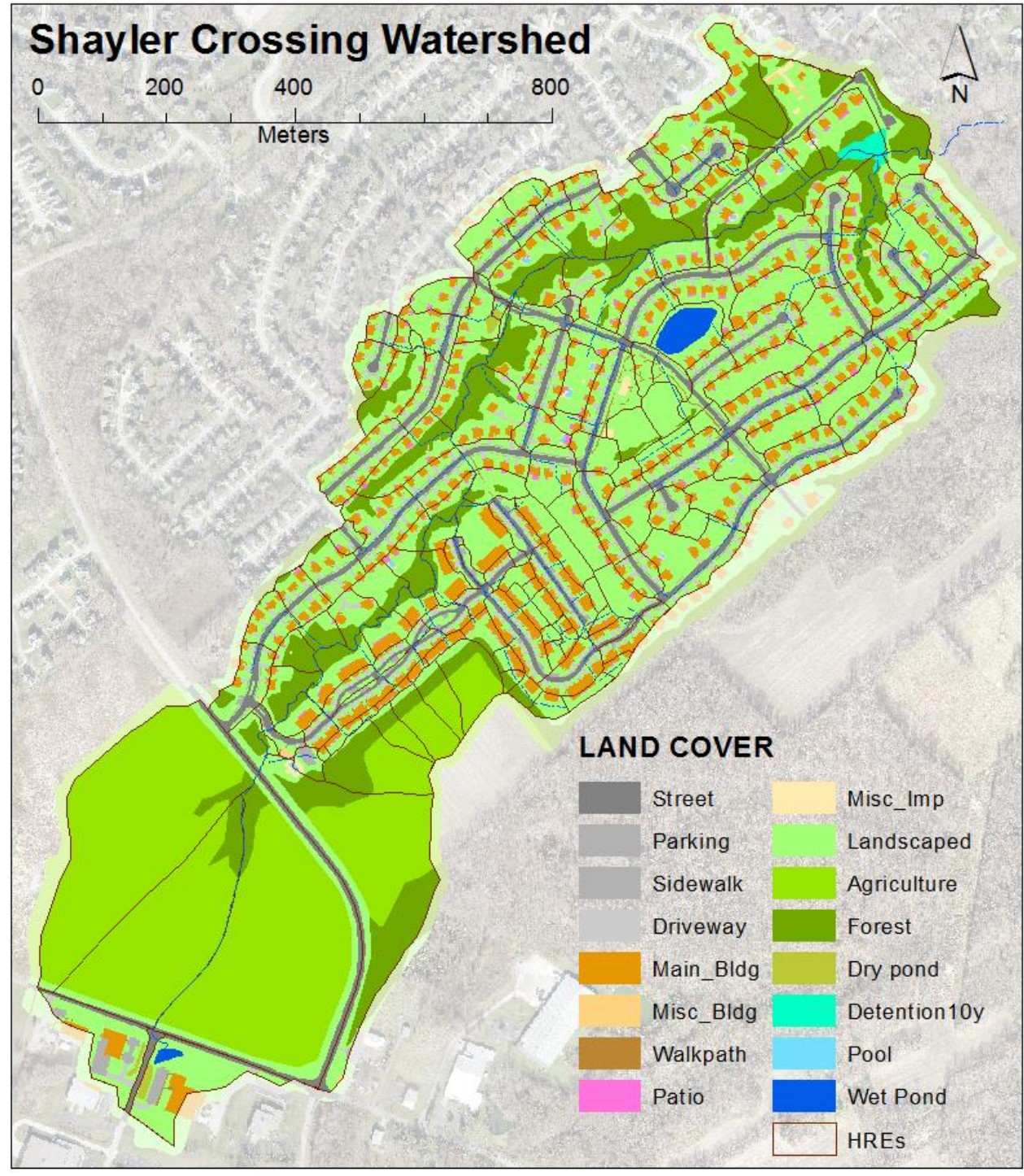

**Figure 4. Detailed spatial representation of the Shayler Crossing watershed.**

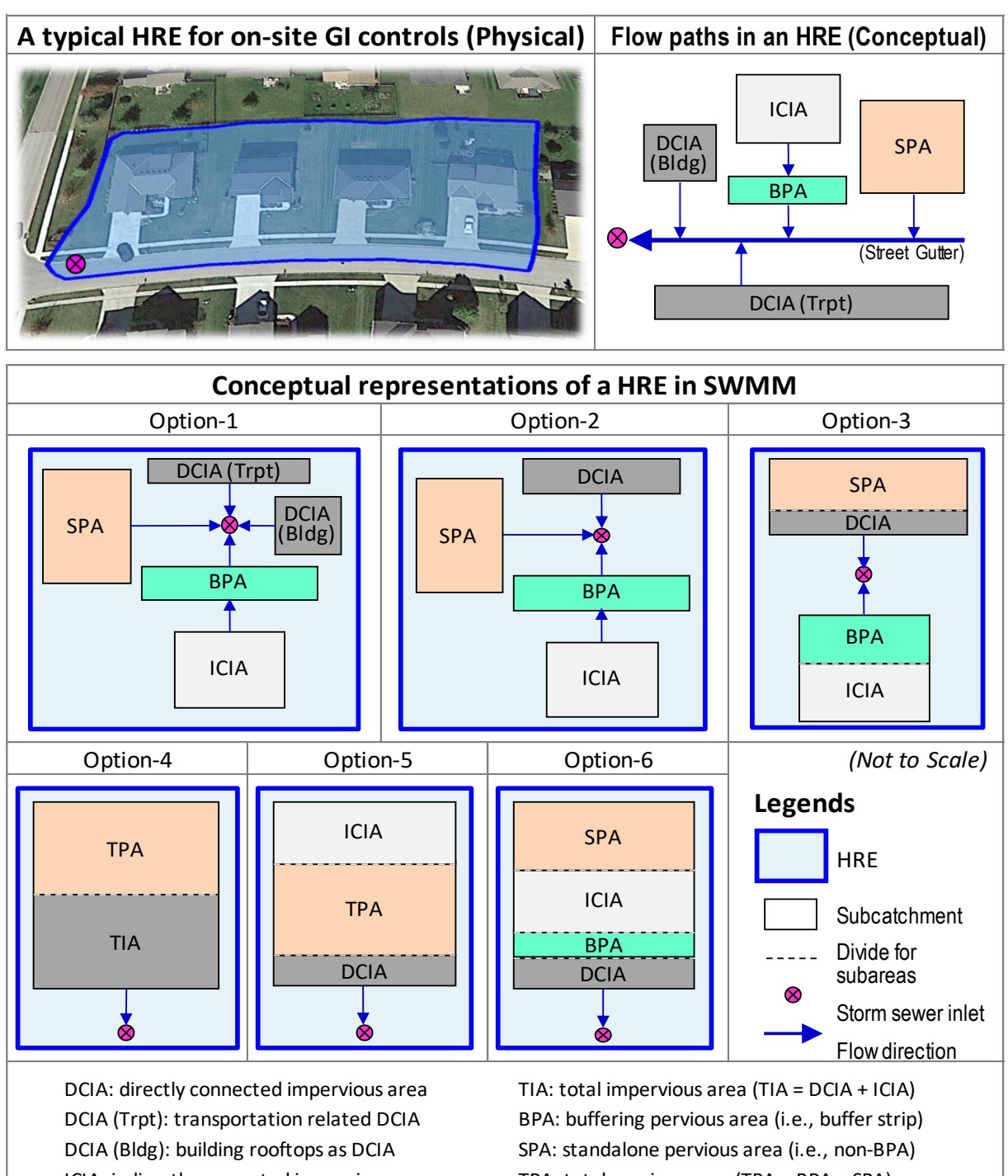

**Figure 5. A conceptual representation of the hypothetical HRE (20% DCIA, 30% ICIA, 10% BPA and 40% SPA) and the 6 options considered for representing this area in the set–up of a SWMM model.**

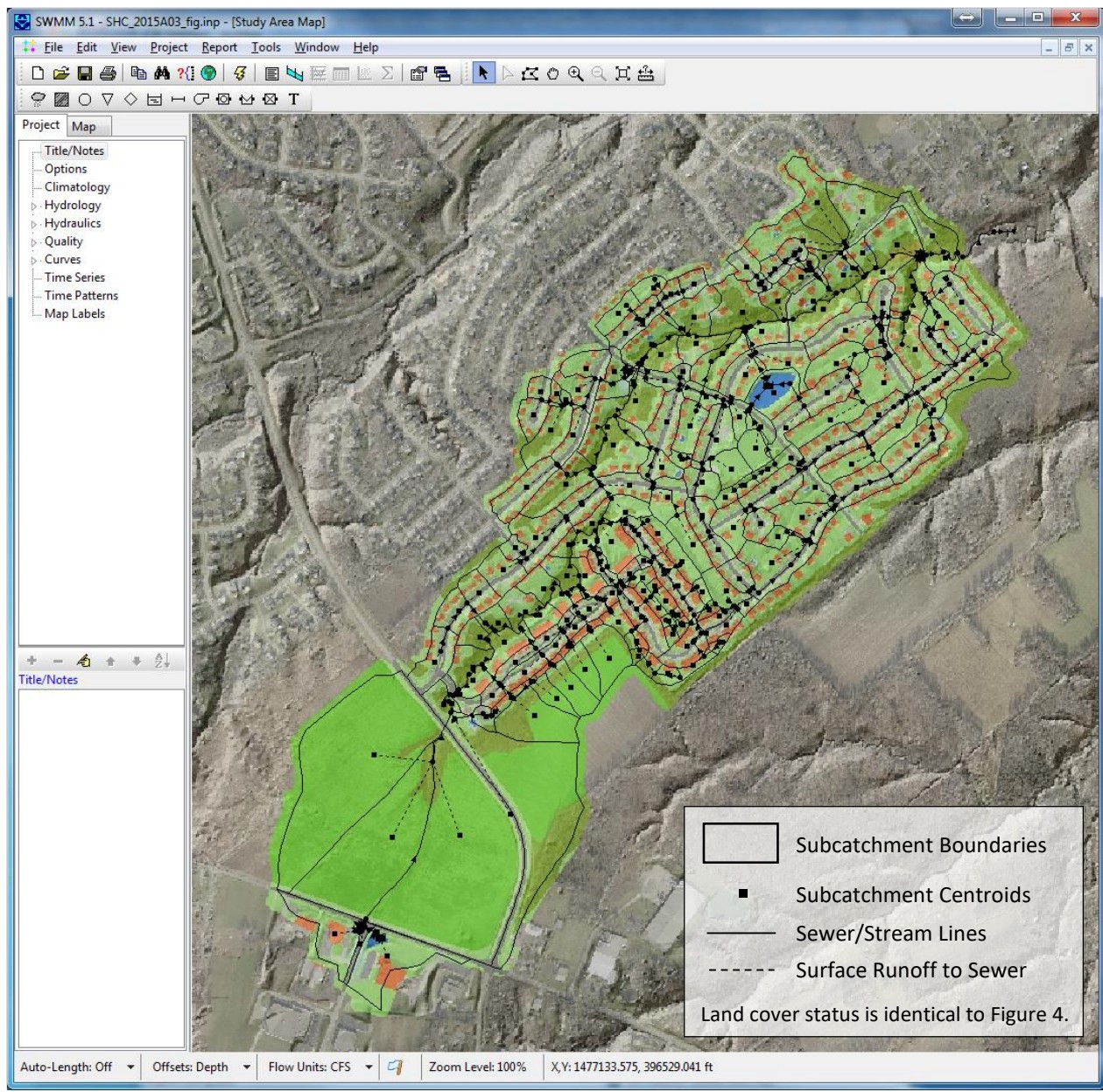

**Figure 6. Diagram of the developed SWMM model for the Shayler Crossing watershed.**

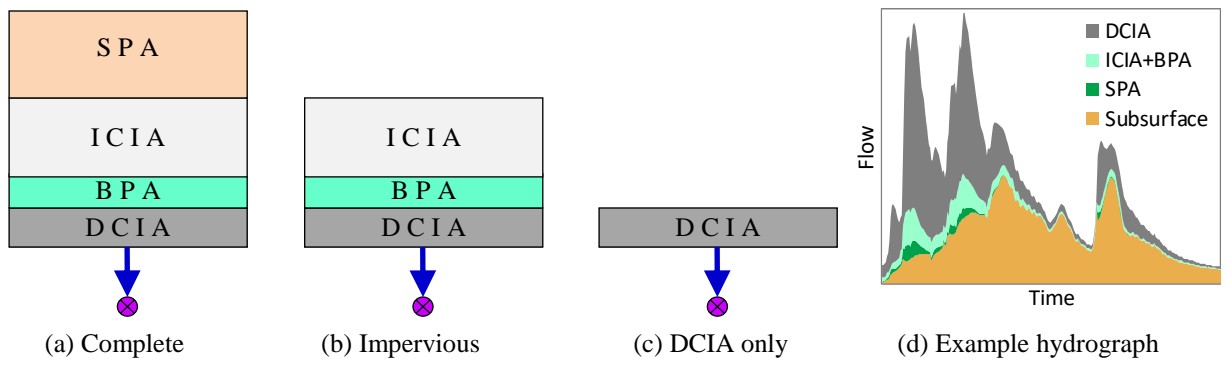

**Figure 7. Conceptual representations of discrete SWMM models for hydrograph separation.**

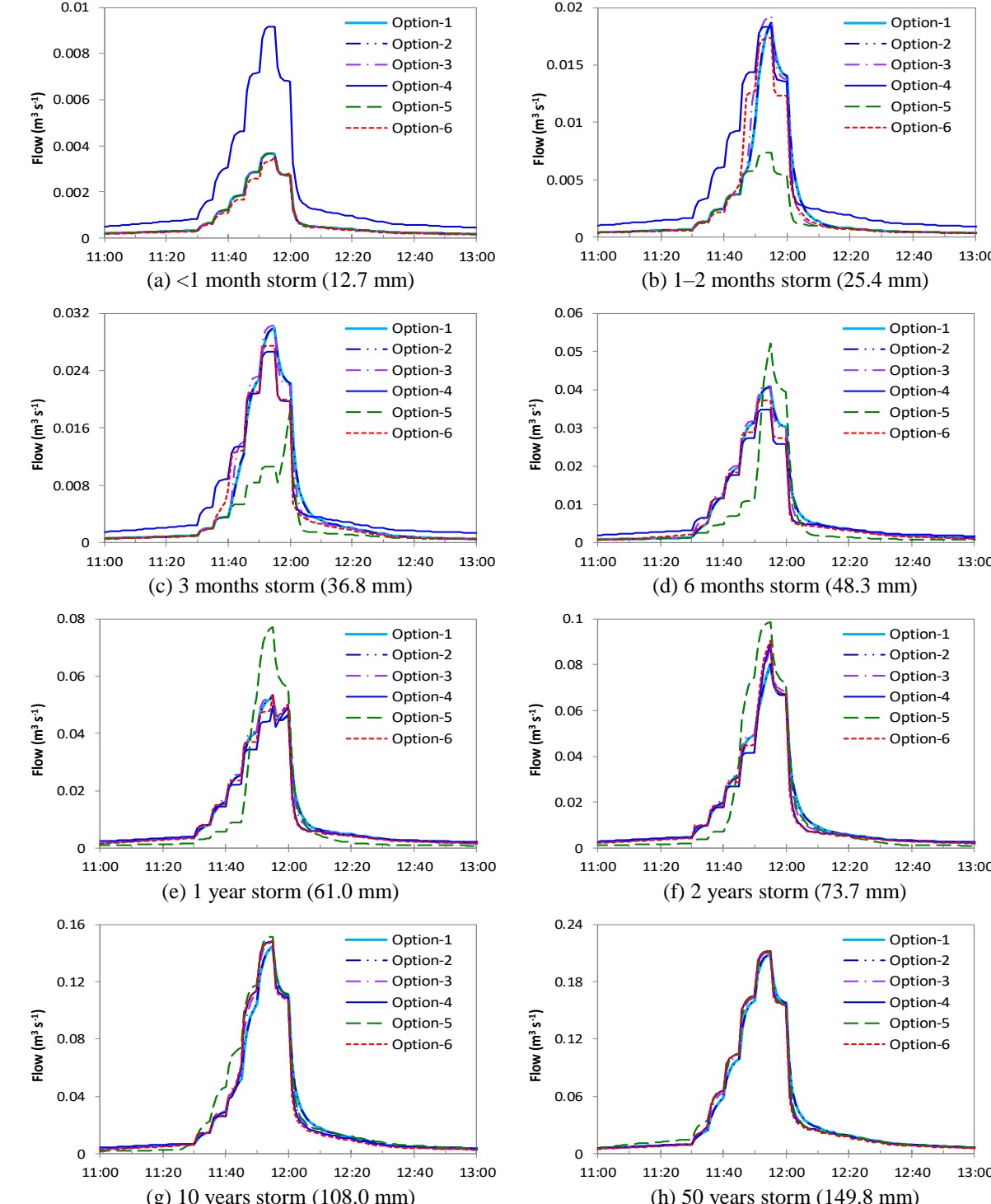

**Figure 8. Hypothetical HRE SWMM modeling results.**

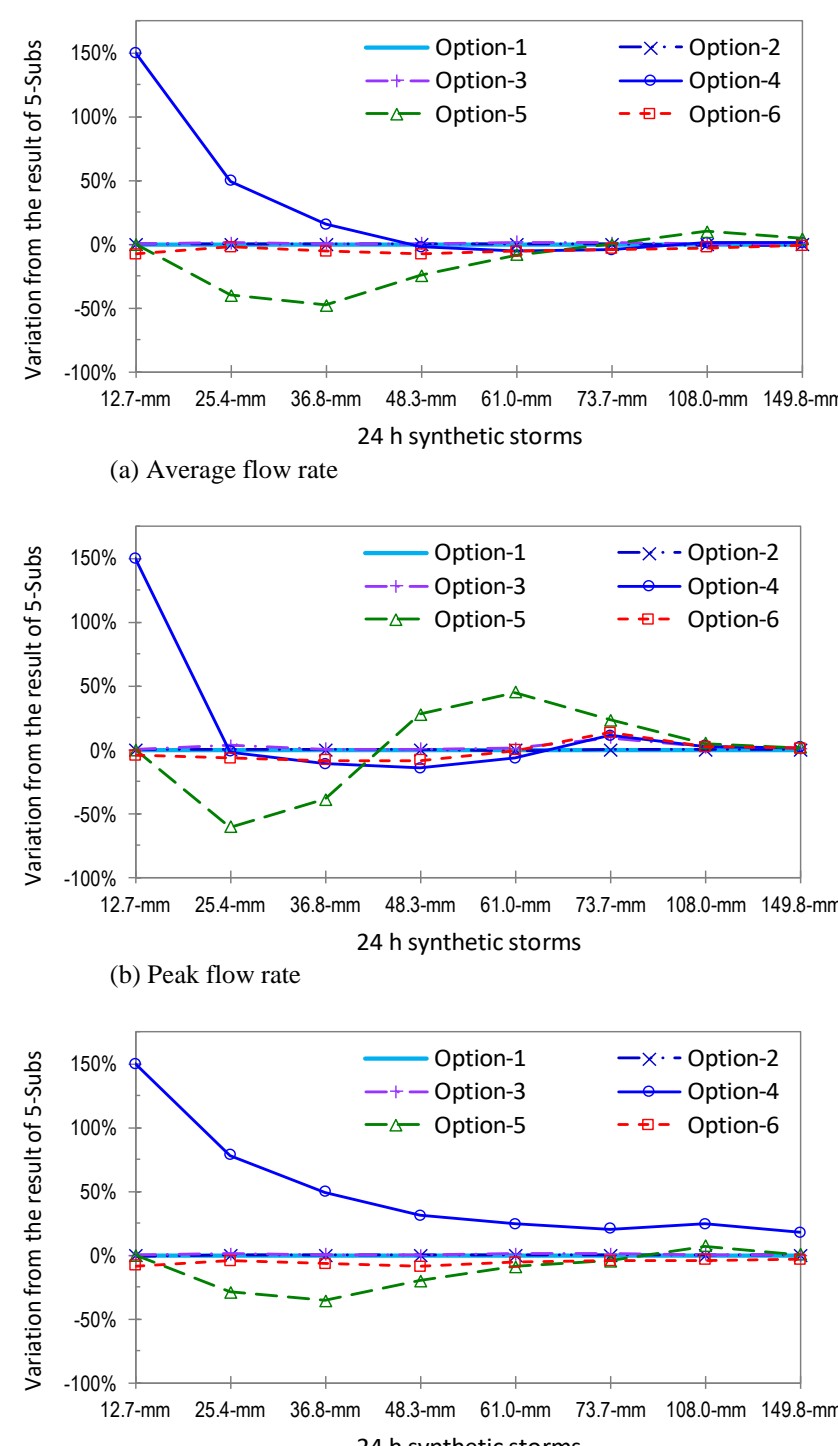

(a) Average flow rate

(b) Peak flow rate

(c) Total runoff volume

**Figure 9. Comparison of the hypothetical HRE modeling results.**

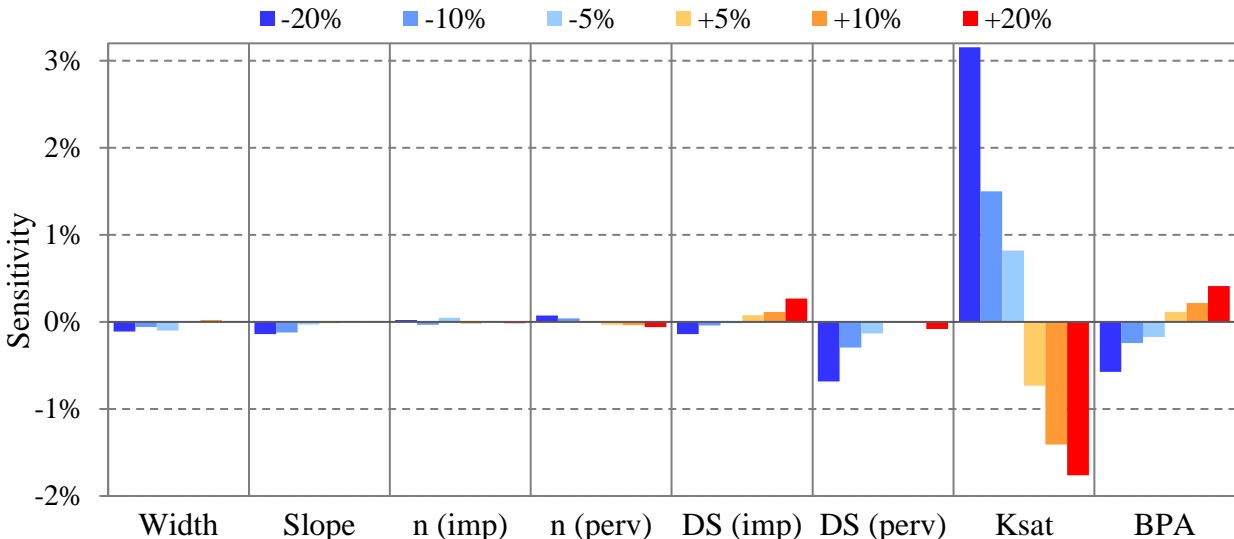

**Figure 10. Sensitivity analysis of the SWMM parameters at SHC.**

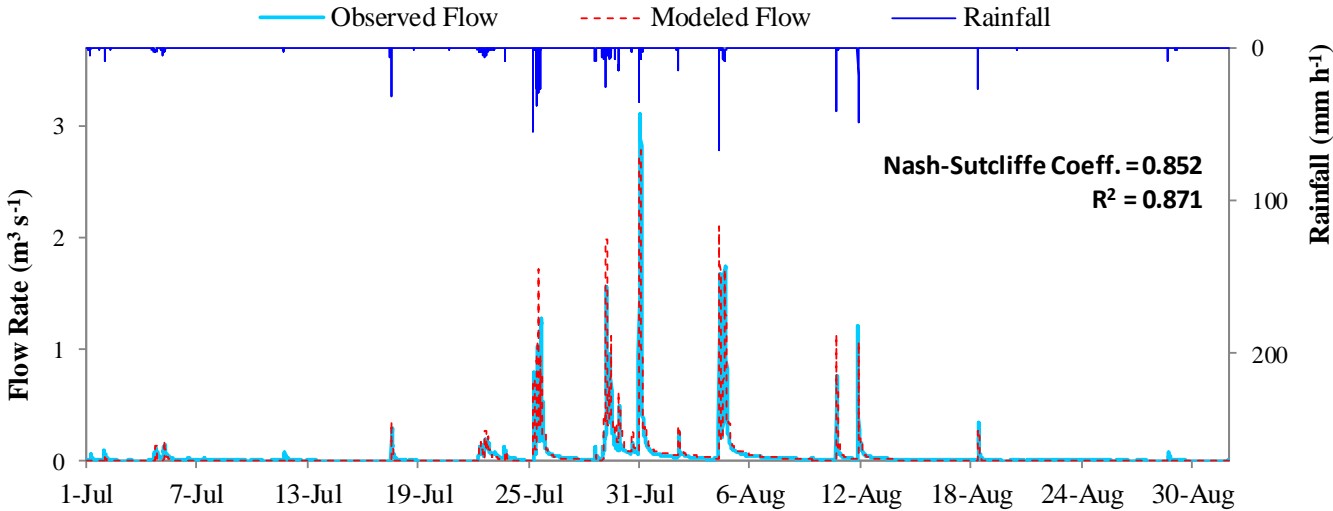

**Figure 11. Watershed-scale SWMM modeling results from 1 July 2009 to 31 August 2009.**

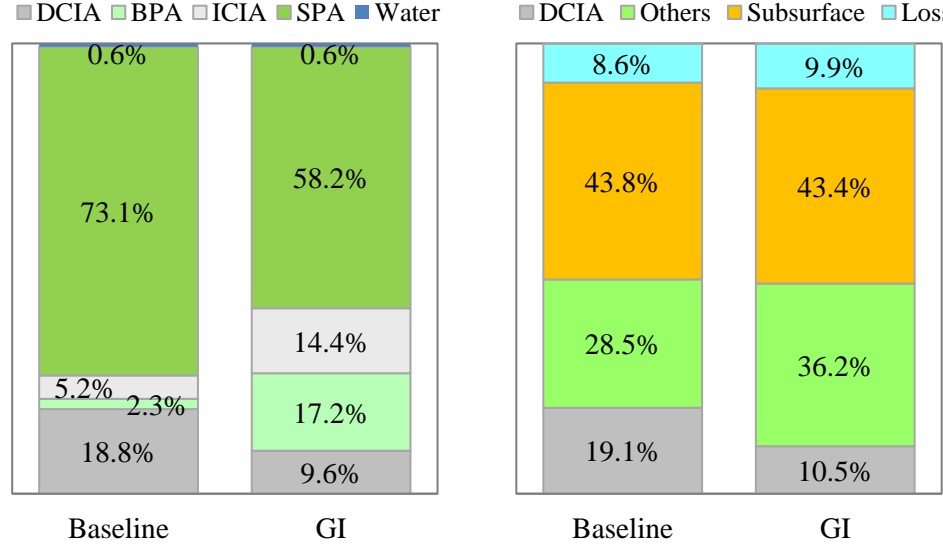

(a) Land cover components      (b) Hydrologic components

**Figure 12. Relative percentages of land cover and hydrologic components computed for the period 1 July 2009 to 31August 2009. In (b) "Others" represents surface runoff from areas other than DCIA, "Subsruface flow" is the subsurface contribution, and "Loss" is rainfall loss by evaporation or deep percolation.**

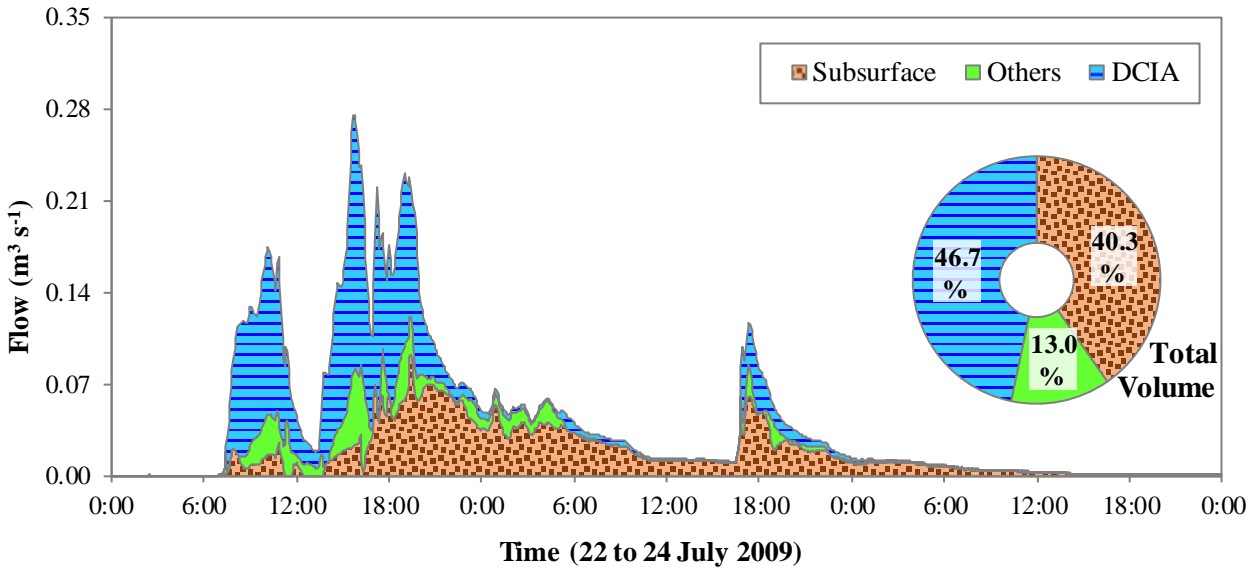

(a) Baseline condition

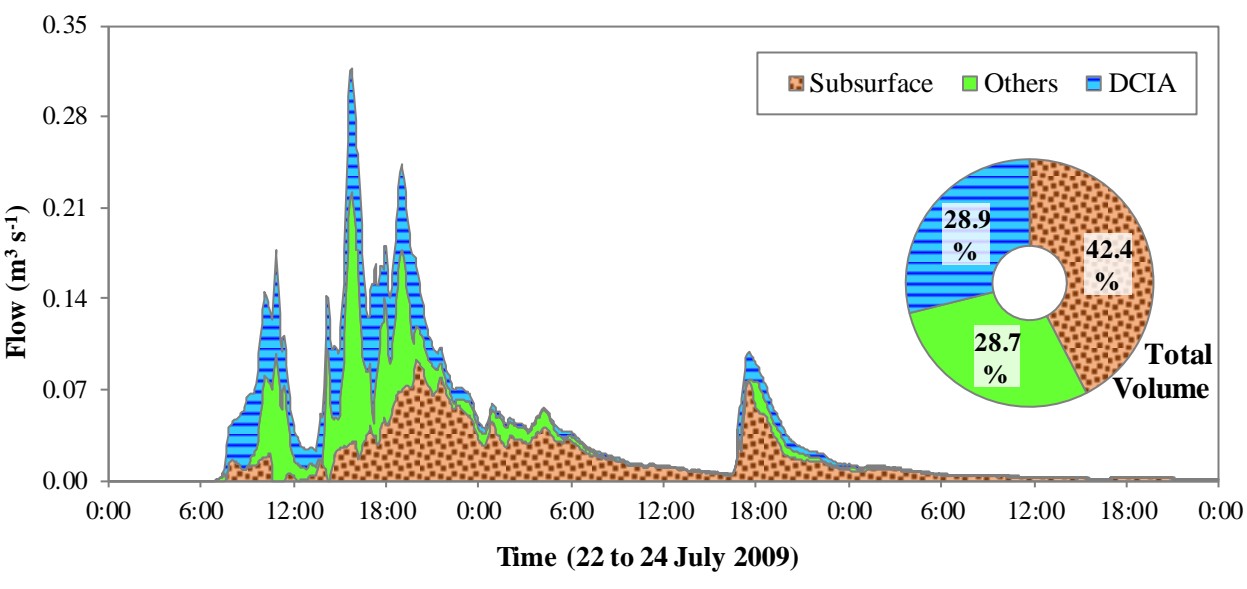

(b) GI implementation scenario

5    **Figure 13. Hydrograph separation and volumetric percentages contributing to stream flow for the period 22 to 24 July 2009.**

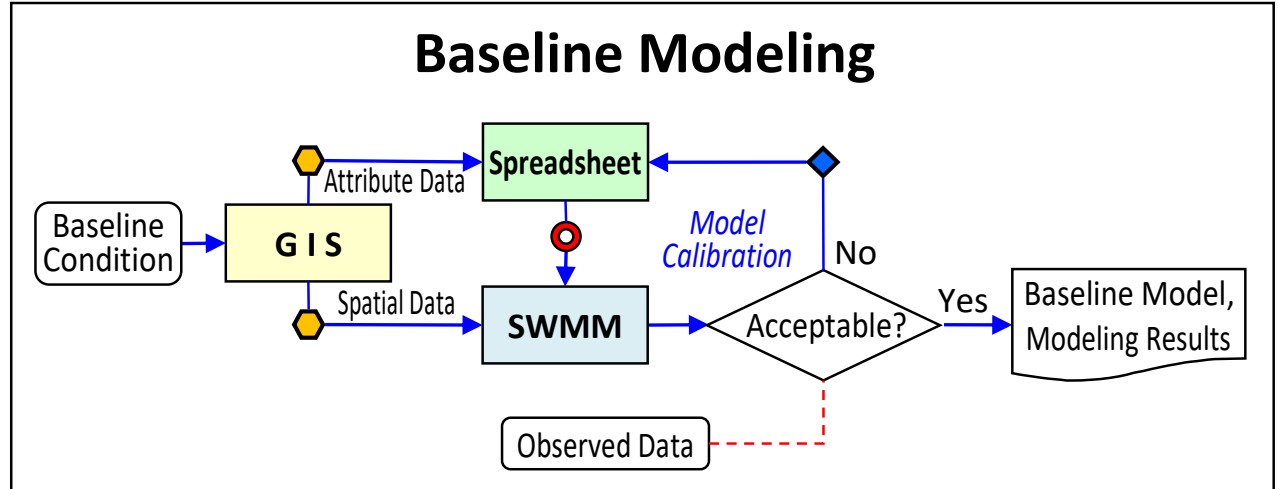

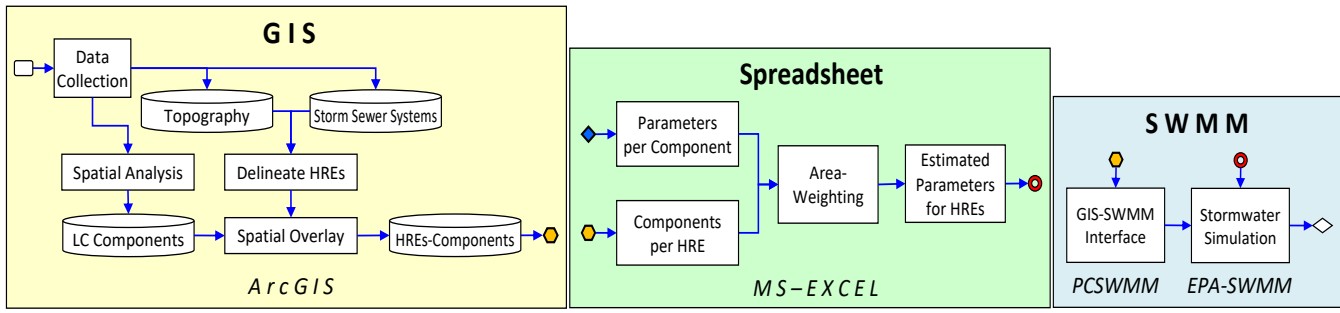

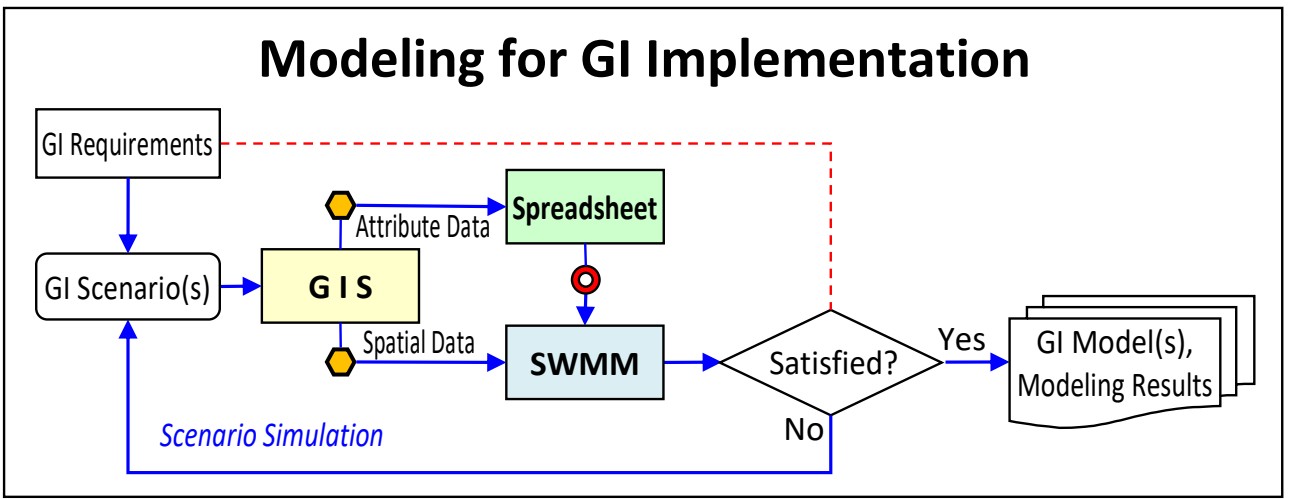

**Figure A-1. Procedures for SWMM modeling for GI analysis. LC stands for land cover. Conceptual workflow for each of the colored boxes is given in the middle of the diagram. Symbols ( ⬡ ◎ ◆ ▢ ◇ ) are used to label workflow connections within the colored boxes. Top panel depicts the general steps used for baseline model set–up, while the bottom panel adds GI considerations. "Acceptable?" means whether the statistical significance between the modeled and the observed hydrographs (e.g., Nash–Sutcliffe coefficient or $R^2$) is 'acceptable'. "Satisfied?" means whether the modeling results are 'satisfied' with the GI requirements.**