# Peer review of "Drainage area characterization for evaluating green infrastructure using the Storm Water Management Model"

_Hydrology and Earth System Sciences, 2017_

## Referee Comment (RC1) · Anonymous Referee #1 · 27 Jun 2017

The article by Lee et al tackles an interesting topic in the field of green infrastructure. The research approaches the paper investigated are meaningful for GI under smaller storm events. Some of the assumptions used in the paper need to be better explained and argued. The conclusions are not attended yet due to insufficient description of their methods, model settings, estimation of key parameters, in particular the part 2.2-2.4. Nonetheless, I think that the article had good potential for being published, provided that the following comments are adequately addressed.

General comments: 1. The research approach replies on a highly resolved spatial database of urban land cover, stormwater drainage feature and topography, what about

its potential application in a general context? Most of urban areas may not have such detailed dataset or require extensive surveying and modeling efforts. 2. Relevant references are needed to support statements in the text, see specific comments for details. The key definitions (e.g., DCIA, ICIA, SPA, BPA) are given, but a conceptual model characterizing these key processes in a watershed and their spatial connections should be provided. 3. The land cover characterization in GIS is an essential step to provide inputs for hydrological evaluation in SWMM. Very limited information is given to understand how it is done in GIS analysis. Also readers need more details on how the four types of subareas are subsequently modeled in SWMM (e.g. parameter settings), e.g., how to parameterize BPA, ICIA, SPA for subcatchments. 4. A better description of model calibration process is recommended, e.g., summary of parameters, inputs and outputs, criteria of performance.

Specific comments: 1. P1, L21-24: it is confusing to mention the dimension and details of calibration parameters in the abstract before the relevant descriptions are provided. 2. P2, L13-15: there are conflicting conclusions about the cost-effectiveness of GI, please provide references for your statements. In particular, the detention pond can be costly in terms of the construction and maintenance costs. 3. P2, L27-30: how the upstream area is discretized and the subcatchment are parameterized matter both in the modeling and calibration. Typical way to discretize subcatchments replies on GIS-based hydrological and landuse analyses to achieve reasonable characterization of natural drainage divisions. Any references to support your statements? 4. P3, L32: please define "a unit-area based analysis" 5. Figure 1: No legend for background landuse 6. P4, L15-20: A sketch of mentioned drainage system (manholes, pipes) is missing. Can author provide more information about the current drainage in the area? How many pipelines and manholes ? what is the current service level of the system? 7. Section 2.2.2: Details are needed to understand the spatial analysis used in the study. what are the inputs and resolution? What types of gis tools and processes are used to identify and digitize the 16 land covers? how do you estimate the future potential for GI implementation (e.g., to evaluate the potential of downspout disconnection for
a main building) and which parameters are used? 8. P6, L1-L10: Though Figure 3 depicts the different boundaries of BPA, I still don't understand how to set the BPA in SWMM and which parameter do you use to represent BPA? how did you choose the buffer widths in this study? Can author provide more information on how to use the "intersect" tool for estimating the BPA and SPA? 9. P7, L15-16: Authors considered DS-IA and DS PA in subcatchments, could authors show how the two parameters are obtained? Is it a simple characterization of the dry ponds and detention areas in subcatchment? 10. P7, L16-20: How did you choose the values for Scut and IMD? Can you provide more details on the division of IA into areas with or without DS? Also you mentioned several ways to route the internal flows, how do you model it in SWMM? 11. Section 2.4.2: (a) vegetation swale (VS) seems an appropriate option to represent BPA, how the authors determined the parameters for VS, e.g., berm height, vegetation volume fraction? (b) how the authors determined the values of initial saturation and % of subcatchment imperviousness draining to the BPA from the geoprocessing steps? (c) I am confused about the way to model BPA, is it modeled as a VS (LID competent), or an individual catchment, or changes in subcatchment imperviousness and width? Why set the width (60 feet) for BPA? 12. L9, L18-19: can authors give an detailed example on the evaluation of the groundwater flow in the study region? Is it calculated using Eq. 3 (then how the authors incorporated the equation in SWMM for groundwater simulation?) or just the difference between individual subcatchment surface and its nearest stream bottom? 13. Figure 5: what is the difference between Figure 4 and 5? it seems that both figures mainly give the depiction of the subcatchments. Adding regional drainage network (manholes, pipelines) are recommended. 14. P10,L26-28: Conceptual illustrations of the 6 options are well presented in Figure 6, but I find it difficult to understand how the 6 options are modeled in SWMM in details? for example, which subcatchment parameters are used to represent the different subareas (e.g. ICIA, TIA) and how to control the flow or routing directions? 15. P11, L20-24: any reference to support your assumptions on the lengths for overland flows and surface slopes? 16. P12, L2: A brief explaining of the method is recommend. 17. P12,
L7-10: one way to represent the GI can be the decrease of DCIA, which impacts the subcatchment imperviousness directly. That is one side of the problem, another is to attenuate the surface flow and slow down the speed. Is there any measure to model this aspect in your approach? 18. P13, Eq. 5-9: how to calculate the different Q values in SWMM? which result files are used to obtain these values? 19. P14, L16-18: I don't understand, if in option 4 where rainfall onto PA is completely captured by DS or infiltrated into soil, how come the simulated flow rates are much higher than the ones from the rest options? 20. P17, L1-2: without field measurement for valuation, how do you interpret the results? Given the clay type soil, 48% is much higher than expected. 21. P17, L15: can you provide some explanations on the increasing peak flow resulting from the GI scenario?

---

## Referee Comment (RC2) · Anonymous Referee #2 · 24 Jul 2017

General Comments: The manuscript "Subcatchment characterization for evaluating green infrastructure using the Storm Water Management Model" demonstrates a new discretization approach within SWMM for better representing green infrastructure (GI) components in urban storm water modeling. The topic is well placed and tackles an increasingly popular area - high-resolution hydrologic modeling as a result of increasing availability of high-resolution imagery. However, the lack of key information on model setup and modeling processes made it very difficult to understand how flow connectivity and thus hydrologic response were better represented on the subcatchment level through finer classification of impervious and pervious areas. I am not convinced by the 'reduced-order' calibration approach, and do not believe that this approach is transferrable to other systems given its fundamental issue (see detailed comments). Lastly, the authors should provide references and/or justifications to many modeling assumptions regarding parameterization in particular. Detailed Comments: 1) P2 Line 16: Please provide references of relevant studies. 2) P3 Line 32: Explain and provide references of unit-area based analysis. 3) P5 Line 27: Add 'to' following 'adjacent'. 4) P6 Line 5-10: It is not clear to me how the 'intersect' tool was used to separate BPA and SPA. It is also unclear how the buffer widths (0.30, 0.61, and 1.52 m) were chosen. 5) P6 Line 18-22: How was 0.5 acre chosen? Why subcatments of similar size help maintain hydrologic continuity? 6) P7 Section 2.4.1: 1) Move the description of calibration procedure from section 2.5 to here; 2) What are the values of Suct and IMD, and how were they initialized? Please also include them in Table 1; 3) Please provide how subarea routing was characterized within each subcatchment? 7) P8 Line 8-10: The authors stated that the initial values for "Length" were decided by averaging multiple field measurements of perceived overland flow lengths for each land cover type. How was overland flow length measured and generalized for each land cover that are spatially dispersed in the catchment? Plus, it is not reasonable to rescale the lumped flow lengths for each land cover to subcatchments with distinctive spatial connectivity to their respective outlet. The conventional SWMM approach is much more reasonable in this context. 8) P8 Section2.4.2: I do not understand how BPA and SPA was represented and spatially connected in SWMM. Based on the description, BPA was modeled as an LID component that receives flow from ICIA of subcatchment(s)? Looking at Figure 3&4, however, BPA seems to be lumped into a subcatchment. Why choosing the buffer width of 18.3m? Please clarify. 9) P10 Section 2.5: It is common in both spatially distributed and lumped hydrologic modeling that the land cover- and soil-specific parameters are fixed across the catchment. However, it is inappropriate to aggregate and calibrate by land cover the parameters of slope and overland flow length that are much more topography than land cover dependent. I can't agree with author's argument that this calibration approach is efficient or can be transferrable to other systems. 10) P10 Section 2.6: 1) If I understand it correctly, SWMM was calibrated using

the option 6 setup. The calibrated parameters include overland flow lengths and slopes as in Table 1. In this section, the authors provide new sets of flow lengths and slope parameters for different cover type, which are different from the values given in Table 1. Did all 6 options use the same parameterization or not? If yes, why not using the calibrated parameters? If no, the comparisons do not seem fair – calibrated option 6 vs. non-calibration options. 11) P14 Line 16-18: Why option 4 has the highest peak flow (in Figure 8a) if only DCIA discharges runoff? Figure 2: I suggest that the authors label the ID and show the baseline flow path of each surface record so the readers can better understand the difference between DCIA and ICA.

СЗ

---

## Author Comment (AC1) · 4 Sep 2017

Please find author's direct responses for the Referee-1 in Supplement (Responses for Referee-1.pdf).
* * *
Dear HESS Editor,

Before addressing the specific reviewer comments we want to re-emphasize the novel aspects of our research. We arranged a modeling approach using SWMM with the primary consideration of simulating GI scenarios for urban watersheds. Emphasis was

placed on the accuracy of the physical representation of the watershed landscape with respect to land cover types and their role in stormwater runoff dynamics. Given the nature of GI designs we consider this a necessity for model simulation. This differs from the more common (or conventional) context for which SWMM has served the urban drainage community, that has been to support drainage network design that minimize flooding risk with emphasis on more centralized and larger sized stormwater management structures. This overall objective is well explained in the Introduction. The subsequently presented novel (or innovative) aspects of our research include: 1) the introduction of a new concept, buffering pervious area (BPA). 2) The study of several different ways of parameterizing the subarea hydrologic connectivity within a SWMM subcatchment, of which we present the performance of 6 different options. Rather than applying each option at the full scale for our case study system we used a hypothetical-unit area approach. This greatly simplified the presentation of the differences among the options while still supporting our major findings and allowing us to explain the rationale for recommending one of the options for GI modeling in SWMM (Option 6) that we go on to demonstrate in the set-up of our SWMM model of the case study watershed. 3) The configuration of a land cover based parameterization framework that is implemented through a supporting spatial database. This explicitly accounts for changes in hydrologic connectivity that result from GI scenarios in the model set-up. While we have no way to prove that this improves the hydrologic simulation performance of the model, it does account for changes in the model set-up that reflect more the real changes to landscape hydrologic properties that would have to be ignored all together, or at best, lumped at a relatively large spatial scale to be represented using more common practices for SWMM set-up. Finally, 4) we present the value of flow hydrograph separation (Figure 13) while evaluating GI scenarios. The volumetric separation provides insight to the potential effectiveness and rationale for developing strategies for GI in a small watershed.

One general aspect of many comments we noted from both reviewers but primarily Reviewer 1 was the request for additional details and specifics on the use of GIS and

SWMM software to implement our approach. To keep the paper a reasonable length (for readability), we reduced some of the level of detail presented from earlier drafts and following suggestions of internal reviewers before we submitted to HESS. For example, the use of ArcGIS to clip, buffer, and intersect, we felt that these details are well known to standard GIS practitioners. However, we realize that these may not be as apparent to traditional hydrologic modelers. Therefore, we have prepared a "hands on" compilation of the detailed procedures with screen-shots that will be released separately through the US EPA as a standalone document in the coming weeks. This report can be referenced in the revision.

Our general interpretation of the Reviews is that Reviewer 1 had little issue or concern with the integrity of the approach presented. Conceptually it all made sense to him/her, but an unfamiliarity with ArcGIS and SWMM software made some of the explanations seem ill-explained or confusing. We had to try to strike a balance between providing highly detailed descriptions of software use while not losing sight of the main theme of demonstrating the utility of adopting the approach to SWMM set-up for GI modeling. We note in the address of each of Reviewer 1's comments where we will amend or adjust the MS to help account for most of their questions.

Reviewer 2, on the other hand, had some concern about the conceptual legitimacy of the approach. He/She was not convinced that the methods we develop and demonstrate have utility over a "conventional" SWMM modeling approach. However, Reviewer 2 did not provide specifics on this qualification of "conventional", so it made responding directly to their comments difficult. We addressed these the best we could based on our interpretation of their main points of concern. We note that in our presentation we never stated explicitly that adopting our approach would result in better hydrologic performance of the SWMM model. While we do expect this to be the case for GI simulation, specifically, we have no way of actually testing this because we lack data on the effect of GI on hydrology post implementation. What we can say exactly about our approach is that it allows for a more realistic expression of reality in the SWMM model

set-up. This should make the model output more accurate, but again, because we have no way to directly test this assertion we will be sure to 'tone-down' such implications where they may exist in the MS.

One aspect that Reviewer 2 (and Reviewer 1 also had a question about) was correct to point out was the potential inaccuracy in our assessment of what adopting our approach means for the effort placed during model calibration and the scalability to larger systems. Our claims in this regard were based on our experience modeling the case study system, which was relatively homogeneous with respect to topography and land cover spatial distribution. So, our approach accommodates GI scenario analysis and likely will reduce the effort required for calibration, but this depends on how homogeneous the landscape characteristics are in the project area. As Reviewer 2 points out, in many urban systems heterogeneity in topography alone could result in spurious modeling results if our framework was adopted as presented. This doesn't mean that the framework is incorrect, just that a higher order classification of subcatchments would be necessary. This can be accounted for in the configuration of the spatial database, and we will add this to the revision of the MS. Also, if a higher order classification for subcatchments is necessary this means that the parameter considerations during calibration increases beyond the dramatic reduction that we noted as a significant result of adopting our approach in the original version. We will correct these statements, as while they were true for our case study system, they might not be completely true for all systems. Either way, though, it is true that adopting the approach does reduce the number of parameters that may have to be considered during model calibration, which is an advantage. The disadvantage is the level of effort required in setting up a spatial database to support the modeling. Finally, we think Reviewer 2 misunderstood the overall point and discussion of our hypothetical unit area analysis. A few of the questions and comments here seemed to convolute the unit area analysis with the calibration of the SWMM model of the case study watershed after applying Option 6 for its setup. We will try to clarify the differences between these two, largely independent analyses, in their presentation in the MS.
Overall the questions and comments provided by these reviewers will result in several improvements in the MS, mainly by forcing some clarifying statements here and there and some adjustments to figures. They are much appreciated, and we hope that you will still consider the presentation of our research a worth contribution to HESS.

Sincerely, Joong Gwang Lee Christopher T. Nietch Srinivas Panguluri

Please also note the supplement to this comment:
https://www.hydrol-earth-syst-sci-discuss.net/hess-2017-166/hess-2017-166-AC1-supplement.pdf

**Supplement:**

**Anonymous Referee #1 Received and published: 27 June 2017**

The article by Lee et al tackles an interesting topic in the field of green infrastructure. The research approaches the paper investigated are meaningful for GI under smaller storm events. Some of the assumptions used in the paper need to be better explained and argued. The conclusions are not attended yet due to insufficient description of their methods, model settings, estimation of key parameters, in particular the part 2.2-2.4. Nonetheless, I think that the article had good potential for being published, provided that the following comments are adequately addressed.

**General response to Reviewer 1:**

Generally, it seems that this reviewer had little issue with the approach, the performance evaluation of the approach, or the demonstration of the effectiveness of the approach at evaluating GI, all of which were the main points to be covered in the MS. Rather, He/She was more interested in understanding how to implement the approach given what seems to be familiarity lack of familiarity with ArcGIS or SWMM software. We will provide clarifying text where appropriate as noted below in the direct responses. We will direct such readers to our Report that addresses this matter to a certain extent as the details apply specifically to the implementation of the approach. But many of the Reviewer's points of confusion and questions can be addressed through consult of existing tutorials and user guides of the software used to develop and conduct this study. We will make attempts to clarify this aspect as noted above (i.e., that this MS is not meant to serve as a tutorial for GIS and SWMM software) in the revision while making sure to cite relevant references where such information exists.

**General comments:**

1. The research approach replies on a highly resolved spatial database of urban land cover, stormwater drainage feature and topography, what about its potential application in a general context? Most of urban areas may not have such detailed dataset or require extensive surveying and modeling efforts.

- ⇒ Indeed, highly resolved spatial databases are not always available for many urban areas. This is because these GIS databases can be expensive to develop and maintain; and/or may not be required for conventional stormwater management purposes. However, in our experience more and more municipalities in the U.S., at least, are developing and improving their spatial databases of stormwater infrastructure. To address this comment we would add the following content to section 2.2.1: "Existing databases that include the details for the stormwater infrastructure in this watershed are not always readily available to the modeler. In these cases, to adopt the subsequently described approach to GI scenario modeling in SWMM could require considerable ground-truthing and site surveying. In lieu of onsite visits, and as will become apparent from the descriptions below, what would be most important is determining the spatial location of storm sewer inlets. These are often visible from readily available aerial photographs. When elevation data for the storm sewer network is unavailable, much can be inferred using surface elevation data and assuming local construction codes for stormwater infrastructures, such as catch basin depths, and conveyance pipe diameters and slopes were applied. Such approximations would suffice for GI scenario analysis considerations, where storm sewer design is not the primary focus."
- Also, the reviewer includes land cover and topographic information in his/her assessment on data availability. For land cover, the availability is somewhat irrelevant, as our approach requires land cover analysis and detailed digitization to do the subarea parameterization in SWMM that we describe in the manuscript (MS). This land cover and subsequent Subarea categorization is described in fairly specific detail in section 2.2.2. We struggled with how detailed the descriptions needed to be during the preparation of earlier drafts. Two internal reviewers, prior to submission to HESS, suggested the detail was too much and that the MS was too long. To address this issue we have prepared a companion report that will be published as a USEPA, Office of Research and

Development contribution, that will be freely available to anyone interested. We will reference this report in the final version of the HESS MS, should it be accepted for publication.

2. Relevant references are needed to support statements in the text, see specific comments for details. The key definitions (e.g., DCIA, ICIA, SPA, BPA) are given, but a conceptual model characterizing these key processes in a watershed and their spatial connections should be provided.

- ⇒ We will add a new figure that will depict a conceptual schematic that provides context to the DCIA, ICIA, SPA, and BPA categorization. This can be in the form of a side view of a home situated along a street with storm sewer infrastructure depicted.
- ⇒ It is important to note, however, that these areas are defined within a subcatchment for SWMM modeling, not at the watershed scale. A watershed in SWMM consists of a number of subcatchments, which interact based on the existing storm collection system.

3. The land cover characterization in GIS is an essential step to provide inputs for hydrological evaluation in SWMM. Very limited information is given to understand how it is done in GIS analysis. Also readers need more details on how the four types of subareas are subsequently modeled in SWMM (e.g. parameter settings), e.g., how to parameterize BPA, ICIA, SPA for subcatchments.

As noted under 1. above, we struggled with the level of details to provide. Our intent for the MS was to focus on evaluating the performance of the approach to modeling GI in SWMM. For readers that want specific guidance on implementing the approach the USEPA report is being prepared. This can be referenced in the final version of the MS. It includes details on how to process clip, intersect, union, and manipulating attribute data in ArcGIS. Much of this will be familiar to users of ArcGIS, so we tried to strike a balance in the MS. If the Editor prefers a different tact to providing this information we could try to include as a supplemental section or appendix.

4. A better description of model calibration process is recommended, e.g., summary of parameters, inputs and outputs, criteria of performance.

⇒ We disagree with this comment. We provide quite a bit of detail on our approach to calibration in section 2.5 and show results of sensitivity analysis in Figure 10; a standard approach to model calibration, as well as giving initial and calibrated values of the sensitive input parameters in Table 1. What we failed to include was the initial and calibrated value for the width of BPA. This will be added to Table 1 in the final version of the MS. The output of calibration, along with performance statistics (i.e., NSE and  $R^2$ ) is provided in Figure 11. So, we believe the level of detail is sufficient and actually contrary to what the reviewer suggests.

**Specific comments:**

1. P1, L21-24: it is confusing to mention the dimension and details of calibration parameters in the abstract before the relevant descriptions are provided.

⇒ This text in the abstract is not meant to note dimension and details of specific calibration parameters, rather to note a significant aspect about the approach to SWMM set-up that is presented in this study. What the text indicates in the abstract is that adopting the approach reduces the number of parameters that might be considered during calibration. However, while Reviewer 1 likely misunderstood the context for the description, Reviewer 2 correctly points out some inaccuracies in this statement (see general responses above and specific responses below) so we will eliminate it from the abstract

2. P2, L13-15: there are conflicting conclusions about the cost-effectiveness of GI, please provide references for your statements. In particular, the detention pond can be costly in terms of the construction and maintenance costs.

⇒ That is why we put the word "may" in the sentence on L12. We will add the following to the sentence at L13: "...., like detention ponds, especially in cases where land is not available or very expensive."

3. P2, L27-30: how the upstream area is discretized and the subcatchment are parameterized matter both in the modeling and calibration. Typical way to discretize subcatchments replies on GIS-based hydrological and landuse analyses to achieve reasonable characterization of natural drainage divisions. Any references to support your statements?

⇒ Seems like these comments are related to P3 (not P2), L27-30. We will try to include the presented criteria by the reviewer with relevant references, e.g., SWMM Application Manual (Gironás et al., 2009).

4. P3, L32: please define "a unit-area based analysis"

⇒ The term unit-area is a relatively common term in the field of stormwater modeling that refers to normalizing model output by using a common spatial dimension, e.g., in our case, 1 acre. We don't think it is warranted to define this relatively standard term in the introduction. Furthermore, in addition to the spatial dimension we define the land cover characteristics of the unit-area specifically for this study beginning on P10, L22. and at P11, L4. We will add to this sentence the word 'unit' to "hypothetical area" to help clarify.

5. Figure 1: No legend for background landuse

⇒ Good catch. A relevant legend for the drainage system and the land use categorization will be added to the map.

6. P4, L15-20: A sketch of mentioned drainage system (manholes, pipes) is missing. Can author provide more information about the current drainage in the area? How many pipelines and manholes ? what is the current service level of the system?

⇒ The existing drainage system is presented in Figure 5. A legend will be added to the figure to help define it. We don't see how statistics on number of pipes and manholes or 'current service level' of the system is relevant.

7. Section 2.2.2: Details are needed to understand the spatial analysis used in the study. what are the inputs and resolution? What types of GIS tools and processes are used to identify and digitize the 16 land covers? how do you estimate the future potential for GI implementation (e.g., to evaluate the potential of downspout disconnection for a main building) and which parameters are used?

- ⇒ We used 0.76 m LiDAR as noted in P6, L18. We felt the level of detail called for by the reviewer unwarranted for the specific purpose of this MS, which is to highlight the specific aspects and provide results of the analysis of performance of the approach developed for GI Analysis in SWMM. And as mentioned earlier, details on GIS analysis are included in the USEPA report that will be referenced in the final version of the MS, or if this is deemed insufficient, we can try to cut and paste relevant sections for addition to supplementary materials section or appendix. The referenced report includes how to process clip, intersect, union, and manipulating attribute data.
- ⇒ For the third question; a systematic approach to 'estimate the future potential for GI implementation' would be quite difficult given uniqueness of place considerations, and is beyond the scope of this research.

8. P6, L1-L10: Though Figure 3 depicts the different boundaries of BPA, I still don't understand how to set the BPA in SWMM and which parameter do you use to represent BPA? how did you choose the buffer widths in this study? Can author provide more information on how to use the "intersect" tool for estimating the BPA and SPA?

⇒ The description of how to set-up the BPA in SWMM starts on P8, L31. We note that the original widths of the BPA are arbitrarily determined and explain why this has to be the case on P6. To provide more details on using the intersect and other functions in ArcGIS would require a step-by-step approach to using ArcGIS software. We, in fact, provide this detail in the USEPA report that is undergoing internal review and will be referenced in the MS, but we feel it is inappropriate for this MS to call for a tutorial on how to use certain functions in ArcGIS. An interested reader can find this information searching the help menu and user guides of the ArcGIS software.

9. P7, L15-16: Authors considered DS-IA and DS\_PA in subcatchments, could authors show how the two parameters are obtained? Is it a simple characterization of the dry ponds and detention areas in subcatchment?

⇒ DS stands for depression storage, as noted in the list of abbreviations. DS is a standard term used in urban hydrology that denotes the depth of water that can collect on urban surfaces. The initial value assigned to DS per land cover type was assigned based on recommendations or defaults described in the SWMM User's manual, as noted on P8, L19. DS has nothing to do with dry ponds or other built detention areas.

10. P7, L16-20: How did you choose the values for Scut and IMD? Can you provide more details on the division of IA into areas with or without DS? Also you mentioned several ways to route the internal flows, how do you model it in SWMM?

⇒ We assign these values, in particular, using recommendations from the SWMM user manuals as noted on P8, L2: We will make note of this for the infiltration parameters earlier in this same section to help clarify. As for the other questions posed here, these can be answered for interested readers by consulting the SWMM user's manual documentations already referenced. We don't think addressing these questions with new additions to the text is warranted. It becomes more apparent with each comment that this reviewer has little experience using SWMM, we feel it is only necessary to go into the details of how to model urban hydrology using SWMM as they pertain to the described approach to GI scenario analysis. It is not our job to provide a tutorial on how to use SWMM. These are available at the SWMM download site, which will be referenced in the revision

11. Section 2.4.2: (a) vegetation swale (VS) seems an appropriate option to represent BPA, how the authors determined the parameters for VS, e.g., berm height, vegetation volume fraction? (b) how the authors determined the values of initial saturation and % of subcatchment imperviousness draining to the BPA from the geoprocessing steps? (c) I am confused about the way to model BPA, is it modeled as a VS (LID competent), or an individual catchment, or changes in subcatchment imperviousness and width? Why set the width (60 feet) for BPA?

⇒ We will try to clarify further in the revision, but generally we already state that parameter values are set based on guidance from the SWMM user manuals or from our experience working in urban areas. All of these details will be added to help clarify, including, berm height (0.1-in or 2.54-mm to minimize any storage effect within the berm, which is the case for real BPA), vegetation volume fraction (0, this is assumed to be negligible.), % imperviousness draining to the BPA (ICIA / TIA, where TIA = DCIA+ICIA). BPA is modeled as a VS (SWMM LID option) within a subcatchment, not as an individual subcatchment. We further acknowledge that many aspects of the BPA are unverifiable, and rationalize why this is not relevant to the integrity of the approach in section 2.3.2.

12. L9, L18-19: can authors give an detailed example on the evaluation of the groundwater flow in the study region? Is it calculated using Eq. 3 (then how the authors incorporated the equation in SWMM for groundwater simulation?) or just the difference between individual subcatchment surface and its nearest stream bottom?

▷ No, we cannot provide more detail on groundwater flow. As mentioned in the manuscript, there was no observational data on groundwater flow. This is typically the case in urban modeling applications using SWMM. We will clarify that groundwater modeling parameters were defined using the SWMM Reference Manual and users' group knowledge base (e.g.,

https://www.openswmm.org/Topic/1465/groundwater-parameters; https://www.openswmm.org/Topic/4840/groundwater-values). The remaining questions in this response are irrelevant to our study.

13. Figure 5: what is the difference between Figure 4 and 5? it seems that both figures mainly give the depiction of the subcatchments. Adding regional drainage network (manholes, pipelines) are recommended.

⇒ Figure 4 is map of the watershed with relevant land cover and subcatchment delineation. Figure 5 is the conceptual representation of the area being model in the SWMM software, which includes the configuration of the storm sewer drainage network. As mentioned earlier, we will add a relevant legend to Figure 5, to help clarify.

14. P10,L26- 28: Conceptual illustrations of the 6 options are well presented in Figure 6, but I find it difficult to understand how the 6 options are modeled in SWMM in details? for example, which subcatchment parameters are used to represent the different subareas (e.g. ICIA, TIA) and how to control the flow or routing directions?

⇒ As shown in the legend, each rectangular represents a subcatchment in SWMM, and the dotted line divides subareas within the subcatchment. A rectangle without a dotted line means the subcatchment consists of a single (homogeneous) subarea, either 100% impervious or pervious. The arrows represent flow routing directions. We will add these clarifications. The legend in the figure will be updated.

15. P11, L20-24: any reference to support your assumptions on the lengths for overland flows and surface slopes?

⇒ Length of overland flow means the flow length where the flow is maintained as overland flow (or sheet flow). It doesn't mean the physical length of a drainage area. This has long been a point of confusion in SWMM modeling. We will attempt to clarify in the revision. Surface slopes of typical urban drainage features are based on construction code or are inferred based on the GIS. The relevant references will be added.

16. P12, L2: A brief explaining of the method is recommend.

The following brief description will be added: "The 95th percentile rainfall event is defined as the measured precipitation depth accumulated over a 24-hour period for the period of record that ranks as the 95th percentile rainfall depth based on the range of all daily event occurrences during this period."

17. P12, L7-10: one way to represent the GI can be the decrease of DCIA, which impacts the subcatchment imperviousness directly. That is one side of the problem, another is to attenuate the surface flow and slow down the speed. Is there any measure to model this aspect in your approach?

⇒ The effect of GI scenarios on the temporal dynamics of runoff is considered by comparing storm hydrographs before and after GI addition to the model. To address this comment we will add a note on the temporal changes to the storm hydrographs shown in Figure 13 to the results. Namely something like this: "It is interesting to note from Figure 13 that the peak flow for the event depicted in the figure is slightly higher in the GI Scenario, but that the duration of flows slightly smaller than this peak is longer in the baseline scenario." 18. P13, Eq. 5-9: how to calculate the different Q values in SWMM? which result files are used to obtain these values?

⇒ This seems to be another question about SWMM modeling basics, to include the details of which are not appropriate for this MS. The reviewers question can be answered by consulting the SWMM user's documentation. If this question is based on the hydrograph separation procedure the calculations for the individual Q values are explained in the manuscript. Further action related to this comment is unwarranted.

19. P14, L16-18: I don't understand, if in option 4 where rainfall onto PA is completely captured by DS or infiltrated into soil, how come the simulated flow rates are much higher than the ones from the rest options?

⇒ The following description will be added to help explain the results observed for option 4: "Hydrologic connectivity is very important. In Option 4, the one-acre area is modeled as a single subcatchment with two subareas: IA and PA. Because this setup ignores the difference between DCIA and ICIA, the entire impervious area (subarea IA) is actually modeled the same as DCIA, which means all of the runoff is discharged to the storm drainage system directly with no abatement. Under a small storm (like <1-month storm), runoff occurs only from impervious area, more specifically only from DCIA. For small storms, runoff from ICIA is completely controlled by BPA (if ICIA exists), but no ICIA is modeled under Option 4. Because of this, modeled runoff from this option is higher than any of the other options.

20. P17, L1-2: without field measurement for valuation, how do you interpret the results? Given the clay type soil, 48% is much higher than expected.

⇒ We think this comment is based on a mis-understanding of the term interflow. We validated the total flow at the outlet of the watershed for the baseline condition using observed flow data. By applying artificial modeling conditions (e.g., DCIA only, or excluding the groundwater component as described in the manuscript), we tried to show how to develop more effective GI implementation scenarios. These conditions are contemplative and do not actually exist, so they cannot be validated with measured data. The 48% interflow doesn't mean all the 48% flow discharges through the entire soil layer as groundwater. There would be considerable amount of very shallow subsurface flow that discharges through a shallow layer near the surface with a relatively high porosity. We will try to clarify in the revision.

21. P17, L15: can you provide some explanations on the increasing peak flow resulting from the GI scenario?

⇒ Definitely. We will add the following explanation: Overall the flow volume is reduced from the GI scenario. However, when the peak occurred around 15:30 (shown in Figure 13 the capacity of the GI for controlling stormwater was already exceeded because of controlling runoff during the previous rainfall that occurred between 7:00 and 14:00. Under this saturated condition, even the direct rainfall to the GI area will be discharged with minimum abatement. If there is no GI (as in the baseline condition), the same area receives only direct rainfall, there is no additional runon from impervious area, and that rainfall is controlled by still available surface depression storage and not-saturated infiltration capacity.

---

## Author Comment (AC2) · 4 Sep 2017

Please find author's direct responses for the Referee-2 in Supplement (Responses for Referee-2.pdf).
* * *
Dear HESS Editor,

Before addressing the specific reviewer comments we want to re-emphasize the novel aspects of our research. We arranged a modeling approach using SWMM with the primary consideration of simulating GI scenarios for urban watersheds. Emphasis was

placed on the accuracy of the physical representation of the watershed landscape with respect to land cover types and their role in stormwater runoff dynamics. Given the nature of GI designs we consider this a necessity for model simulation. This differs from the more common (or conventional) context for which SWMM has served the urban drainage community, that has been to support drainage network design that minimize flooding risk with emphasis on more centralized and larger sized stormwater management structures. This overall objective is well explained in the Introduction. The subsequently presented novel (or innovative) aspects of our research include: 1) the introduction of a new concept, buffering pervious area (BPA). 2) The study of several different ways of parameterizing the subarea hydrologic connectivity within a SWMM subcatchment, of which we present the performance of 6 different options. Rather than applying each option at the full scale for our case study system we used a hypothetical-unit area approach. This greatly simplified the presentation of the differences among the options while still supporting our major findings and allowing us to explain the rationale for recommending one of the options for GI modeling in SWMM (Option 6) that we go on to demonstrate in the set-up of our SWMM model of the case study watershed. 3) The configuration of a land cover based parameterization framework that is implemented through a supporting spatial database. This explicitly accounts for changes in hydrologic connectivity that result from GI scenarios in the model set-up. While we have no way to prove that this improves the hydrologic simulation performance of the model, it does account for changes in the model set-up that reflect more the real changes to landscape hydrologic properties that would have to be ignored all together, or at best, lumped at a relatively large spatial scale to be represented using more common practices for SWMM set-up. Finally, 4) we present the value of flow hydrograph separation (Figure 13) while evaluating GI scenarios. The volumetric separation provides insight to the potential effectiveness and rationale for developing strategies for GI in a small watershed.

One general aspect of many comments we noted from both reviewers but primarily Reviewer 1 was the request for additional details and specifics on the use of GIS and

SWMM software to implement our approach. To keep the paper a reasonable length (for readability), we reduced some of the level of detail presented from earlier drafts and following suggestions of internal reviewers before we submitted to HESS. For example, the use of ArcGIS to clip, buffer, and intersect, we felt that these details are well known to standard GIS practitioners. However, we realize that these may not be as apparent to traditional hydrologic modelers. Therefore, we have prepared a "hands on" compilation of the detailed procedures with screen-shots that will be released separately through the US EPA as a standalone document in the coming weeks. This report can be referenced in the revision.

Our general interpretation of the Reviews is that Reviewer 1 had little issue or concern with the integrity of the approach presented. Conceptually it all made sense to him/her, but an unfamiliarity with ArcGIS and SWMM software made some of the explanations seem ill-explained or confusing. We had to try to strike a balance between providing highly detailed descriptions of software use while not losing sight of the main theme of demonstrating the utility of adopting the approach to SWMM set-up for GI modeling. We note in the address of each of Reviewer 1's comments where we will amend or adjust the MS to help account for most of their questions.

Reviewer 2, on the other hand, had some concern about the conceptual legitimacy of the approach. He/She was not convinced that the methods we develop and demonstrate have utility over a "conventional" SWMM modeling approach. However, Reviewer 2 did not provide specifics on this qualification of "conventional", so it made responding directly to their comments difficult. We addressed these the best we could based on our interpretation of their main points of concern. We note that in our presentation we never stated explicitly that adopting our approach would result in better hydrologic performance of the SWMM model. While we do expect this to be the case for GI simulation, specifically, we have no way of actually testing this because we lack data on the effect of GI on hydrology post implementation. What we can say exactly about our approach is that it allows for a more realistic expression of reality in the SWMM model

set-up. This should make the model output more accurate, but again, because we have no way to directly test this assertion we will be sure to 'tone-down' such implications where they may exist in the MS.

One aspect that Reviewer 2 (and Reviewer 1 also had a question about) was correct to point out was the potential inaccuracy in our assessment of what adopting our approach means for the effort placed during model calibration and the scalability to larger systems. Our claims in this regard were based on our experience modeling the case study system, which was relatively homogeneous with respect to topography and land cover spatial distribution. So, our approach accommodates GI scenario analysis and likely will reduce the effort required for calibration, but this depends on how homogeneous the landscape characteristics are in the project area. As Reviewer 2 points out, in many urban systems heterogeneity in topography alone could result in spurious modeling results if our framework was adopted as presented. This doesn't mean that the framework is incorrect, just that a higher order classification of subcatchments would be necessary. This can be accounted for in the configuration of the spatial database, and we will add this to the revision of the MS. Also, if a higher order classification for subcatchments is necessary this means that the parameter considerations during calibration increases beyond the dramatic reduction that we noted as a significant result of adopting our approach in the original version. We will correct these statements, as while they were true for our case study system, they might not be completely true for all systems. Either way, though, it is true that adopting the approach does reduce the number of parameters that may have to be considered during model calibration, which is an advantage. The disadvantage is the level of effort required in setting up a spatial database to support the modeling. Finally, we think Reviewer 2 misunderstood the overall point and discussion of our hypothetical unit area analysis. A few of the questions and comments here seemed to convolute the unit area analysis with the calibration of the SWMM model of the case study watershed after applying Option 6 for its setup. We will try to clarify the differences between these two, largely independent analyses, in their presentation in the MS.

[Figure]

Overall the questions and comments provided by these reviewers will result in several improvements in the MS, mainly by forcing some clarifying statements here and there and some adjustments to figures. They are much appreciated, and we hope that you will still consider the presentation of our research a worth contribution to HESS.

Sincerely, Joong Gwang Lee Christopher T. Nietch Srinivas Panguluri

Please also note the supplement to this comment:
https://www.hydrol-earth-syst-sci-discuss.net/hess-2017-166/hess-2017-166-AC2-supplement.pdf

**Supplement:**

**General Comments:**
The manuscript "Subcatchment characterization for evaluating green infrastructure using the Storm Water Management Model" demonstrates a new discretization approach within SWMM for better representing green infrastructure (GI) components in urban storm water modeling. The topic is well placed and tackles an increasingly popular area - high-resolution hydrologic modeling as a result of increasing availability of high-resolution imagery. However, the lack of key information on model setup and modeling processes made it very difficult to understand how flow connectivity and thus hydrologic response were better represented on the subcatchment level through finer classification of impervious and pervious areas. I am not convinced by the 'reduced-order' calibration approach, and do not believe that this approach is transferrable to other systems given its fundamental issue (see detailed comments). Lastly, the authors should provide references and/or justifications to many modeling assumptions regarding parameterization in particular.

*General Response to Reviewer 2.*
⇨ *The fundamental criteria for the presented approach to GI analysis in SWMM is 1) to base the subcatchment delineation on landscapes draining to storm sewer inlets and 2) configure the subarea routing within each subcatchment so that the real relevance of differences among DCIA, ICIA, and BPA are accounted for in the SWMM parameterization. The first part requires knowing the location of the storm sewer inlets and the second piece relies on the highly resolved spatial database to conduct the set-up.*
⇨ *Flow connectivity within a drainage area is presented in Figure 6. We introduced a new concept of buffering pervious area (BPA) for improving the physical representation of hydrologic response. In common SWMM modeling, all pervious area is treated the same (as in Options 4 or 5 in Figure 6), even though only the BPA can receive waters from impervious area, specifically from ICIA. As shown in Figure 8, simulated runoff by Options 4 or 5 would be very inaccurate, especially for the <1 year small storms. We explain in the MS why Options 4 and 5 resulted in dissimilar responses as they depart significantly from physical reality for SWMM set up.*
⇨ *We are not trying to argue that the approach to SWMM set-up is a 'better' representation of hydrologic response. While we do expect this to be the case for GI simulation, specifically, we have no way of actually testing this because we lack data on the effect of GI on hydrology post implementation. What we can say exactly about our approach is that it allows for a more realistic expression of reality in the SWMM model set-up. This should make the model output more accurate by reducing overall model uncertainty, but again, because we have no way to directly test this assertion we will be sure to 'tone-down' such implications where they exist in the MS. We due compare the performance of our recommended subcatchment set-up approach to others in Figures 6 and 8. While Option 1 should be the most accurate among all options presented, we advocate option 6 for GI analysis in SWMM because options 1, 2, and 3 would result in many more subcatchments to parameterize, and more effort would have to be placed adjusting model set-up to account for GI scenarios. Furthermore, Option 6 allows for subcatchment delineation based on topography and, therefore, has a physical meaning within the context of a watershed approach, while Options 1 through 3 would require disassociating the subcatchment context from reality to a more conceptual basis in the model. We will add this explanation to the MS.*
⇨ *Our approach to calibration in SWMM is no different than what would be considered the more typical approach in terms of the actually modeling steps required, i.e., sensitivity analysis followed by one at a time adjustment, re-run, compare simulated vs. observed. What is different is that we argue that the number of parameters that one might consider to adjust during calibration can be quite large if each subcatchment has unique values, or has been considered independently of all the other*

*subcatchments, or as we describe it j x k number of parameters. We rely on the detailed spatial resolution of reality and the relatively small subcatchment size (driven by the storm sewer inlet, delineation requirement), to standardize parameter values across them. The Reviewer is correct to point out that in some watersheds spatial heterogeneity in topography and soils may nullify the assumption of commonality among all of them. If a land cover does not maintain the sufficient level of homogeneity within the target watershed for modeling, we need to use more than one set of parameters for the land cover. In this case, we should divide the land cover into sub-groups that represent the heterogeneous hydrologic properties independently. We will add more details and discussion to address this valid concern.*

**Detailed Comments:**

1) P2 Line 16: Please provide references of relevant studies.

⇨ *References will be provided, e.g., the recently published SWMM review paper by Niazi et al. (2017).*

2) P3 Line 32: Explain and provide references of unit-area based analysis.

⇨ *As mentioned in the manuscript (P10, L22), a unit-area is a hypothetical area, which represents a typical urban drainage area. SWMM can model a drainage area with various level of spatial aggregation, as shown in Figure 6. We arranged the unit-area based analysis demonstrate a "balanced" way for subcatchment characterization, based on the level of effort in model set-up and the accuracy in modeling results particularly for GI analysis.*

3) P5 Line 27: Add 'to' following 'adjacent'.

⇨ *We will add 'to'.*

4) P6 Line 5-10: It is not clear to me how the 'intersect' tool was used to separate BPA and SPA. It is also unclear how the buffer widths (0.30, 0.61, and 1.52 m) were chosen.

⇨ *We used ArcGIS to process the intersect analysis. As mentioned earlier, we feel it is inappropriate for this MS to call for a tutorial on how to use certain functions in ArcGIS. To provide more details on using the intersect and other functions in ArcGIS, we are preparing the USEPA report. The buffer width was selected when we calibrated the model. We arranged three SWMM models that represent three different sizes of BPA. We determined which one among the three cases of sizing BPA provided the more accurate simulation compared to the observed flow data. In this way, the BPA width was treated as a calibration parameter (see Figure 10).*

5) P6 Line 18-22: How was 0.5 acre chosen? Why subcatchments of similar size help maintain hydrologic continuity?

⇨ *Before conducting subcatchment delineation, we rather arbitrarily chose 0.5 acre to combine a drainage area with a neighboring subcatchment to minimize effort in model setup. In the actual analysis this happened only a few times. Maintaining similarity among subcatchment sizes confines the hydrologic loads received by the drainage system to a narrow range that helps to minimize errors in the simulation that might arise from surcharging or flooding due to mis-matched pipe network sizing. We can add this to the MS.*

6) P7 Section 2.4.1: 1) Move the description of calibration procedure from section 2.5 to here; 2) What are the values of Suct and IMD, and how were they initialized? Please also include them in Table 1; 3) Please provide how subarea routing was characterized within each subcatchment?

⇨ *We don't think it makes better sense to discuss model calibration until all of the major aspects of model parametrization and set-up are attended to. Based on the soil type, the values for Suct were selected using the SWMM User's Manual. The actual IMD is dynamically updated at every modeling time step. As presented in P10, L1-2, the developed SWMM model for the study area was run for a*

*six-month period (01 April 2009 to 31 August 2009) where the first four months of this period were used to stabilize the continuous simulation. While IMD was modeled using the default values in the EPA-SWMM, the IMD at the beginning of reporting the modeling results may not be affected (or minimally affected) by the initial values in model setup. We will provide more explanation of how subarea routing is configured in SWMM.*

7) P8 Line 8-10: The authors stated that the initial values for "Length" were decided by averaging multiple field measurements of perceived overland flow lengths for each land cover type. How was overland flow length measured and generalized for each land cover that are spatially dispersed in the catchment? Plus, it is not reasonable to rescale the lumped flow lengths for each land cover to subcatchments with distinctive spatial connectivity to their respective outlet. The conventional SWMM approach is much more reasonable in this context.

⇨ *We measured length based on the physical overland flow path of each individual land cover type. For example, length for a building is measured from roof crest to gutter. Length for street is measured from pavement crest (or centerline) to curb (or pavement boundary). Length for a driveway is measured as the distance between the house and street. The SWMM Applications Manual (Gironás et al., 2009), cited in the MS (P8 Line 12), suggests "If the overland flow length varies greatly within the subcatchment, then an area-weighted average should be used."*

⇨ *We don't understand what the Reviewer means by "it is not reasonable to rescale the lumped flow lengths for each land cover to subcatchments with distinctive spatial connectivity to their respective outlet". We acknowledge that maintaining spatial homogeneity among subcatchments properties should be a priority in the application of our approach, and as discussed above will provide content to explain what to do in the spatial database set-up to account for this.*

⇨ *We can't be completely sure of what the Reviewer means by 'conventional approach". Nonetheless, we did not intend to imply that our approach is 'better', generally. However, a common approach to subcatchment set-up is studied in Options 4 or 5 of Figure 6. Using either of these makes the application of GI an implicit consideration. We will add this explanation and clarification to the MS. In this study, we intended to examine an alternative for characterizing a drainage area.*

8) P8 Section2.4.2: I do not understand how BPA and SPA was represented and spatially connected in SWMM. Based on the description, BPA was modeled as an LID component that receives flow from ICIA of subcatchment(s)? Looking at Figure 3&4, however, BPA seems to be lumped into a subcatchment. Why choosing the buffer width of 18.3m? Please clarify.

⇨ *We will clarify as follows: In SWMM, BPA is modeled as vegetated swale. The size of BPA can be defined for each subcatchment. The % contributing impervious area to the BPA can be also defined for each subcatchment, which is ICIA/TIA where TIA=DCIA+ICIA. Since the total pervious area (TPA) remains identical for each subcatchment, the sizes of SPA for individual subcatchments can be decided as SPA = TPA – BPA for the three different sizes of BPA (which were derived by applying three different distances for proximity analysis in GIS). When we calibrated the model, we checked which one, among the three cases of sizing BPA, would calibrate the best for various storm sizes. More clarification will be added to the figures also.*

9) P10 Section 2.5: It is common in both spatially distributed and lumped hydrologic modeling that the land cover- and soil-specific parameters are fixed across the catchment. However, it is inappropriate to aggregate and calibrate by land cover the parameters of slope and overland flow length that are much more topography than land cover dependent. I can't agree with author's argument that this calibration approach is efficient or can be transferrable to other systems.

⇨ *The Reviewer is correct to point this out, and we will qualify our statements about transferability accordingly. If a land cover does not maintain the sufficient level of homogeneity across the target watershed we need to use more than one set of parameters for the land cover. In this case, we should*

*divide the land cover into sub-groups that represent the heterogeneous hydrologic properties independently. As noted above we will explain the relevance of this issue and provide a remedy for it in the MS. While we agree that it is more accurate to select values for slope and overland flow length based on topography, generally speaking, SWMM subcatchment areas as defined by modelers tend to be somewhat topographically homogeneous, otherwise accurate model representation is difficult. Also, urban sewer collection systems are zoned in a manner accounting for local variation in topography. Even land use generally follows topography, therefore, land cover is a reasonable surrogate.*

10) P10 Section 2.6: 1) If I understand it correctly, SWMM was calibrated using the option 6 setup. The calibrated parameters include overland flow lengths and slopes as in Table 1. In this section, the authors provide new sets of flow lengths and slope parameters for different cover type, which are different from the values given in Table 1. Did all 6 options use the same parameterization or not? If yes, why not using the calibrated parameters? If no, the comparisons do not seem fair – calibrated option 6 vs. non-calibration options.

⇨ *Table 1 shows the initial and calibrated parameters for the study area, not the hypothetical unit-area analysis. Therefore, there seems to be some confusion here, so we will attempt to clarify further in the MS. For the hypothetical area analysis, all of the 6 options were arranged using the same spatial and hydrologic characteristics as presented in P11, L23-30. However, the ways to model DCIA, ICIA, BPA, and SPA are different among the options. For the hypothetical unit area analysis calibration was not necessary.*

11) P14 Line 16-18: Why option 4 has the highest peak flow (in Figure 8a) if only DCIA discharges runoff?

⇨ *In option 4, the entire impervious area is modeled as DCIA, i.e., ICIA is also modeled as DCIA. There is no runon from impervious to pervious areas in option 4. (Please also see the responses under the comment #19 for Reviewer 1.)*

Figure 2: I suggest that the authors label the ID and show the baseline flow path of each surface record so the readers can better understand the difference between DCIA and ICA.

⇨ *The figure will be amended. The attribute table shown in the figure will also be amended to minimize any miss-interpretation.*

---

## Author Response (AR1)

**General Notes from the Authors:**

- We included our original detailed responses (in italics) to reviewer's comments submitted on 09/02/2017. In those we highlighted how we WOULD change the manuscript in response to the comment. Here, after considering the Editor's response to our original comments, we include the location of those changes and give further explanations as appropriate (in bold and italicized).
- ⇒ We amended the title to reflect changes we made in the manuscript to try to avoid confusion around the term "Subcatchment". In the original we were using subcatchment to describe both real-physical landscape areas as well as a fundamental modeling component in SWMM. In this version we use the term subcatchment only when writing about the SWMM model, as this is the term that the model developers use. We use drainage area or hydrologic response element (HRE) to refer to physical landscape areas draining to the stormwater conveyance system. Please see P3/L8-15 in the revised manuscript.
- ⇒ We made major changes to one of the key Figures (now Figure 5) to help clarify what changes were made in the hypothetical HRE analysis (previously referred to as hypothetical unit-area analysis).
- ⇒ We added a new figure, Figure 6, that diagrams the workflow for developing the baseline SWMM model and adding GI scenarios.
- ⇒ Section 2.6 in the previous version is relocated to Section 2.4. The section title is also amended.

**Anonymous Referee #1 Received and published: 27 June 2017**

The article by Lee et al tackles an interesting topic in the field of green infrastructure. The research approaches the paper investigated are meaningful for GI under smaller storm events. Some of the assumptions used in the paper need to be better explained and argued. The conclusions are not attended yet due to insufficient description of their methods, model settings, estimation of key parameters, in particular the part 2.2-2.4. Nonetheless, I think that the article had good potential for being published, provided that the following comments are adequately addressed.

**General comments:**

1. The research approach replies on a highly resolved spatial database of urban land cover, stormwater drainage feature and topography, what about its potential application in a general context? Most of urban areas may not have such detailed dataset or require extensive surveying and modeling efforts.

- Original Response: Indeed, highly resolved spatial databases are not always available for many urban areas. This is because these GIS databases can be expensive to develop and maintain; and/or may not be required for conventional stormwater management purposes. However, in our experience more and more municipalities in the U.S., at least, are developing and improving their spatial databases of stormwater infrastructure. To address this comment we would add the following content to section 2.2.1: "Existing databases that include the details for the stormwater infrastructure in this watershed are not always readily available to the modeler. In these cases, to adopt the subsequently described approach to GI scenario modeling in SWMM could require considerable ground-truthing and site surveying. In lieu of onsite visits, and as will become apparent from the descriptions below, what would be most important is determining the spatial location of storm sewer inlets. These are often visible from readily available aerial photographs. When elevation data for the storm sewer network is unavailable, much can be inferred using surface elevation data and assuming local construction codes for stormwater infrastructures, such as catch basin depths, and conveyance pipe diameters and slopes were applied. Such approximations would suffice for GI scenario analysis considerations, where storm sewer design is not the primary focus.
  - Please see P5/L14-21 of the revised manuscript where this information is included.

- Original Response: Also, the reviewer includes land cover and topographic information in his/her assessment on data availability. For land cover, the availability is somewhat irrelevant, as our approach requires land cover analysis and detailed digitization to do the subarea parameterization in SWMM that we describe in the manuscript (MS). This land cover and subsequent Subarea categorization is described in fairly specific detail in section 2.2.2. We struggled with how detailed the descriptions needed to be during the preparation of earlier drafts. Two internal reviewers, prior to submission to HESS, suggested the detail was too much and that the MS was too long. To address this issue we have prepared a companion report that will be published as a USEPA, Office of Research and Development contribution, that will be freely available to anyone interested. We will reference this report in the final version of the HESS MS, should it be accepted for publication.
  - We added more details as well as adding the USEPA report (Lee et al., 2017) as reference.

2. Relevant references are needed to support statements in the text, see specific comments for details. The key definitions (e.g., DCIA, ICIA, SPA, BPA) are given, but a conceptual model characterizing these key processes in a watershed and their spatial connections should be provided.

- ▷ Original Response: We will add a new figure that will depict a conceptual schematic that provides context to the DCIA, ICIA, SPA, and BPA categorization. This can be in the form of a side view of a home situated along a street with storm sewer infrastructure depicted.
  - Please see Fig. 5 of the revised manuscript. We significantly changed this figure to better explain how the different subareas could be represented in SWMM.
- Original Response: It is important to note, however, that these areas are defined within a subcatchment for SWMM modeling, not at the watershed scale. A watershed in SWMM consists of a number of subcatchments, which interact based on the existing storm collection system.
  - The areas (e.g., DCIA, ICIA, SPA, BPA) are defined within a SWMM subcatchment, not at the watershed scale. This clarification is made with the new use of the term hydrologic response element (HRE) coined to help distinguish physical landscape and modeled areas.

3. The land cover characterization in GIS is an essential step to provide inputs for hydrological evaluation in SWMM. Very limited information is given to understand how it is done in GIS analysis. Also readers need more details on how the four types of subareas are subsequently modeled in SWMM (e.g. parameter settings), e.g., how to parameterize BPA, ICIA, SPA for subcatchments.

- Original Response: As noted under 1. above, we struggled with the level of details to provide. Our intent for the MS was to focus on evaluating the performance of the approach to modeling GI in SWMM. For readers that want specific guidance on implementing the approach the USEPA report is being prepared. This can be referenced in the final version of the MS. It includes details on how to process clip, intersect, union, and manipulating attribute data in ArcGIS. Much of this will be familiar to users of ArcGIS, so we tried to strike a balance in the MS. If the Editor prefers a different tact to providing this information we could try to include as a supplemental section or appendix.
  - Please see P6/L2-11, P7/L8-16, and P9/L6-9 of the revised manuscript.
  - We referenced the companion USEPA (Lee et al., 2017) report that provides relevant details.

4. A better description of model calibration process is recommended, e.g., summary of parameters, inputs and outputs, criteria of performance.

Original Response: We disagree with this comment. We provide quite a bit of detail on our approach to calibration in section 2.5 and show results of sensitivity analysis in figure 10; a standard approach to model calibration, as well as giving initial and calibrated values of the sensitive input parameters in Table 1. What we failed to include was the initial and calibrated value for the width of BPA. This will be added to Table 1 in the final version of the MS. The output of calibration, along with performance statistics (i.e., NSE and R2) is provided in figure 11. So, we believe the level of detail is sufficient and actually contrary to what the reviewer suggests.

• Please see the addition to Section 2.6 and the new Fig. 6 of the revised manuscript. Hopefully this extra description and clarification

**Specific comments:**

1. P1, L21-24: it is confusing to mention the dimension and details of calibration parameters in the abstract before the relevant descriptions are provided.

- ▷ Original Response: This text in the abstract is not meant to note dimension and details of specific calibration parameters, rather to note a significant aspect about the approach to SWMM set-up that is presented in this study. What the text indicates in the abstract is that adopting the approach reduces the number of parameters that might be considered during calibration. However, while Reviewer 1 likely misunderstood the context for the description, Reviewer 2 correctly points out some inaccuracies in this statement (see general responses above and specific responses below) so we will eliminate it from the abstract
  - Please see P1/L21-23 of the revised manuscript. The text was removed and replaced with a more accurate statement to account for both Rev1's and Rev2's comments.

2. P2, L13-15: there are conflicting conclusions about the cost-effectiveness of GI, please provide references for your statements. In particular, the detention pond can be costly in terms of the construction and maintenance costs.

- ⇒ Original Response: That is why we put the word "may" in the sentence on L12. We will add the following to the sentence at L13: "..., like detention ponds, especially in cases where land is not available or very expensive."
  - Please see P2/L13-14 of the revised manuscript.

3. P2, L27-30: how the upstream area is discretized and the subcatchment are parameterized matter both in the modeling and calibration. Typical way to discretize subcatchments replies on GIS-based hydrological and landuse analyses to achieve reasonable characterization of natural drainage divisions. Any references to support your statements?

- ⇒ Original Response: We will try to include the presented criteria by the reviewer with relevant references, e.g., SWMM Reference Manual (Rossman and Huber, 2016).
  - Please see P2/L34-P3/L9 of the revised manuscript.

4. P3, L32: please define "a unit-area based analysis"

- Original Response: The term unit-area is a relatively common term in the field of stormwater modeling that refers to normalizing model output by using a common spatial dimension, e.g., in our case, 1 acre. We don't think it is warranted to define this relatively standard term in the introduction. Furthermore, in addition to the spatial dimension we define the land cover characteristics of the unitarea specifically for this study beginning on P10, L22.andat P11, L4. We will add to this sentence the word 'unit' to "hypothetical area" to help clarify.
  - We reconsidered this original response. The original term used was jargon and not adequate to describe an important piece of our approach to GI modeling. We have made significant changes in the revision as a result:
    - We changed "unit-area" to "hydrologic response element (HRE)" to minimize confusion.
    - Please check Section 2.4 and, Fig. 5 in the revised manuscript.
    - Please see P3/L8-15 of the revised manuscript.

- 5. Figure 1: No legend for background landuse
- ⇒ Original Response: Good catch. A relevant legend for the land use categorization will be added to the map.
  - Updated. Please see Fig. 1 of the revised manuscript.

6. P4, L15-20: A sketch of mentioned drainage system (manholes, pipes) is missing. Can author provide more information about the current drainage in the area? How many pipelines and manholes? what is the current service level of the system?

- Original Response: The existing drainage system is presented in Figure 5. A legend will be added to the figure to help define it. We don't see how statistics on number of pipes and manholes or 'current service level' of the system is relevant.
  - Please see Fig. 7 of the revised manuscript. Relevant legends were added in the figure.
  - Please see P18/L31-P19/L1 of the revised manuscript for a short description of the existing drainage system.

7. Section 2.2.2: Details are needed to understand the spatial analysis used in the study. what are the inputs and resolution? What types of GIS tools and processes are used to identify and digitize the 16 land covers? how do you estimate the future potential for GI implementation (e.g., to evaluate the potential of downspout disconnection for a main building) and which parameters are used?

- Original Response: We used 0.76 m LiDAR as noted in P6, L18. We felt the level of detail called for by the reviewer unwarranted for the specific purpose of this MS, which is to highlight the specific aspects and provide results of the analysis of performance of the approach developed for GI Analysis in SWMM. And as mentioned earlier, details on GIS analysis are included in the USEPA report that will be referenced in the final version of the MS, or if this is deemed insufficient, we can try to cut and paste relevant sections for addition to supplementary materials section or appendix. The referenced report includes how to process clip, intersect, union, and manipulating attribute data. For the third question; a systematic approach to 'estimate the future potential for GI implementation' would be quite difficult given uniqueness of place considerations, and is beyond the scope of this research.
  - We used 0.76 m LiDAR as noted in the manuscript (P5/L13, P7/L20).
  - We referenced the companion USEPA report (Lee et al., 2017) that provides relevant details (P6/L20-21).

8. P6, L1-L10: Though Figure 3 depicts the different boundaries of BPA, I still don't understand how to set the BPA in SWMM and which parameter do you use to represent BPA? how did you choose the buffer widths in this study? Can author provide more information on how to use the "intersect" tool for estimating the BPA and SPA?

- Original Response: The description of how to set-up the BPA in SWMM starts on P8, L31. We note that the original widths of the BPA are arbitrarily determined and explain why this has to be the case on P6. To provide more details on using the intersect and other functions in ArcGIS would require a step-by-step approach to using ArcGIS software. We, in fact, provide this detail in the USEPA report that is undergoing internal review and will be referenced in the MS, but we feel it is inappropriate for this MS to call for a tutorial on how to use certain functions in ArcGIS software.
  - Please see P7/L3-16 and P9/L6-9 of the revised manuscript. We added more description to help clarify.
  - We referenced the companion USEPA report (Lee et al., 2017) that provides relevant details.

9. P7, L15-16: Authors considered DS-IA and DS\_PA in subcatchments, could authors show how the two parameters are obtained? Is it a simple characterization of the dry ponds and detention areas in subcatchment?

- Original Response: DS stands for depression storage, as noted in the list of abbreviations. DS is a standard term used in urban hydrology that denotes the depth of water that can collect on urban surfaces, due to surface roughness properties. The initial value assigned to DS per land cover type was assigned based on recommendations or defaults described in the SWMM User's manual, as noted on P8, L19. DS has nothing to do with dry ponds or other built detention areas.
  - As noted in the manuscript, the initial value assigned to DS per land cover type was assigned based on recommendations or defaults described in the SWMM User's manual, as noted on P12/L11-14.

10. P7, L16-20: How did you choose the values for Scut and IMD? Can you provide more details on the division of IA into areas with or without DS? Also you mentioned several ways to route the internal flows, how do you model it in SWMM?

Original Response: We assign these values, in particular, using recommendations from the SWMM user manuals as noted on P8, L2: We will make note of this for the infiltration parameters earlier in this same section to help clarify. As for the other questions posed here, these can be answered for interested readers by consulting the SWMM user's manual documentations already referenced. We don't think addressing these questions with new additions to the text is warranted. It becomes more apparent with each comment that this reviewer has little experience using SWMM, we feel it is only necessary to go into the details of how to model urban hydrology using SWMM as they pertain to the described approach to GI scenario analysis. It is not our job to provide a tutorial on how to use SWMM. These are available at the SWMM download site, which will be referenced in the revision.
Please see P12/L14-20 of the revised manuscript.

11. Section 2.4.2: (a) vegetation swale (VS) seems an appropriate option to represent BPA, how the authors determined the parameters for VS, e.g., berm height, vegetation volume fraction? (b) how the authors determined the values of initial saturation and % of subcatchment imperviousness draining to the BPA from the geoprocessing steps? (c) I am confused about the way to model BPA, is it modeled as a VS (LID competent), or an individual catchment, or changes in subcatchment imperviousness and width? Why set the width (60 feet) for BPA?

- Original Response: We will try to clarify further in the revision, but generally we already state that parameter values are set based on guidance from the SWMM user manuals or from our experience working in urban areas. All of these details will be added to help clarify, including, berm height (0.1-in or 2.54-mm to minimize any storage effect within the berm, which is the case for real BPA), vegetation volume fraction (0, this is assumed to be negligible.), % imperviousness draining to the BPA (ICIA / TIA, where TIA = DCIA+ICIA). BPA is modeled as a VS (SWMM LID option) within a subcatchment, not as an individual subcatchment. We further acknowledge that many aspects of the BPA are unverifiable, and rationalize why this is not relevant to the integrity of the approach in section 2.3.2.
  - Please see P9/L6-9 and P12/L33-P13/L7 of the revised manuscript.

12. L9, L18-19: can authors give an detailed example on the evaluation of the groundwater flow in the study region? Is it calculated using Eq. 3 (then how the authors incorporated the equation in SWMM for groundwater simulation?) or just the difference between individual subcatchment surface and its nearest stream bottom?

- Original Response: No, we cannot provide more detail on groundwater flow. As mentioned in the manuscript, there was no observational data on groundwater flow. This is typically the case in urban modeling applications using SWMM. We will clarify that groundwater modeling parameters were defined using the SWMM Reference Manual and users' group knowledge base (e.g., https://www.openswmm.org/Topic/1465/groundwater-parameters; https://www.openswmm.org/Topic/4840/groundwater-values). The remaining questions in this response are irrelevant to our study.
  - Amended. Please see P13/L20-24 of the revised manuscript.

13. Figure 5: what is the difference between Figure 4 and 5? it seems that both figures mainly give the depiction of the subcatchments. Adding regional drainage network (manholes, pipelines) are recommended.

- ⇒ Original Response: Figure 4 is map of the watershed with relevant land cover and subcatchment delineation. Figure 5 is the conceptual representation of the area being model in the SWMM software, which includes the configuration of the storm sewer drainage network. As mentioned earlier, we will add a relevant legend to Figure 5, to help clarify.
  - Updated. Please see Figures 4 and 7 in the revised manuscript.

14. P10, L26- 28: Conceptual illustrations of the 6 options are well presented in Figure 6, but I find it difficult to understand how the 6 options are modeled in SWMM in details? for example, which subcatchment parameters are used to represent the different subareas (e.g. ICIA, TIA) and how to control the flow or routing directions?

- Original Response: As shown in the legend, each rectangular represents a subcatchment in SWMM, and the dotted line divides subareas within the subcatchment. A rectangle without a dotted line means the subcatchment consists of a single (homogeneous) subarea, either 100% impervious or pervious. The arrows represent flow routing directions. We will add these clarifications. The legend in the figure will be updated.
  - Amended. Please see Fig. 5 in the revised manuscript. It has undergone significant changes to help clarify.
  - Please see P9/L6-9 of the revised manuscript.

15. P11, L20-24: any reference to support your assumptions on the lengths for overland flows and surface slopes?

- Original Response: Length of overland flow means the flow length where the flow is maintained as overland flow (or sheet flow). It doesn't mean the physical length of a drainage area. This has long been a point of confusion in SWMM modeling. We will attempt to clarify in the revision. Surface slopes of typical urban drainage features are based on construction code or are inferred based on the GIS. The relevant references will be added.
  - Please see P9/L12-13 of the revised manuscript.
- 16. P12, L2: A brief explaining of the method is recommend.
- ⇒ Original Response: The following brief description will be added: "The 95th percentile rainfall event is defined as the measured precipitation depth accumulated over a 24-hour period for the period of record that ranks as the 95th percentile rainfall depth based on the range of all daily event occurrences during this period."
  - This brief description was added at P9/L30-32 of the revised manuscript.

17. P12, L7-10: one way to represent the GI can be the decrease of DCIA, which impacts the subcatchment imperviousness directly. That is one side of the problem, another is to attenuate the surface flow and slow down the speed. Is there any measure to model this aspect in your approach?

- Original Response: The effect of GI scenarios on the temporal dynamics of runoff is considered by comparing storm hydrographs before and after GI addition to the model. To address this comment we will add a note on the temporal changes to the storm hydrographs shown in Figure 13 to the results. Namely something like this: "It is interesting to note from Figure 13 that the peak flow for the event depicted in the figure is slightly higher in the GI Scenario, but that the duration of flows slightly smaller than this peak is longer in the baseline scenario."
  - Please see P15/L4-7 of the revised manuscript.
  - Please also see P19/L31-33 of the revised manuscript.

18. P13, Eq. 5-9: how to calculate the different Q values in SWMM? which result files are used to obtain these values?

- Original Response: This seems to be another question about SWMM modeling basics, to include the details of which are not appropriate for this MS. The reviewers question can be answered by consulting the SWMM user's documentation. If this question is based on the hydrograph separation procedure the calculations for the individual Q values are explained in the manuscript. Further action related to this comment is unwarranted.
  - As described in the manuscript, the different Qs were the modeling results from the different SWMM models.

19. P14, L16-18: I don't understand, if in option 4 where rainfall onto PA is completely captured by DS or infiltrated into soil, how come the simulated flow rates are much higher than the ones from the rest options?

- Original Response: The following description will be added to help explain the results observed for option 4: "Hydrologic connectivity is very important. In Option 4, the one-acre area is modeled as a single subcatchment with two subareas: IA and PA. Because this setup ignores the difference between DCIA and ICIA, the entire impervious area (subarea IA) is actually modeled the same as DCIA, which means all of the runoff is discharged to the storm drainage system directly with no abatement. Under a small storm (like
- We are not trying to argue that the approach to SWMM set-up is a 'better' representation of hydrologic response. While we do expect this to be the case for GI simulation, specifically, we have no way of actually testing this because we lack data on the effect of GI on hydrology post implementation. What we can say exactly about our approach is that it allows for a more realistic

expression of reality in the SWMM model set-up. This should make the model output more accurate by reducing overall model uncertainty, but again, because we have no way to directly test this assertion we will be sure to 'tone-down' such implications where they exist in the MS. We due compare the performance of our recommended subcatchment set-up approach to others in Figures 6 and 8 (**now fig. 5 and 9**). While Option 1 should be the most accurate among all options presented, we advocate option 6 for GI analysis in SWMM because options 1, 2, and 3 would result in many more subcatchments to parameterize, and more effort would have to be placed adjusting model setup to account for GI scenarios. Furthermore, Option 6 allows for subcatchment delineation based on topography and, therefore, has a physical meaning within the context of a watershed approach, while Options 1 through 3 would require disassociating the subcatchment context from reality to a more conceptual basis in the model. We will add this explanation to the MS.

- Our approach to calibration in SWMM is no different than what would be considered the more typical approach in terms of the actual modeling steps required, i.e., sensitivity analysis followed by one at a time adjustment, re-run, and compare simulated vs. observed. What is different is that we argue that the number of parameters that one might consider to adjust during calibration can be quite large if each subcatchment has unique values, or has been considered independently of all the other subcatchments, or as we describe it j x k number of parameters. We rely on the detailed spatial resolution of reality and the relatively small subcatchment size (driven by the storm sewer inlet, delineation requirement), to standardize parameter values across them. The Reviewer is correct to point out that in some watersheds spatial heterogeneity in topography and soils may nullify the assumption of commonality among all of them. If a land cover does not maintain the sufficient level of homogeneity within the target watershed for modeling, then we need to use more than one set of parameters for the land cover. In this case, we should divide the land cover into sub-groups that represent the heterogeneous hydrologic properties independently. We will add more details and discussion to address this valid concern.
  - Please check the amended Fig. 5 in the revised manuscript.
  - Please see P18/L7-9 of the revised manuscript.
  - Please find a new figure (Fig. 6) and related descriptions in the revised manuscript.

**Detailed Comments:**

1) P2 Line 16: Please provide references of relevant studies.

- ⇒ *References will be provided.*
  - Please see P2/L18-24 of the revised manuscript.

**2) P3 Line 32: Explain and provide references of unit-area based analysis.**

- Original Response: As mentioned in the manuscript (P10, L22), a unit-area is a hypothetical area, which represents a typical urban drainage area. SWMM can model a drainage area with various level of spatial aggregation, as shown in Figure 6. We arranged the unit-area based analysis demonstrate a "balanced" way for subcatchment characterization, based on the level of effort in model set-up and the accuracy in modeling results particularly for GI analysis.
  - We changed the term "unit-area" to "hydrologic response element (HRE)" to minimize confusion.
  - Please check Section 2.4 and, Fig. 5 in the revised manuscript.
  - Please see P3/L8-15 of the revised manuscript

**3) P5 Line 27: Add 'to' following 'adjacent'.**

• We added 'to'. (P6/L28 of the revised manuscript)

4) P6 Line 5-10: It is not clear to me how the 'intersect' tool was used to separate BPA and SPA. It is also unclear how the buffer widths (0.30, 0.61, and 1.52 m) were chosen.

- Original Response: We used ArcGIS to process the intersect analysis. As mentioned earlier, we feel it is inappropriate for this MS to call for a tutorial on how to use certain functions in ArcGIS. To provide more details on using the intersect and other functions in ArcGIS, we are preparing the USEPA report. The buffer width was selected when we calibrated the model. We arranged three SWMM models that represent three different sizes of BPA. We determined which one among the three cases of sizing BPA provided the more accurate simulation compared to the observed flow data. In this way, the BPA width was treated as a calibration parameter (see figure 10 (**now Fig. 11**)).
  - Please see P7/L13-16 of the revised manuscript.
  - We referenced the companion USEPA report (Lee et al., 2017) that provides relevant details.

5) P6 Line 18-22: How was 0.5 acre chosen? Why subcatchments of similar size help maintain hydrologic continuity?

- Original Response: Before conducting subcatchment delineation, we rather arbitrarily chose 0.5 acre to combine a drainage area with a neighboring subcatchment to minimize effort in model setup. In the actual analysis this happened only a few times. Maintaining similarity among subcatchment sizes confines the hydrologic loads received by the drainage system to a narrow range that helps to minimize errors in the simulation that might arise from surcharging or flooding due to mis-matched pipe network sizing. We can add this to the MS.
  - Please see P7/L23-28 of the revised manuscript.

6) P7 Section 2.4.1: 1) Move the description of calibration procedure from section 2.5 to here; 2) What are the values of Suct and IMD, and how were they initialized? Please also include them in Table 1; 3) Please provide how subarea routing was characterized within each subcatchment?

- Original Response: We don't think it makes better sense to discuss model calibration until all of the major aspects of model parametrization and set-up are attended to. Based on the soil type, the values for Suct were selected using the SWMM User's Manual. The actual IMD is dynamically updated at every modeling time step. As presented in P10, L1-2, the developed SWMM model for the study area was run for a six-month period (01 April 2009 to 31 August 2009) where the first four months of this period were used to stabilize the continuous simulation. While IMD was modeled using the default values in the EPA-SWMM, the IMD at the beginning of reporting the modeling results may not be affected (or minimally affected) by the initial values in model setup. We will provide more explanation of how subarea routing is configured in SWMM.
  - We think it makes better sense to keep the sections.
  - Please see P12/L14-20 of the revised manuscript.
  - Please see P9/L6-9 and Fig. 5 of the revised manuscript.

7) P8 Line 8-10: The authors stated that the initial values for "Length" were decided by averaging multiple field measurements of perceived overland flow lengths for each land cover type. How was overland flow length measured and generalized for each land cover that are spatially dispersed in the catchment? Plus, it is not reasonable to rescale the lumped flow lengths for each land cover to subcatchments with distinctive spatial connectivity to their respective outlet. The conventional SWMM approach is much more reasonable in this context.

▷ Original Response: We don't understand what the Reviewer means by "it is not reasonable to rescale the lumped flow lengths for each land cover to subcatchments with distinctive spatial connectivity to their respective outlet" We acknowledge that maintaining spatial homogeneity among subcatchments properties should be a priority in the application of our approach, and as discussed above will provide content to explain what to do in the spatial database set-up to account for this.

- ⇒ We can't be completely sure of what the Reviewer means by 'conventional approach". Nonetheless, we did not intend to imply that our approach is 'better', generally. However, a common approach to subcatchment set-up is studied in Options 4 or 5 of figure 6 (**now fig. 5**). Using either of these makes the application of GI an implicit consideration. We will add this explanation and clarification to the MS. In this study, we intended to examine an alternative for characterizing a drainage area.
  - Please see P11/L18-30 of the revised manuscript.

8) P8 Section2.4.2: I do not understand how BPA and SPA was represented and spatially connected in SWMM. Based on the description, BPA was modeled as an LID component that receives flow from ICIA of subcatchment(s)? Looking at Figure 3&4, however, BPA seems to be lumped into a subcatchment. Why choosing the buffer width of 18.3m? Please clarify.

- Original Response: We will clarify as follows: In SWMM, BPA is modeled as vegetated swale. The size of BPA can be defined for each subcatchment. The % contributing impervious area to the BPA can be also defined for each subcatchment, which is ICIA/TIA where TIA=DCIA+ICIA. Since the total pervious area (TPA) remains identical for each subcatchment, the sizes of SPA for individual subcatchments can be decided as SPA = TPA BPA for the three different sizes of BPA (which were derived by applying three different distances for proximity analysis in GIS). When we calibrated the model, we checked which one, among the three cases of sizing BPA, would calibrate the best for various storm sizes. More clarification will be added to the figures also.
  - Please see P9/L6-9, P7/L13-16, and P12/L29-P13/L7 of the revised manuscript.

9) P10 Section 2.5: It is common in both spatially distributed and lumped hydrologic modeling that the land cover- and soil-specific parameters are fixed across the catchment. However, it is inappropriate to aggregate and calibrate by land cover the parameters of slope and overland flow length that are much more topography than land cover dependent. I can't agree with author's argument that this calibration approach is efficient or can be transferrable to other systems.

- Original Response: The Reviewer is correct to point this out, and we will qualify our statements about transferability accordingly. If a land cover does not maintain the sufficient level of homogeneity across the target watershed we need to use more than one set of parameters for the land cover. In this case, we should divide the land cover into sub-groups that represent the heterogeneous hydrologic properties independently. As noted above we will explain the relevance of this issue and provide a remedy for it in the MS. While we agree that it is more accurate to select values for slope and overland flow length based on topography, generally speaking, SWMM subcatchment areas as defined by modelers tend to be somewhat topographically homogeneous, otherwise accurate model representation is difficult. Also, urban sewer collection systems are zoned in a manner accounting for local variation in topography. Even land use generally follows topography, therefore, land cover is a reasonable surrogate.
  - Please see P14/L19-26 of the revised manuscript.

10) P10 Section 2.6: 1) If I understand it correctly, SWMM was calibrated using the option 6 setup. The calibrated parameters include overland flow lengths and slopes as in Table 1. In this section, the authors provide new sets of flow lengths and slope parameters for different cover type, which are different from the values given in Table 1. Did all 6 options use the same parameterization or not? If yes, why not using the calibrated parameters? If no, the comparisons do not seem fair – calibrated option 6 vs. non-calibration options.

▷ Original Response: Table 1 (now Table 2) shows the initial and calibrated parameters for the study area, not the hypothetical unit-area analysis. Therefore, there seems to be some confusion here, so we will attempt to clarify further in the MS. For the hypothetical area analysis, all of the 6 options

were arranged using the same spatial and hydrologic characteristics as presented in P11, L23-30. However, the ways to model DCIA, ICIA, BPA, and SPA are different among the options. For the hypothetical unit area analysis calibration was not necessary.

- The table (Table 2 in the revised manuscript) shows the initial and calibrated parameters for the study area, not the hypothetical unit-area analysis (HRE in the revised manuscript).
- We added more descriptions to clarify. Please see the amended Fig. 5 and P9/L19-21 of the revised manuscript

11) P14 Line 16-18: Why option 4 has the highest peak flow (in Figure 8a) if only DCIA discharges runoff?
⇒ Original Response: In option 4, the entire impervious area is modeled as DCIA, i.e., ICIA is also modeled as DCIA There is no run on from impervious to pervious areas in option 4. (Please also see the responses under the comment #19 for Reviewer 1).

• Please see P17/L14-20 of the revised manuscript.

Figure 2: I suggest that the authors label the ID and show the baseline flow path of each surface record so the readers can better understand the difference between DCIA and ICA.

- ⇒ Original Response: The figure will be amended. The attribute table shown in the figure will also be amended to minimize any miss-interpretation.
  - Amended. Please see Fig. 2 of the revised manuscript.

---

## Referee Report (RR1)

**Summary:**

This paper leverages a highly resolved database of urban land cover to evaluate how effectively different SWMM model set-ups capture the hydrologic processes of urban watersheds with GI at a small (I'll call 'hillslope') scale. Once the ideal model set up (ideal in that balances complexity with *perceived* accuracy) is identified, this model set up is used to simulate flows at the watershed scale. The watershed scale simulations are done to evaluate parameter sensitivity and compare source area contributions to hydrographs generated by GI vs. non-GI scenarios.

**General comments:**

The research questions posed are certainly worth answering. The data used, along with the author's demonstrated modeling skill, is suitable for answering the questions. Specifically, the author's attempts to improve representation of "reality" (i.e., an HRE) of urban surface connectivity to GI within the SWMM framework (i.e., a subwatershed) is valuable. However, the paper is often bogged down by an extensive discussion of the methods used, relative to an exploration of the results generated. Also, conclusions are often made without a reference to a specific metric of evaluation.

Study Limitations

There are some considerable limitations to the methods used in this manuscript. Specifically:

1. The 6 model set ups are evaluated at the hillslope scale using a design storm approach. Model "accuracy" (as the authors refer to it) seems to be based solely the author's expectations of physical processes in the watersheds. While I'm sure the authors have an excellent understanding of these processes in a historically well-studied watershed - using observed flows to validate the model set-ups would be a far better approach.

   If there is some other way that model accuracy is assessed (i.e., beyond comparing the output to expected behavior) in this portion of the manuscript (Sections 3.2) it needs to be stated explicitly. Are results compared relative to model set-up "Option 1" because this modeling framework is most complicated? Are they compared to the mean? Also - the "MR5subs" metric is not defined. Nor is "Variation from 5 Subs" on the y-axis in Figure 10. Is this how accuracy is assessed? Whatever assumptions are made to compare, they must be stated explicitly (and justified)

   To that end: I would not suggest comparing model output to other model output. I would suggest comparing the 6 model set-up options at the **watershed scale** - and see which set-up is best able to represent the **observed flows** in the July 2009 period with and without calibration. Then, the can truly be assessed, and there is potential for a discussion of the complexities associated with the various model set ups and calibrations.

   Alternatively, the authors could re-frame this entire section of the manuscript as "critical evaluation of model physics" or something – an avoid using the term accuracy. However, the assumptions that go into the "critical evaluation" need to be explicit and justified with literature and/or specific observations at the site.

2. With regard to the hydrograph separation method: separating the sources of runoff is definitely valuable, and the way the authors do this is clever. However, is SWMM not able to output these variables explicitly? It is a physical model. Why do the authors need to back-calculate these fluxes by adding/subtracting different model scenario output? If either SWMM cannot output these variables directly or if it is too complicated to aggregate the output from all the subwatersheds at the watershed scale: a justification for the "back-calculation" approach should be provided in the introduction.

3. I do not see the value in "calibrating" – or even restricting - BPA width. The actual BPA will change with storm size (as the authors point out). Why not just leave the BPA as the total pervious area – which is a better representation of the potential to infiltrate run-on. Then, just change the connectivity to the pervious area in the modelling scenarios and allow the model's infiltration routine calculate how much water is infiltrated vs. how much runs off the surface? This is more realistic. Some specific comments on this can be found in the next section of the review. Additionally, the specific metrics by which different BPA widths were compared is not explicitly mentioned in the text.

Manuscript Organization

I also have a series of suggestion regarding the organization of the manuscript. The manuscript, as it is, is very, very long. It includes 14 figures. I'm aware of HESS's page limitation, but it seems like this manuscript could be streamlined. Some suggestion I have for removing/reorganizing content are:

1. Great length is spent regarding the watershed scale model parameterization and set up (i.e. Fig 6 and its paragraphs of discussion). It seems to me like much of the detailed methods discussed are standard practices in SWMM. I would suggest the authors determine which of these steps is novel vs. which are just typical urban hydrologic modeling practices, and focus on just writing up the novel components. Direct the reader to the SWMM user manual, or assume they have some basic understanding of hydrologic modelling. I have some specific suggestions in the "Specific Comments" section of this review.

2. The SWMM model and its components (i.e., parameters, use of subwatersheds, etc. – Section 2.5) is introduced AFTER these components have been discussed extensively during the description of the 6 model set-ups (Section 2.4). Perhaps moving this ahead of the set-ups would be a better organizational set up, and allow for the removal of duplicate text from the current Section 2.4.

3. The ratio of methods to results in this paper is extremely large. I understand that the steps presented in the manuscript are meant to be used as a model to future SWMM users modelling GI, but the introduction frames the manuscript are purely evaluating SWMM model set up for evaluating GI. Perhaps some text in the introduction that describes how this manuscript is really a suggested method for modelling GI in SWMM - with some critical evaluation - would be more suitable for the actual contents of the manuscript.

4. Reference the Lee, 2017 report when describing the watershed's land cover instead of documenting it in this manuscript

5. It is often unclear which spatial scale (i.e., hillslope vs. watershed) specific parts of the manuscript are describing. Clarifications should be added in almost all of the manuscript's sub-sections.

**Specific comments:**

A note before beginning – continuous line numbers would make completing a future review of this manuscript much easier

Abstract
- Pg 1, Ln 17-21: Add clarification on spatial scale: design storms = hillslope; continuous = watersheds

Introduction
- Pg 2, Ln 9-10: What about infiltration as an objective of GI?
- Pg 2, Ln 12: Each installation is relatively less expensive, but also has relative less impact. Should add some detail as to what costs your describing while making this claim
- Pg 3, Ln 12: "modeld" >> "modeled"
- Pg 3, Ln 28: "After the HRE delineation is performed in SWMM, each one undergoes" >> Clarify sentence structure so "one" is clear
- Pg 3, Ln 34: Clarify "conventional objectives"

Methods and Material
- Pg 5, Ln 28: What is a stormwater detention area vs. a dry/wet pond?
- Pg 6, Figure 2: Possibly add some of the shapefile IDs to the map? Generally, though, I'm not sure this figure is needed. The text adequately describes the spatial database.
- Pg 7, Ln 5-10: BPA is variable based on storm size (as the authors point out) – so why try limit yourself to one BPA width? Why not just allow for the entire pervious area to potentially infiltrate? As would be the case in real life. If the run-on is greater than the infiltration capacity at any time, the infiltration model will handle this appropriate and generate surface runoff.
- Ph 7, Ln 11: Where do these widths come from?
- Pg 7, Figure 3: While it is certainly attractive, this figure is not that valuable (especially considering there are so many in the manuscript already). I think the readers can conceptualize what a 0.3m buffer looks like without the visual aid.
- Pg 7, Figure 4: Consider adding the inlets to this map
- Pg 8, Figure 5: For the subcatchment boxes: do the sub-subcatchment areas (i.e., "TPA" and "TIA" in Option-4) run in series or in parallel? There is discussion of this (somewhat) in the text when discussing the SWMM model connectivity scenarios – but including a set of arrows and/or arranging the sub-subcatchment areas vertically when connected in parallel and horizontally when in series would be of great value to this figure

- Pg 9, Ln 33: This could be my own lack of understanding – but I'm unclear on what the "cumulative" precipitation values represent. Either way - I'm not sure it adds considerable value to include this in the text and in Table 1. Percentile is likely sufficient.
- Pg 10: As mentioned in the "General Comments" – the SWMM model and its components/parameters should probably be introduced prior to the discussion of the 6 model set-ups
- Pg 10: This subwatershed set up was done for the entire watershed – may want to clarify this at some point. Same with the discussion of parameterization on Page 11
- Pg 10, Ln 30: Why bother introducing the 16 land cover types at all? Just introduce the 10 the first time, and remove this later clarification.
- Pg 11, Figure 6: This figure seems too fundamental to the process of modelling to warrant inclusion. As mentioned: unless the paper is reframed as a "new method" for running SWMM with GI – I would remove this. I would actually advocate for this framing because it aligns better with the content (namely, all the nitty-gritty set up and parameterization details included). However, I'm not sure HESS would be the place for such a manuscript
- Pg 11, Ln 15-32 and Pg 12, Ln 1-24: There is far too much SWMM set up 101 in this section. Is it possible to reduce this to a table of parameter values? And just refer the user to the SWMM manual when describing what each parameter is and how it was determined?
- Pg 12, Ln 31: Hillslope or watershed scale?
- Pg 13, Ln 7: Here is evidence that one BPA width is probably not the best way to model this. Again – I'd advocate allowing all of the BPA to infiltrate water, and let the model physics determine how much of that water makes it to the subsurface – this seems more representative of reality
- Pg 13, Ln 9-15: Refer to SWMM manual
- Pg 13, Ln 27: PCSWMM should be mentioned earlier in the manuscript if it is a critical step to the model set up
- Pg 13, Figure 7: Is it possible to combine Figure 4 and 7?
- Pg 14, Ln 9: BPA is not in Figure 6
- Ph 15, Ln 1: What is the spatial scale these simulations were done at? And what model set-up was used (out of the 6 tested)?
- Pg 16, Ln1-10: See "General Comment" on including justification for back-calculating these values vs. taking model output directly. A good place to do with would be in the last paragraph of the Introduction on page 4

Results and Discussion
- Pg 16, Ln 19ff: This not a result, it is a description of the watershed. Aren't all of these details in an EPA report somewhere (e.g., Lee, 2017)?
- Pg 16, Ln 22: What scale was the BPA calibration performed at? If watershed scale, then I would assume the July 2009 time period was used. What size storm events occurred in this period? If they're all small – I would expect an underestimation of the BPA width. If it occurred at the hillslope scale – what design storms were used?
- Page 16, Ln 22: These results do not mean "that the runoff from ICIA is discharged

to the adjacent 0.61 m of pervious area" – soften the language. E.g.: "a 0.61m buffer width best mimicked hydrologic behavior, evaluated by metric XYZ"

- o Note: what was used to compare these runoff widths? Total runoff? Peak flow? NSE? Include this, it seems important.
- Pg 17, Ln 1: What spatial scale did this occur at?
- Pg 17-18: See General Comment #1 on the use of the word "accuracy" in this section 3.2
- Pg 18, Ln 17: Define MR5subs – it is important
- Pg 19, Ln 3-4: "Fig 10" >> "Fig 11"
- Pg 19, Ln 1-10: 3% change in total runoff doesn't seem like a lot of sensitivity. This should be noted. How much did total runoff change during calibration?
- Pg 19, Ln 13-24: Move calibration discussion of calibration ahead of discussion of sensitivity analysis. Alternatively, combine and condense these paragraphs.
- Pg 20, Ln 12: "Fig. 12b and 13" >> "Fig 13b and 14"

Conclusions
- Pg 21, Ln 3: "accuracy" is not the best word

---

## Author Response (AR2)

**Preamble to the Editor:**

The authors want to make note of the fact that the manner in which the review of our manuscript has been handled HESS is not well aligned with what we have experienced in numerous other contributions to the peer-reviewed literature over our careers.

Our original paper received two reviews that we provided detail response to and that resulted in major changes to the manuscript. Apparently, the Editor could not get these reviewers to comment on the revision, so instead asked two new reviewers to comment. The result was two additional reviews with nearly opposite criticisms to the original reviews. The original reviewers called for more explanation of methods. These new reviewers call for a deletion of all the added explanation. We tried to again strike a balance in the content presented in the new version.

The bottom line is that our study presents "modeling methods development" research. Our focus is Green Infrastructure simulation using the commonly-used SWMM model. In the United States, SWMM is used to make billion-dollar decisions about urban drainage design, particularly in Cities that are under Consent Decree over combined sewer overflow violations. SWMM modelers need a valid approach to GI simulation at larger urban watershed scales. There is very little reporting or direct research on this issue to guide SWMM users toward good modeling practices for GI simulation.

SWMM users that adopt the approach that we develop can use the HESS article as a reference to support its strength for balancing model complexity and output uncertainty. We think this will be the primary utility of the paper over the long term. As such, we wanted to provide enough detail for users to adopt the approach while presenting the results of unique and novel analyses that we conducted to validate the approach in terms of it over all utility at GI simulation for planning purposes. The bulk of the paper was intentionally focused on describing the "why" and "how" of our approach to GI simulation in SWMM. The results/discussion section has been structured to 1) validate the approach that we developed in terms of balancing model complexity and output uncertainty (accuracy) and 2) demonstrate the utility of adopting the approach to watershed-scale GI effectiveness considerations.

We have gone through this second round of completely new responses and made adjustments. These reviews were in some ways more critical of the manuscript than the first. At this point we request that the Editor makes a decision about this manuscript. We will make adjustments as needed for a final version, but the current feedback we have received represents a dichotomy that is difficult to know how to handle.

Sincerely,
The authors.

**1st Reviewer:**

**Summary:**
This paper leverages a highly resolved database of urban land cover to evaluate how effectively different SWMM model set-ups capture the hydrologic processes of urban watersheds with GI at a small (I'll call 'hillslope') scale. Once the ideal model set up (ideal in that balances complexity with *perceived* accuracy) is identified, this model set up is used to simulate flows at the watershed scale. The watershed scale simulations are done to evaluate parameter sensitivity and compare source area contributions to hydrographs generated by GI vs. non-GI scenarios.

**General comments:**
The research questions posed are certainly worth answering. The data used, along with the author's demonstrated modeling skill, is suitable for answering the questions. Specifically, the author's attempts to improve representation of "reality" (i.e., an HRE) of urban surface connectivity to GI within the SWMM framework (i.e., a subwatershed) is valuable. However, the paper is often bogged down by an extensive discussion of the methods used, relative to an exploration of the results generated. Also, conclusions are often made without a reference to a specific metric of evaluation.

**Authors' general responses:**

⇨ The reviewer had a good grasp of the major themes of our paper as we wanted them to be interpreted. We appreciate the consideration he/she gave to the material. Our general responses to 1) the criticism about extensive methods and 2) conclusions without a reference to a metric for evaluation are as follows:

1) We have moved several paragraphs and 2 sections from the methods section to an appendix, but in our defense in the first round of reviews of this paper, both reviewers asked for more description of the methods. In our revision we tried to include all of the specific details that those reviewers asked for, while also noting that we have published an EPA Report as a "user's guide" that provides methodological detail as well. We moved 2.5 pages of content to a new Appendix. Hopefully that satisfys the opposing criticisims about methodological content that we have received from 4 different reviewers.

2) We think this comments stems from some confusion over the difference between the two primary analyses that we conducted and present in the paper to evaluate GI modeling approaches is SWMM: A) We used a hypothetical approach to assess differences in subcatchment parameterization in SWMM and validate the approach we developed for spatial discretization and parameterizing SWMM for GI modeling. B) We then used the validated approach to subcatchment characterization to parameterize a SWMM model at the watershed scale, and use that model to demonstrate how the effects of GI scenarios can be incorporated and evaluated using the SWMM output at this scale. We decided early on that it would be inappropriate to use the watershed scale and the observed flow data available for this scale to gauge the differences among

the discretization approaches because of potentially confounding effects introduced when the drainage network and groundwater algorithm are included to run the simulation. We now explain this reasoning and try to clarify differences between the analysis in this revision (see first paragraph of section 2.5).

3) We think that if a reader is clear about the relevance of the hypothetical analysis then there should not be any confusion about the metric for evaluating what is approach we determined to be best. It should be clear from Figures 8 and 9 are two of the primary data figures in the paper. Both were meant to depict the relative differences that the approaches to subcatchment parameterization have on SWMM output. We discuss both in detail and we clarify that our primary means of assessing the performance of each option considered is based relative to what we assume is the approach that would provide the least output uncertainty, because it is the most spatially explicit one. But this approach is impractical for SWMM modeling at a watershed scale, so we base our assessment of the 'best' approach among the 5 others considered relative to this most spatially explicit one with the criteria of balancing complexity in model set-up with accuracy. It can be seen in Figure 9. That Option 6 is nearly identical in terms of average flow rate, peak flow rate, and total runoff volume as the most spatially explicit one (Option 1). So is option 2 and 3, but those are harder to implement during parameterization of SWMM. Option 6 is the best. This is explained in detail in the discussion of figures 8 and 9. We make this clear in the 2$^{nd}$ sentence of the conclusion section.

Study Limitations

There are some considerable limitations to the methods used in this manuscript. Specifically:

1. The 6 model set ups are evaluated at the hillslope scale using a design storm approach. Model "accuracy" (as the authors refer to it) seems to be based solely the author's expectations of physical processes in the watersheds. While I'm sure the authors have an excellent understanding of these processes in a historically well-studied watershed - using observed flows to validate the model set-ups would be a far better approach.

⇨ Response: We disagree with the reviewer on this point. We decided early on that it was inappropriate to use the watershed scale and the observed flow data available for this scale to gauge the differences among the discretization approaches because of potentially confounding effects introduced when the drainage network and groundwater algorithm are included to run the simulation. It is most rational to conduct the evaluation at the hillslope scale (as the reviewer calls it), because this is the scale that defines the HRE, i.e., the land area that drains to a storm sewer inlet. We felt it more rational to assume that the most spatially explicit characterization of this area (option 1) would be the most accurate, or have the lowest output uncertainty, and provide references to support that assumption. However, we acknowledge that it would be best to have supporting observational data at this scale to prove this assumption, but obtaining flow data at the point of entry to a storm sewer inlet can be very difficult in practice.

If there is some other way that model accuracy is assessed (i.e., beyond comparing the output to expected behavior) in this portion of the manuscript (Sections 3.2) it needs to be stated explicitly. Are results compared relative to model set-up "Option 1" because this modeling framework is most complicated? Are they compared to the mean? Also - the "MR5subs" metric is not defined. Nor is "Variation from 5 Subs" on the y-axis in Figure 10. Is this how accuracy is assessed? Whatever assumptions are made to compare, they must be stated explicitly (and justified)

⇨ Response: Based on the above. Our approach to characterizing the accuracy of the subcatchment parameterization in SWMM using the hypothetical analysis is rational.

To that end: I would not suggest comparing model output to other model output. I would suggest comparing the 6 model set-up options at the **watershed scale** - and see which set-up is best able to represent the **observed flows** in the July 2009 period with and without calibration. Then, the can truly be assessed, and there is potential for a discussion of the complexities associated with the various model set ups and calibrations. Alternatively, the authors could re-frame this entire section of the manuscript as "critical evaluation of model physics" or something – an avoid using the term accuracy. However, the assumptions that go into the "critical evaluation" need to be explicit and justified with literature and/or specific observations at the site.

⇨ Response: We appreciate the reviewer giving this issue considerable thought, but as noted above it is more rational to conduct the HRE analysis using the smaller hypothetical area. Just because we have observed flow data for the watershed scale does not mean that it makes best sense to use that data to qualify our modeling approach to parameterizing SWMM subcatchments at the HRE scale.
⇨ We think if the reader understands the reasoning behind why we conducted the hypothetical analysis at the scale we did and the chosen evaluation criteria then there should be no concern with how we framed the results/discussion or how we characterize the relevance of the SWMM output. In fact, with any of the 6 options we studied, we could make the watershed level model output align closely with the observed data during the calibration process, but with the option we suggest this calibration effort will be reduced and the amount of model input and output data will be considerably lower.
⇨ Finally, it is noteworthy that none of the other Reviewers raised this concern.

2. With regard to the hydrograph separation method: separating the sources of runoff is definitely valuable, and the way the authors do this is clever. However, is SWMM not able to output these variables explicitly? It is a physical model. Why do the authors need to back-calculate these fluxes by adding/subtracting different model scenario output? If either SWMM cannot output these variables directly or if it is too complicated to aggregate the output from all the subwatersheds at the watershed scale: a justification for the "back-calculation" approach should be provided in the introduction.

⇨ Response: We appreciate the Reviewer's recognition of the cleverness of our approach to hydrograph separation. While, SWMM provides some (e.g., system-wide surface runoff and

groundwater flow) not all the source flow paths that we have presented, there is no provision in SWMM for obtaining the information as the reviewer suggests in the current framework of the model.

⇨ As described in the manuscript (after Figure 11 in page 19), the relative contribution of the primary hydrologic components with and without GI implementation can be examined by this hydrograph separation approach, which can provide valuable insights for GI planning.

⇨ Hydrograph separation is mentioned in the last paragraph of the Introduction. We think it would be confusing to the reader to add a justification for the hydrograph separation approach before we actually explain how it works. We added a sentence at the end of the Introduction to note that there is currently no provision in SWMM for accomplishing this.

3. I do not see the value in "calibrating" – or even restricting - BPA width. The actual BPA will change with storm size (as the authors point out). Why not just leave the BPA as the total pervious area – which is a better representation of the potential to infiltrate run-on. Then, just change the connectivity to the pervious area in the modelling scenarios and allow the model's infiltration routine calculate how much water is infiltrated vs. how much runs off the surface? This is more realistic. Some specific comments on this can be found in the next section of the review. Additionally, the specific metrics by which different BPA widths were compared is not explicitly mentioned in the text.

⇨ Response: We don't understand what the Reviewer is talking about. BPA width actually represents the BPA size. As described in Sections 2.2.3 and 3.1, we estimated three choices of potential BPA with 0.30, 0.61, and 1.52 m buffering widths. The three choices of BPA were compared during the calibration. The results are presented in Section 3.1.

⇨ As we argued and proved in this study, that the total pervious area (TPA) should not be treated the same as BPA. Please check out the simulation results from Options 4 and 5 in Figures 8 and 9, which treat TPA as BPA, compared to the other options that clearly identify the BPA, especially when smaller storms were simulated which are key flow capture scenarios for GI implementations.

⇨ As described in Section 2.2.3, the entire surface of TPA does not receive runoff from upgradient impervious areas, in general. Actually, this is one of the main areas that we have examined in this study: The relevance of explicitly modeling BPA vs. TPA.

Manuscript Organization

I also have a series of suggestion regarding the organization of the manuscript. The manuscript, as it is, is very, very long. It includes 14 figures. I'm aware of HESS's page limitation, but it seems like this manuscript could be streamlined. Some suggestion I have for removing/reorganizing content are:

⇨ Response: Agreed, but much of the detail was added at the request of earlier reviewers. Clearly is difficult to strike a balance that will make all readers happy. Have relocated some of the details on model set-up that are independent of the new approaches we present here to an Appendix.

1. Great length is spent regarding the watershed scale model parameterization and set up (i.e. Fig 6 and its paragraphs of discussion). It seems to me like much of the detailed methods discussed are standard practices in SWMM. I would suggest the authors determine which of these steps is novel vs. which are just typical urban hydrologic modeling practices, and focus on just writing up the novel components. Direct the reader to the SWMM user manual, or assume they have some basic understanding of hydrologic modelling. I have some specific suggestions in the "Specific Comments" section of this review.

⇨ Response: Agreed, we kept the novel aspects in, and moved the other more 'standard' SWMM set-up details for the study watershed to an Appendix.

2. The SWMM model and its components (i.e., parameters, use of subwatersheds, etc. – Section 2.5) is introduced AFTER these components have been discussed extensively during the description of the 6 model set-ups (Section 2.4). Perhaps moving this ahead of the set-ups would be a better organizational set up, and allow for the removal of duplicate text from the current Section 2.4.

⇨ Response: Agreed. The order of Sections 2.4 and 2.5 are switched in the revised manuscript.

3. The ratio of methods to results in this paper is extremely large. I understand that the steps presented in the manuscript are meant to be used as a model to future SWMM users modelling GI, but the introduction frames the manuscript are purely evaluating SWMM model set up for evaluating GI. Perhaps some text in the introduction that describes how this manuscript is really a suggested method for modelling GI in SWMM - with some critical evaluation - would be more suitable for the actual contents of the manuscript.

⇨ Response: We added the suggested qualifier to the introduction: 2nd sentence of the last paragraph.

4. Reference the Lee, 2017 report when describing the watershed's land cover instead of documenting it in this manuscript

⇨ Response: As a technical paper, we tried to minimize any duplications or too detailed contents.
⇨ We have referenced the Lee et al. (2017) report.
⇨ We also re-located some of the details to Appendix, which are not completely presented in the referenced report.

5. It is often unclear which spatial scale (i.e., hillslope vs. watershed) specific parts of the manuscript are describing. Clarifications should be added in almost all of the manuscript's sub-sections.

⇨ Response: We're surprised by the confusion. HRE was defined earlier on as our 'small scale' unit of consideration. If this is understood, then whenever we write about HRE the scale of attention would be known. We never used or intended to use 'hillslope vs. watershed' scales for differentiating spatial scale in this study. "An HRE is the drainage area (i.e., a real

spatial element of the landscape being modeled) where GI practices may be implemented to control the element's surface runoff prior to discharge to the stormwater collection system." A watershed can consist of a single HRE (which would be very rare) to numerous HREs.

**Specific comments:**

A note before beginning – continuous line numbers would make completing a future review of this manuscript much easier

⇨ Response: The manuscript was prepared using the HESS-provided MS-Word template, including the way of presenting line numbers.

Abstract

- Pg 1, Ln 17-21: Add clarification on spatial scale: design storms = hillslope; continuous = watersheds
- ⇨ Response: Amended.

Introduction

- Pg 2, Ln 9-10: What about infiltration as an objective of GI?
- ⇨ Amended to include infiltration.

- Pg 2, Ln 12: Each installation is relatively less expensive, but also has relative less impact. Should add some detail as to what costs your describing while making this claim
- ⇨ Response: Some clarification was added.
- ⇨ Cost-effectiveness of GI application depends on various site-specific conditions. Thus, we used conditional sentences and added "may" in the sentence. We tried to provide a kind of GI potential, rather than a concrete general fact, as we responded to the previous revision. Because this is not so critical part of our study, we can remove these sentences especially if it creates any mis-conceptions or confusion.

- Pg 3, Ln 12: "modeld" >> "modeled"
- ⇨ Response: Corrected.

- Pg 3, Ln 28: "After the HRE delineation is performed in SWMM, each one undergoes" >> Clarify sentence structure so "one" is clear
- ⇨ Response: Clarified.

- Pg 3, Ln 34: Clarify "conventional objectives"
- ⇨ Response: Clarified with fundamental examples.

Methods and Material

- Pg 5, Ln 28: What is a stormwater detention area vs. a dry/wet pond?

⇨ Response: In this watershed the stormwater detention area was created by adding a culvert with an orifice control structure into the existing stream channel to satisfy the 10yr detention requirement. It does not really fit the definition of a dry or wet pond.

- Pg 6, Figure 2: Possibly add some of the shapefile IDs to the map? Generally, though, I'm not sure this figure is needed. The text adequately describes the spatial database.
⇨ Response: We believe some of the readers will benefit from the visual for the spatial representation and the organization of the attribute data.

- Pg 7, Ln 5-10: BPA is variable based on storm size (as the authors point out) – so why try limit yourself to one BPA width? Why not just allow for the entire pervious area to potentially infiltrate? As would be the case in real life. If the run-on is greater than the infiltration capacity at any time, the infiltration model will handle this appropriate and generate surface runoff.
⇨ Response: The entire pervious area can infiltrate. The BPA defines a subset of that pervious area that receives runoff from ICIA. Not all the PA receives the runoff. This is the critical distinction we make and show in the paper why it is important.
⇨ As mentioned earlier, the impervious runoff from the upgradient area physically does not get evenly distributed to the entire surface of TPA. Only a small connected often downgradient part of the pervious area can receive impervious runoff and work like a buffering strip or swale. If we model TPA as BPA, we may end up either over-estimating the onsite infiltration of the impervious runoff or under-estimating the infiltration rate compared to the actual value (depending upon TPA to BPA ratio). This is all explained in the discussion of the results of the hypothetical analysis (Figs 8 and 9).

- Ph 7, Ln 11: Where do these widths come from?
⇨ Response: As mentioned in the manuscript, we arbitrary selected these numbers to use for the calibration. We believe this range of values will work for most urban/suburban areas where GI's are often implemented. In case of where the three suggested options do not work, one can apply linear interpolation/extrapolation between the width and the size of BPA; or conduct additional buffering analysis using ArcGIS to derive more choices for defining and refining BPA widths.

- Pg 7, Figure 3: While it is certainly attractive, this figure is not that valuable (especially considering there are so many in the manuscript already). I think the readers can conceptualize what a 0.3m buffer looks like without the visual aid.
⇨ Response: We disagree. The figure gives a visualization of how the different BPA lengths translate to space and also provide context for the extent of BPA around all of the ICIA.

- Pg 7, Figure 4: Consider adding the inlets to this map
⇨ Response: While adding the inlets would be valuable, we think it would make the figure look too cluttered. The inlets are presented in Figure 7.

- Pg 8, Figure 5: For the subcatchment boxes: do the sub-subcatchment areas (i.e., "TPA" and "TIA" in Option-4) run in series or in parallel? There is discussion of this (somewhat) in the text when discussing the SWMM model connectivity scenarios – but including a set of arrows and/or arranging the sub-subcatchment areas vertically when connected in parallel and horizontally when in series would be of great value to this figure

⇨ Response: We tried to improve this figure to minimize any mis-conceptions and provide better understanding based on the first set of reviews. To describe the internal flow directions, we have added the physical and conceptual representations of HRE including the directions of flow pathways. We think this figure is a reasonable and balanced representation. Furthermore, we write in Section 2.5 "Option 4 configures the single subcatchment with only two subareas, impervious and pervious. The runoff from pervious area discharges through impervious area (i.e., TIA = DCIA)."

- Pg 9, Ln 33: This could be my own lack of understanding – but I'm unclear on what the "cumulative" precipitation values represent. Either way - I'm not sure it adds considerable value to include this in the text and in Table 1. Percentile is likely sufficient.

⇨ Response: Rain event 'Percentiles' are a common way of characterizing the size of an individual event. The Cumulative statistics quantifies the percentage of the annual precipitation that contributed from storms which are smaller than or equal to the considered event. So, if GI is designed to control smaller storm sizes based on the results in this chapter it will control the majority of the total annual precipitation over the long term. We present the information to show the reader what the eight, 24hr events translate to in terms of sizes and percentages of annual rainfall. We added the following to the text of the Table 2. "Note: The Percentile and Cumulative, percentage-based statistics qualify the exceedance probability of each event and the relative contribution of events of similar size or lower to the annual rainfall, respectively."

- Pg 10: As mentioned in the "General Comments" – the SWMM model and its components/parameters should probably be introduced prior to the discussion of the 6 model set-ups

⇨ Response: Thanks for the suggestion. We switched the sections.

- Pg 10: This subwatershed set up was done for the entire watershed – may want to clarify this at some point. Same with the discussion of parameterization on Page 11

⇨ Response: We have added more text to clarify this matter throughout the manuscript.

- Pg 10, Ln 30: Why bother introducing the 16 land cover types at all? Just introduce the 10 the first time, and remove this later clarification.

⇨ Response: The 16 types were important for the original development of the spatial database. Because it is possible to examine other types of generalization or simplification of the land use, we included the original configuration. We did move the description of step to aggregate to 10 types to the Appendix.

- Pg 11, Figure 6: This figure seems too fundamental to the process of modelling to warrant inclusion. As mentioned: unless the paper is reframed as a "new method" for running SWMM with GI – I would remove this. I would actually advocate for this framing because it aligns better with the content (namely, all the nitty-gritty set up and parameterization details included). However, I'm not sure HESS would be the place for such a manuscript

⇨ Response: We somewhat agree, but, again, the figure was prepared in response to the first round of reviews. We think the new approach to GI modeling in SWMM is novel and useful to be beneficial to a more technical audience. We think that the other part of this comment comes from a lack of understanding of the relevance of the hypothetical analysis described above. This analysis supports our approach to SWMM subcatchment characterization, and this type of analysis is never available to SWMM modelers.

⇨ SWMM users that adopt our approach can use the HESS article as a reference to support its strength for balancing model complexity and output uncertainty. We think this will be primary utility of the paper over the long term.

- Pg 11, Ln 15-32 and Pg 12, Ln 1-24: There is far too much SWMM set up 101 in this section. Is it possible to reduce this to a table of parameter values? And just refer the user to the SWMM manual when describing what each parameter is and how it was determined?

⇨ Response: Many of the set-up details are important to fully understand what adjustments need to be made in the parameterization of SWMM for users interested in adopting the approach. For the more standard parameterizations that are not dependent on our new HRE-based approach to subcatchment parameterization, those have been moved to an Appendix. Again, much of the detail was added at the request of the first reviewers.

- Pg 12, Ln 31: Hillslope or watershed scale?

⇨ Response: We never intended to use the term 'hillslope' or present an explicit dichotomy between 'hillslope vs. watershed'. This is the construct that the Reviewer is using to gain context for the content presented. At a general level it works, but it is creating confusing as noted several times previously. We have used HRE, which is a single drainage area that discharges surface runoff to a storm sewer inlet. Because of urban drainage design, this does force HREs to represent a small scale, and it also should imply why we wouldn't want to conduct the analysis of how we translate the HRE concept to the SWMM model at the watershed scale.

- Pg 13, Ln 7: Here is evidence that one BPA width is probably not the best way to model this. Again – I'd advocate allowing all of the BPA to infiltrate water, and let the model physics determine how much of that water makes it to the subsurface – this seems more representative of reality

⇨ Response: The reviewer is confused about BPA. We've tried to make the description as explicit as we can. All the BPA does infiltrate water as does the SPA. BPA is a portion of the TPA. One of our main points is that all the TPA is not working as BPA as the Reviewer

seems to want to suggest. Our new approach accommodates this reality in the SWMM model set-up, and, therefore, makes the model more realistic.

- Pg 13, Ln 9-15: Refer to SWMM manual
⇨ Response: We disagree. We used the equation and descriptions to explain GW parameterization, thus it would be helpful for readers to see the related content. However, we have moved the section on groundwater parameterization to an Appendix.

- Pg 13, Ln 27: PCSWMM should be mentioned earlier in the manuscript if it is a critical step to the model set up
⇨ Response: We used PCSWMM just for importing GIS data in this study. This aspect is described in detail in Lee et al. 2017 (EPA Report). Since this is not relevant to the main objectives of this paper we will remove reference to PCSWMM.

- Pg 13, Figure 7: Is it possible to combine Figure 4 and 7?
⇨ Response: While it is possible, we think the resulting figure would be too cluttered we choose to keep the current figures as-is.

- Pg 14, Ln 9: BPA is not in Figure 6
⇨ Response: Corrected. It was supposes to be Figure 3.

- Ph 15, Ln 1: What is the spatial scale these simulations were done at? And what model set-up was used (out of the 6 tested)?
⇨ Response: We think this was clearly described in the manuscript: this is done for the entire watershed modeled by configuring HRE with Option 6. The six options were examined only for a hypothetical HRE.

- Pg 16, Ln1-10: See "General Comment" on including justification for back-calculating these values vs. taking model output directly. A good place to do with would be in the last paragraph of the Introduction on page 4
⇨ Response: As explained earlier, there is no provision in SWMM for acquiring all of the components gained from the hydrograph separation approach we presented in this manuscript.

Results and Discussion
- Pg 16, Ln 19ff: This not a result, it is a description of the watershed. Aren't all of these details in an EPA report somewhere (e.g., Lee, 2017)?
⇨ Response: Yes, 'it is a description of the watershed', but is the "result" of applying the approach that we describe for setting-up the spatial database. While the EPA report includes this description, the spatial information of the study site is critical to the content presented.

- Pg 16, Ln 22: What scale was the BPA calibration performed at? If watershed scale, then I would assume the July 2009 time period was used. What size storm events occurred in

this period? If they're all small – I would expect an underestimation of the BPA width. If it occurred at the hillslope scale – what design storms were used?

⇨ Response: As presented in the manuscript, model calibration was conducted for the study watershed using observed data. The actual storm intensities during the modeling period are also presented in Figure 11. We characterize the rainfall during the period in section 3.3 as "There was a total of 164.6 mm of rainfall during the three days of this period; this storm is smaller than the 1 yr return period design storm (61.0 mm/d) but larger than the 6 month storm (48.3 mm/d) based on the storm statistics for the study area (see Table 2)."

- Page 16, Ln 22: These results do not mean "that the runoff from ICIA is discharged to the adjacent 0.61 m of pervious area" – soften the language. E.g.: "a 0.61m buffer width best mimicked hydrologic behavior, evaluated by metric XYZ"
  - o Note: what was used to compare these runoff widths? Total runoff? Peak flow? NSE? Include this, it seems important.

⇨ Response: We amended as the reviewer suggests.

⇨ Channel flow is a combined discharge from surface runoff and subsurface flow (GW flow in SWMM) in both modeled and observed values. We compared the whole hydrographs based on $R^2$ and NSE as shown in Figure 11. By accident, both $R^2$ and NSE were omitted in the previous revision while those values were presented in the original manuscript. Now both values were included in the figure.

- Pg 17, Ln 1: What spatial scale did this occur at?

⇨ Response: This is based on the entire study watershed. Again, the modeling analysis for a hypothetical urban HRE with synthetic storms was just conducted to determine the most appropriate approach to configure an HRE in SWMM. No additional analysis was applied to the single HRE with design storms in this study.

- Pg 17-18: See General Comment #1 on the use of the word "accuracy" in this section 3.2

⇨ Response: See above. Accuracy is not an incorrect way to present the evaluation. Where appropriate we now use "presumed accuracy" to qualify the fact that we have no means of directly testing the assumptions we made regarding evaluation of the HRE configuration options in SWMM.

- Pg 18, Ln 17: Define MR5subs – it is important

⇨ Response: Defined.

- Pg 19, Ln 3-4: "Fig 10" >> "Fig 11"

⇨ Response: Corrected.

- Pg 19, Ln 1-10: 3% change in total runoff doesn't seem like a lot of sensitivity. This should be noted. How much did total runoff change during calibration?

⇨ Response: The text was amended to include the suggested point.

- Pg 19, Ln 13-24: Move calibration discussion of calibration ahead of discussion of sensitivity analysis. Alternatively, combine and condense these paragraphs.
  ⇨ Response: We made some changes here, but it makes more sense to us to discuss sensitivity before calibration.

- Pg 20, Ln 12: "Fig. 12b and 13" >> "Fig 13b and 14" Conclusions
  ⇨ Response: Corrected.

- Pg 21, Ln 3: "accuracy" is not the best word
  ⇨ Response: We feel otherwise: See above.

**2nd Reviewer:**

**Review of "Drainage area characterization for evaluating green infrastructure using the Storm Water Management Model" for HESS.**

This manuscript endeavors to solve a problem in the popular SWMM model, which is how to deal with the disconnection of impervious surfaces that result from green infrastructure (GI) implementation. The existing method is, at best, clunky when a high density of GI is modeled. The approach proposed by the authors seems like an improvement, but the manuscript gets so bogged down in the minutiae of the methodology that the reader is left not quite knowing which of the six options presented is the recommended solution. The manuscript toggles between synthetic storms applied within a small hypothetical area, with no calibration or validation (as everything is hypothetical and synthetic), and a calibrated (but not validated) SWMM model of a 100 ha suburban watershed. The introduction is good, but there is no discussion section to bring the reader back to the big picture questions that motivated the work and to identify remaining challenges and next steps. I expect high quality articles to provide such insights.

⇨ Response: We clearly disagree with the criticism about the structure and the quality of the content presented. We've tried to clarify the rationale and relevance of the hypothetical analysis. The goal of the discussion component was to describe the relevance of applying the proposed approach for configuring SWMM for GI modeling through demonstration at the watershed scale using a case study. Our intent was to demonstrate how the output from a SWMM model configured as we suggest could be valuable for GI design considerations. There is very little SWMM GI literature that can be drawn upon to provide context and depth to the discussion. Modelers using SWMM and considering GI effects rarely report the methods used to configure the GI simulation. We think we did the best we could do without overstating the relevance of the results.

In reviewing the previous reviewers' critiques and the authors I responses I note that there are lingering issues. A previous reviewer asked for a better description of the model calibration process, including criteria of performance. However, the only place I saw such a criteria was in the abstract where the Nash-Sutcliffe is reported. The authors response suggests that they view the sensitivity analysis of the synthetic storms in the hypothetical area as a calibration. It is not. The issue of higher peak flows in the watershed-scale GI scenario is still inadequately explained (see later comment). I note that previous reviewers did ask for more methodological detail. I feel that the level of methodological detail in the manuscript is now excessive in places. Perhaps this is a matter of personal preference, but I have made later comments about places where the detail seems (to me) to be more than sufficient.

⇨ Response: $R^2$ and NSE are presented in the main text as well (sentences before Figure 11; within Figure 11; and Conclusions).

⇨ We agree that "the level of methodological detail in the manuscript is now excessive in places". As expressed in the responses to the 1st reviewer's comments earlier, the details of the methodology are now presented in the Appendix to the revised manuscript.

**Specific Comments**

Abstract, line 17. It is unclear how the approach can be validated by comparing synthetic storms without having any reference to the hydrograph that would actually be produced by such events.

⇨ Response: Please see the responses to Reviewer 1 comments above.

Abstract, line 18. Tell the readers what the suggested approach is, because all they know is that six options were evaluated.
⇨ Response: Additional descriptions/clarifications have been added.

Abstract, line 19. I believe this is the only place where model calibration results are reported.
⇨ Response: Actually, we reported the estimated $R^2$ and NSE between the observed and the modeled in Section 3.3 (see Figure 11 and related contents) and Conclusions, in addition to the Abstract. The values for $R^2$ and NSE in Figure 11 were accidentally omitted in the previously revised manuscript while those values were presented in the original manuscript. The values have been included again in the figure.

p. 3, 1st paragraph: I think this section could be clearer in describing how SWMM does or does not include stormwater control measures (including GI) in the subcatchments in a typical setup. It's my understanding that runoff is generated from a subcatchment and then routed through one or more stormwater control elements before being added to the pipe or drainage network.
⇨ Response: So, we are supposed to delete content about how a SWMM Model is set-up in general, but here the Reviewer wants additional content on currently standard GI modeling practices in SWMM. In SWMM vernacular, the capture and retention of rainfall/runoff onsite are referred to as low impact development (LID) practices. While SWMM can explicitly model eight different generic types of LID controls, but there are some nuances and limitations to placing LID controls within a subcatchment. We consider an extended discussion of those nuances would distract the main discussion and would be outside the scope of this paper.
⇨ If the reviewer means the 'control elements' as detention/retentions systems before discharging the stormwater flow to the main receiving water (e.g., channels, streams, or rivers), those types of controls would be part of conventional downstream controls, not modern GI practices.

p. 3, line 25: Is homogeneous really what you are trying to say here? Or is it really that each subarea can consist of multiple different landcover components (e.g., DCIA can include paved streets, rooftops, driveways, and sidewalks)?
⇨ Response: We intended to mean the same as the reviewer.
⇨ We have amended the text to remove any potential misunderstandings.

p. 3, line 28: But HRE delineation isn't really done in SWMM in the sense that the drainage area isn't defined by topography or pipe networks. Instead the HRE characteristics are assigned to a particular GI, as you've described.
⇨ Response: We have defined/proposed HRE in this manuscript. The reviewer is limiting the definition of GI as a kind of (small) downstream control practice: a GI system receives runoff from an HRE to control onsite. But GI practices can be part of an HRE.

p. 3, line 32: It's true that the more subcatchments there are the more input and output values are created, but why is this appearing halfway through a discussion of HREs. This paragraph may

need some reorganization to help the reader understand the flow/overarching connective concept.

⇨ Response: We are unsure what the reviewer is asking us to address here.

p. 4, lines 1-5: I agree with this description of how adding GI to SWMM goes. It gets complicated really quickly!

⇨ Response: We appreciate reviewer's recognition on this matter.

p. 5, line 18: Having done this sort of work, it may be overstating the case that storm sewer inlets are often visible from aerial photographs. Image quality, shadows, parked cars, tree canopies, and other obstructions can make them very difficult to locate without on-the-ground verification.

⇨ Response: We amended the sentence to include the suggested clarification.

p. 6, line 4: It might be helpful to remind readers what BPA and SPA stand for here, since these acronyms are unique to this manuscript.

⇨ Response: That might be helpful, but the copy editors will typically remove any abbreviations that are spelled out again in the same manuscript. Therefore, we have placed a 'List of Abbreviations' at the end of the manuscript.

p. 7: A lot of the information on this page feels overly detailed, more like what I'd expect to find in a technical report or manual than a journal article.

p. 8-9: There is so much detail on the model setup in these pages that the reader who is not trying to replicate this exact experiment does not need to know. The flow of the story is entirely lost as the reader wades through these details. Can some of this detailed be moved to an appendix? Or is it already covered in the published technical report, so that can be referenced instead?

p. 11: The writing through here continues to be overly long and detailed. For example, at line 25, the authors write that characteristic width was estimated using an area-weighted flow length as recommended in Gironás et al. (2009). That would seem to be an adequate description for the purposes of a journal article, yet it continues for a full 10 lines of detail. Again, the risk here is that the reader misses the forest for the trees or loses interest in the paper entirely.

p. 12. The excessive detail just goes on. For example, it's not necessary to tell readers that the area-weighting was doing in Excel.

⇨ Response: We agree with these comments. As mentioned previously and noted by Reviewer 1 as well, we were asked to provide more detailed descriptions on the methods by the previous reviewers. We tried to respond to the request even though we felt this might be too much (or unnecessary). Unfortunately, the previous reviewers did not respond to our update and they were left in the manuscript.

⇨ Many of the details of the methods are now re-located under an Appendix.

p. 13, line 9. I am bothered by the use of the term interflow interchangeably with groundwater flow. In the usage with which I am familiar, interflow originates from the unsaturated or transiently saturated zone and is not synonymous with groundwater movement. We would not expect to adequately model interflow with the simple equation shown in equation 3. I suggest that the authors clearly define what they are attempting to model and then use the correct term throughout the manuscript

⇨ Response: In the revised manuscript, 'interflow' is entirely replaced with 'subsurface flow'.

p. 14, line 31. What was the metric used to quantify the modeling result in the calibration process? This is an important piece of missing information.

⇨ Response: The used metric is added. It is based on flow volume.

p. 15-16, Section 2.8. Here interflow and groundwater flow are used interchangeably in the context of hydrograph separation. In my experience, interflow would be considered part of stormflow while groundwater would contribute to baseflow. I suggest the authors revise their terminology to reduce confusion.

⇨ Response: Again, 'interflow' is replaced with 'subsurface flow' throughout the revised manuscript.

p. 16, bottom. Somewhere in here it seems important to explain what the model calibration results looked like and how the model parameters were judged to be sufficiently calibrated.

⇨ Response: The $R^2$ and NSE between the modeled and the observed are presented in Figure 11.

p. 18, line 9. Since the authors are comparing model results from simulated storms, I am curious how they concluded that some options produced "inaccurate" results, since there is no correct or observed results that can be tested against. (See also page 17, line 21.) At line 12 on page 18, the authors propose that Option 1 is the most accurate because it is the least spatially lumped. If that is the criteria for accuracy, that should be explained somewhere earlier in the paper (e.g., move this paragraph closer to the beginning of section 3.2). However, the problem with arguing that least spatially lumped is most accurate is that there are lots of parameters to be calibrated and we really don't know whether those parameters are correct or not.

⇨ Response: We explain more exactly the accuracy is gauged relative to the most spatially explicit option (See responses to Reviewer 1).

⇨ We added some clarification on this matter using literature citations as well.

p. 19, line 10. Less sensitive than what?

⇨ Response: Amended.

p. 20, line 22. An explanation of the physical or modeled processes that could generate higher peak flows as a result of downspout disconnection is required. I understand that the pervious area is saturated and producing runoff, but wouldn't that runoff end up in the same place as water from the connected downspouts would have? And wouldn't the connected downspout runoff, which travels continuously through pipes arrive at the stream faster or in a more concentrated fashion than runoff generated by overland flow through vegetation? I'm struggling to understand the mechanism here.

⇨ Response: We are assuming that there is a typo in the line and page reference for this comment (i.e., p. 20, line 22). As a description for the mechanism: Assume all downspouts from a rooftop are disconnected from the direct pipeline to the storm sewer (i.e., DCIA is converted into ICIA). If there is a rain storm with low intensities but an extended period, the pervious area that receives rooftop runoff (i.e., BPA) will be saturated earlier than the other pervious area (i.e., SPA) because of the extra run on from the rooftop. After saturation, the

saturated pervious area hydrologically responds more like impervious area (i.e., minimal loss by the saturated infiltration rate). Under this condition, the rooftop runoff cannot be hydrologically controlled by the BPA. However, if the downspouts are not disconnected (i.e., the rooftop is still DCIA), the pervious area, where possibly receives runoff from the rooftop if the downspouts are disconnected, can be maintained a hydrologically unsaturated status longer than BPA. If the pervious area is not saturated yet, the area can control the low intensity rain directly fallen onto the area (i.e., This area is part of SPA). Under this physical situation, the surface runoff rate with downspout disconnection can be higher than that with downspout connection because less stormwater can be controlled onsite with downspout disconnection. If this physical situation occurs at the peak rainfall intensity during a storm event, the peak surface runoff rate with downspout disconnection can be higher than that with downspout connection (i.e., the area of rooftop and the BPA may respond like impervious area increasing the peak flow). This would be a very special but possible case. Of course, the overall event-based surface runoff volume will be decreased with downspout disconnection in any cases.

Figure 1. The main portion of Figure 1 is not at the optimal scale to see either the study watershed's distribution of land uses or to understand its context relative to a major city. It seems like the scale of the map should aim to communicate at least one of those objectives.

⇨ Response: The closest major city, Cincinnati, is placed on the map with an additional description in the title of the figure.

Figure 5. It would be helpful to define "Trpt" and "Bldg" in the figure caption so that readers do not need to find this information in the manuscript text.

⇨ Response: "Trpt" and "Bldg" are defined at the bottom of the figure. Both are also presented in 'List of Abbreviations' at the end of the manuscript.

[revised manuscript text omitted]

**Commented [JGL1]:** Followed by the 1st reviewer's suggestion, the order of the previous sections 2.4 and 2.5 are switched in the revised manuscript.

[revised manuscript text omitted]

**Commented [JGL4]:** Followed by the 1st reviewer's suggestion, the order of the previous sections 2.4 and 2.5 are switched in the revised manuscript.

[revised manuscript text omitted]